# LEARNING RANDOMIZED REDUCTIONS

## ABSTRACT

A self-corrector for a function $f$ takes a black-box oracle computing $f$ that is correct on most inputs and turns it into one that is correct on every input with high probability. Self-correctors exist for any function that is randomly self-reducible (RSR), where the value $f$ at a given point $x$ can be recovered by computing $f$ on random correlated points. While RSRs enable powerful self-correction capabilities and have applications in complexity theory and cryptography, their discovery has traditionally required manual derivation by experts. We present Bitween, a method and tool for automated learning of randomized self-reductions for mathematical functions. We make two key contributions: First, we demonstrate that our regression-based learning framework with linear regression backend outperforms alternative backends including genetic algorithms, symbolic regression, and mixed-integer linear programming for discovering RSRs from correlated samples. Second, we introduce Agentic Bitween, a neuro-symbolic approach where large language models dynamically propose novel query functions for RSR property discovery, leveraging the inference and verification tools of Vanilla Bitween, moving beyond the fixed query functions ($x + r$, $x - r$, $x \cdot r$, $x$, $r$) previously used in the literature. On RSR-Bench, our benchmark suite of 80 scientific and machine learning functions, the linear regression backend surpasses alternative symbolic backends integrated in Vanilla Bitween, while Agentic Bitween discovers new RSR properties using frontier models.

## 1 INTRODUCTION

Random self-reducibility was first defined by Goldwasser & Micali (1984) in the context of worst-case to average-case reductions to show that concrete encryption schemes were hard on the average to break if the underlying problem was hard in the worst case. In subsequent work, Blum et al. (1993) introduced self-correcting programs, showing that a *self-corrector* can transform a program correct on most inputs into one correct on every input with high probability, using only black-box access. Such self-correctors exist for any *randomly self-reducible* (RSR) function, where $f(x)$ can be recovered by computing $f$ on random correlated points. RSRs have found applications in cryptography protocols (Goldwasser & Micali, 2019), average-case complexity (Feigenbaum & Fortnow, 1993), instance hiding (Abadi et al., 1987), result checkers (Blum et al., 1993), and interactive proof systems (Blum & Kannan, 1995; Goldwasser et al., 2019).

Yet for over four decades since Goldwasser and Micali's original work, discovering RSR properties has remained a manual, expert-driven process. Previous work (Blum et al., 1993; Rubinfeld, 1999) required manual derivation by experts, and existing methods are limited to a handful of fixed query functions: $x + r$, $x - r$, $x \cdot r$, $x$, and $r$. This severely restricts discoverable RSRs, as many functions require sophisticated patterns involving derivatives, integrals, or domain-specific transformations. The core challenge is to construct a hypothesis space with the right query functions and algebraic relationships that enable the reduction.

We present BITWEEN for automated RSR learning. Our approach samples programs on random values, uses regression with heuristics, attempts to find self-reductions, and formally verifies results. BITWEEN has two variants: Vanilla Bitween (V-BITWEEN) integrates different regression backends within the traditional fixed query functions setting, while Agentic Bitween (A-BITWEEN) depends on V-BITWEEN and employs large language models to dynamically discover novel query functions beyond the fixed set. On RSR-BENCH, our benchmark of 80 scientific and machine learning functions, V-BITWEEN demonstrates that our linear regression-based backend outperforms

```
1  double Π(double x) {
2    const int trms = 30;
3    double sum = 1.0;
4    double trm = 1.0;
5    double neg_x = -x;
6    for (int n = 1; n < trms; n++) {
7      trm *= neg_x / n;
8      sum += trm;
9    }
10   return 1.0 / (1.0 + sum);
11 }
```

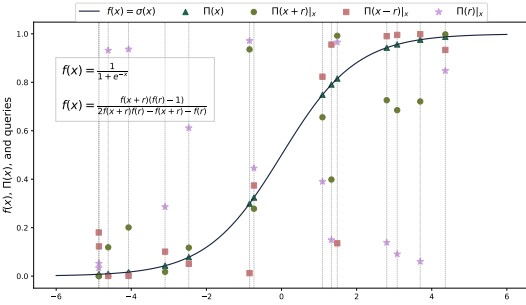

Figure 1: An approximate implementation of the sigmoid activation function used as oracle and a graph representing randomly selected points by BITWEEN which are then used to discover an RSR.

alternative backends including genetic programming (GPLearn (Stephens, 2016)), symbolic regression (PySR (Cranmer et al., 2023)), and mixed-integer programming (Gurobi (Gurobi Optimization, 2023)) for RSR discovery, while A-BITWEEN discovers new RSR properties using the inference and verification tools of V-BITWEEN, producing less false positives than pure neural approaches.

This work makes four key contributions: (1) Introducing a rigorous theoretical framework for learning randomized self-reductions with formal definitions and sample complexity analysis; (2) Demonstrating that the linear regression-based backend outperforms genetic programming, symbolic regression, and mixed-integer programming backends integrated in our framework for RSR discovery with fixed query functions; (3) Introducing LLM-based discovery of novel query functions beyond the standard set; and (4) Creating RSR-BENCH, a comprehensive benchmark for evaluating RSR discovery methods.

## 2 MOTIVATING EXAMPLE

In this section we illustrate the use of randomized self-reductions (RSR) and how BITWEEN computes them on the example of the sigmoid function, $\sigma(x) = 1/(1 + e^{-x})$. The sigmoid function is commonly used in neural networks (Han & Moraga, 1995). The program $\Pi(x)$ given in Figure 1 approximates the $\sigma(x)$ by using a Taylor series expansion. Here, $\Pi$ denotes the implementation of sigmoid function $\sigma$. Clearly, $\Pi(x)$ computes only an approximate value of $\sigma(x)$. We invoked BITWEEN on $\Pi$ and it derived the following RSR:

$$\sigma(x) = \frac{\sigma(x + r)(\sigma(r) - 1)}{2\sigma(x + r)\sigma(r) - \sigma(x + r) - \sigma(r)} \tag{1}$$

where $r$ is some random value. We verified that indeed the sigmoid function satisfies Equation (1). To the best of our knowledge, this is the first known RSR for the sigmoid function. In this example, BITWEEN inferred this RSR with 15 independent and random samples of $x$ and $r$. In the plot in Figure 1, we depict the value of $\Pi(x)$, identified on the graph with ▲, and the values of $\Pi(x + r), \Pi(x - r)$ and $\Pi(r)$, shown on the graph with ●, ■ and ⋆. Notice that all ▲'s are lying on the line depicting $\sigma(x)$.

Moreover, our derived RSR computes $\sigma(x)$ by using only $\sigma(x + r)$ and $\sigma(r)$, although the algorithm of BITWEEN is computing $\Pi$ on random (yet correlated) inputs $x + r, x - r$, and $r$. The learned RSR in Equation (1) can be used to compute the value of $\Pi(x)$ at any point $x$ by evaluating $\Pi(x + r)$ and $\Pi(r)$. Since $\Pi(x) \neq \sigma(x)$ for some values of $x$, randomized reductions can be used to construct self-correcting programs (Tompa & Woll, 1987; Blum et al., 1993; Rubinfeld, 1994).

The RSR in Equation (1) can also be used in an instance hiding protocol (Abadi et al., 1987). As an illustration, if a weak device needs to compute $\sigma$ on its private input $x$, it can do that by sending computing requests to two powerful devices that do not communicate with each other: the first powerful device computes $\sigma(r)$ with random $r$, and the second powerful one computes $\sigma(x + r)$. After receiving their outputs, the weak device can compute $\sigma(x)$ by evaluating Equation (1).

Additionally, in this particular case the derived RSR can be used to reduce the computation costs. If a weak device computes the value of $\sigma(r)$ beforehand and stores it as some constant $C$, then the

computation of $\sigma(x)$ simplifies to $\frac{\sigma(x+r)(C-1)}{2\sigma(x+r)C-\sigma(x+r)-C}$. While $x$ is a `double`, $x+r$ might require less precision.

## 3 RELATED WORK

*Symbolic Regression as Computational Backend.* Symbolic regression discovers mathematical expressions from data without assuming functional forms, using approaches ranging from genetic programming (Koza, 1992; Cranmer et al., 2023; Stephens, 2016), physics-inspired methods (Udrescu & Tegmark, 2020; Udrescu et al., 2020), to neural approaches (Petersen et al., 2021). Recent advances include RAG-SR (Zhang et al., 2025) (retrieval-augmented generation), ParFam (Scholl et al., 2025) (neural-guided continuous optimization), and MetaSymNet (Li et al., 2025) (adaptive tree-like networks).

Our task differs fundamentally: rather than discovering expressions from data, we seek randomized self-reduction properties for *known* mathematical functions. We use symbolic regression methods as computational backends; V-BITWEEN employs them within our regression-based learning framework to discover polynomial relationships among correlated query evaluations, while A-BITWEEN uses the inference and verification tools of V-BITWEEN with novel query functions. Within this framework, linear regression backend outperforms genetic programming (Stephens, 2016), symbolic regression (Cranmer et al., 2023), and MILP (Cozad & Sahinidis, 2018; Austel et al., 2017) backends, though these methods often timeout or produce approximations insufficient for RSR verification. Recent methods (Zhang et al., 2025; Scholl et al., 2025; Li et al., 2025) could potentially serve as alternative backends in future work.

*Mathematical Discovery and Neuro-Symbolic Learning.* Automated mathematical discovery dates back to AM (Lenat, 1976) and EURISKO (Lenat, 1983), with recent systems like the Ramanujan Machine (Raayoni et al., 2021) discovering mathematical constants and MathConcept (Davies et al., 2021) guiding mathematical intuition. Neural-symbolic integration has produced systems like Neural Module Networks (Andreas et al., 2016) and Neurosymbolic Programming (Chaudhuri et al., 2021). Recent LLM-based systems (GPT-4 (OpenAI, 2023), Claude (Anthropic, 2024), Llemma (Azerbayev et al., 2023)) demonstrate strong mathematical reasoning but suffer from hallucination. Our Agentic Bitween uniquely uses LLMs to discover novel query functions validated through symbolic regression, combining neural creativity with formal verification.

*Self-Correcting Algorithms and Property Inference.* The theoretical foundations of randomized self-reductions were established by Blum et al. (Blum et al., 1990; Lipton, 1991; Blum et al., 1993), showing that RSR properties enable self-correction for faulty programs. While program property inference tools like Daikon (Ernst et al., 2007) and DIG (Nguyen et al., 2012) discover invariants dynamically, they focus on program properties rather than mathematical functions and cannot discover complex randomized reductions. Our work provides the first practical system for automatically discovering RSRs, operationalizing decades-old theoretical results with a novel learning framework that discovers properties beyond the capabilities of existing tools.

## 4 THEORETICAL FOUNDATIONS

In this section, we give a definitional treatment of learning randomized self-reductions (RSRs). Our goal is to rigorously define the setting in which BITWEEN resides, which may of independent interest for future theoretical work. Throughout this section, we will say that a set $Z$ is *uniformly-samplable* if it can be equipped with a uniform distribution; we let $z \sim Z$ denote a uniformly random sample from $Z$. We start from a definition of randomized self-reductions (Goldwasser & Micali, 1984). Our presentation takes after Lipton (1989) and Goldreich (2017), modified for convenience.

**Definition 1** (Randomized self-reduction). *Fix a uniformly-samplable* input domain $X$, a *range* $Y$, *and uniformly-samplable* randomness domain $R$. Let $f\colon X \to Y$; $q_1, \ldots, q_k\colon X \times R \to X$ *(query functions); and* $p\colon X \times R \times Y^k \to Y$ *(recovery function) such that for all* $i \in [k]$ *and* $x \in X$, $u_i := q_i(x,r)$ *is distributed uniformly over* $X$ *when* $r \sim R$ *is sampled uniformly at random.*[1]

---

[1] Importantly, the $u_i$'s must only satisfy *marginal uniformity*, but may be correlated among themselves.

*We say that $(q_1, \ldots, q_k, r)$ is a* (perfect) randomized self-reduction (RSR) *for $f$ if for all $r \in R$, letting $u_i := q_i(x, r)$ for all $i \in [k]$, the following holds:*

$$f(x) = p\left(x, r, f(u_1), \ldots, f(u_k)\right). \tag{2}$$

*In other words, Equation (2) holds with probability 1 over randomly sampled $r \sim R$. For errors $\rho, \xi \in (0, 1)$, we say that $(q_1, \ldots, q_k, p)$ is a $(\rho, \xi)$-approximate randomized self-reduction ($(\rho, \xi)$-RSR) for $f$ if, for all but a $\xi$-fraction of $x \in X$, Equation (2) holds with probability $\geq 1 - \rho$ over the random samples $r \sim R$. That is,*

$$\Pr_{x \sim X} \left[ \Pr_{r \sim R} \left[ \begin{array}{c} f(x) = p(x, r, f(u_1), \ldots, f(u_k)) \\ \text{where } \forall i \in [k] \ u_i := q_i(x, r) \end{array} \right] \geq 1 - \rho \right] \geq 1 - \xi.$$

*Given a class of query functions $Q \subseteq X^{X \times R}$ and recovery functions $P \subseteq Y^{X \times R \times Y^k}$, we let $\mathrm{RSR}_k(Q, P)$ denote the class of functions $f \colon X \to Y$ for which there exist $q_1, \ldots, q_k \in Q$ and $p \in P$ such that $f$ is perfectly RSR with $(q_1, \ldots, q_k, p)$. We write $\mathrm{RSR}(Q, P)$ when the number of queries is irrelevant.*

In the literature (Lipton, 1989; Goldreich, 2017), the query functions are defined as randomized functions of the input $x$ to be recovered. That is, as random variables $\tilde{q}_i(x) \sim X$ rather than our deterministic $q_i(x, r) \in X$. The definitions are equivalent; we simply make the randomness in $\widetilde{q}_i$ explicit by giving it $r$ as input. This choice will have two benefits: (1) It makes explicit the amount of random bits used by the self-reduction, namely, $\log_2 |R|$. (2) It lets us think about the query functions as deterministic. This could allow one to relate traditional complexity measures (e.g., VC dimensions) of the function classes $Q$ and $P$ to those of the function $f$.

To elaborate more on the second point, let us restrict the discussion to polynomials over finite fields, i.e., $f \colon \mathbb{F}_n^m \to \mathbb{F}_n$ for some $n \in \mathbb{N}$. Lipton (1989) showed that even extremely simple choices of $Q$ and $P$ can be very expressive.

**Fact 2** (Lipton (1989)). *Any $m$-variate polynomial $f \colon \mathbb{F}_\ell^m \to \mathbb{F}_\ell$ of degree $d < \ell - 1$ is perfectly randomly self-reducible with $k = d + 1$ queries and randomness domain $R = \mathbb{F}_\ell^m$. Furthermore, the queries $q_1, \ldots, q_{d+1} \colon \mathbb{F}_\ell^{2m} \to \mathbb{F}_\ell^m$ and recovery function $p \colon \mathbb{F}_\ell^{2m+d+1} \to \mathbb{F}_n$ are linear functions.*

The question of interest is whether, given access to samples from $f \in \mathrm{RSR}_k(Q, P)$, it is possible to learn an (approximate) RSR for $f$. Before we can continue, we must first specify how these samples are drawn. Typically, learners are either given input-output pairs $(x, f(x))$ where $x$'s are either *independent random samples*, or chosen by the learner herself. Learning RSRs will occur in an intermediate access type, in which $x$'s are drawn in a correlated manner. We formally define these access types next. Our presentation is based on that of O'Donnell (2014), who, like us, is focused on samples drawn from the uniform distribution. We note that learning from uniformly random samples has been studied extensively in the literature (Hancock, 1993; Golea et al., 1996; Jackson et al., 2002; Klivans et al., 2004; Kucera et al., 1994; Verbeurgt, 1990; Jackson & Servedio, 2006).

**Definition 3** (Sample access). *Fix a uniformly-samplable set $X$, function $f \colon X \to Y$ and a probabilistic algorithm $\Lambda$ that takes as input $m$ (labeled) samples from $f$. We consider three types of sample access to $f$: (1) Independent random samples: $\Lambda$ is given $(x_j, f(x_j))_{j=1}^m$ for independent and uniformly sampled $x_j \sim X$. (2) Correlated random samples: $\Lambda$ is given $(x_j, f(x_j))_{j=1}^m$ drawn from a distribution such that, for each $j \in [m]$, the marginal on $x_j$ is uniformly random over $X$. However, different $x_j$'s may be correlated. (3) Oracle queries: During $\Lambda$'s execution, it can request the value $f(x)$ for any $x \in X$. The type of sample access will be explicitly stated, unless clear from context.*

**Remark 4.** *Facing forward, we note that more restrictive access types will correspond to more challenging settings of learning (formally defined in Theorem 5). Intuitively (and soon, formally), if $F$ is learnable from $m$ independent samples, it is also learnable from $m$ correlates samples, and $m$ oracle queries as well.*

Learning from correlated samples is one of two main theoretical innovations introduced in this work (the other will appear shortly). We note that PAC learning (Valiant, 1984) requires learning under *any* distribution $\mu$ of inputs over $X$, whereas we consider learning only when samples are drawn from the uniform distribution over $X$ (with possible correlation). Learning from correlated samples could be adapted to arbitrary distributions $\mu$ by considering correlated samples $(x_j, f(x_j))_j$ such

that the marginal on each $x_j$ is distributed according to $\mu$. This interesting setting is beyond the scope of this work.

Finally, for an input $x$ we will use $n = |x|$ to denote its. This will allow us to place an *efficiency requirement* on the learner (e.g., polynomial time in $n$). For a class of functions $F$ from inputs $X$ to outputs $Y$, we will use $F_n$ to denote the class restricted to inputs of length $n$, and similarly we let $Q_n$ (resp. $P_n$) denote the restriction of the query (resp. recovery) class.[2]

**Definition 5** (Learning RSR). *Fix a reduction class $(Q, P) = (\bigcup_n Q_n, \bigcup_n P_n)$ and a function class $F = \bigcup_n F_n$ where $F_n \subseteq \mathrm{RSR}_k(Q_n, P_n)$ for some constant $k \in \mathbb{N}$.[3] A $(Q, P, k)$-learner $\Lambda$ for $F$ is a probabilistic algorithm that is given inputs $n \in \mathbb{N}$ and $m$ samples of $f \in F_n$, collected in one of the three ways defined in Theorem 3. $\Lambda$ outputs query functions $q_1, \ldots, q_k \in Q_n$ and a recovery function $p \in P_n$.*

*We say $F$ is $(Q, P)$-RSR$_k$-learnable if there exists a $(Q, P, k)$-learner $\Lambda$ such that for all $f \in F$ and $\rho, \xi, \delta \in (0, 1)$, given $m := m(\rho, \xi, \delta)$ labeled samples from $f$, with probability $\geq 1 - \delta$ over the samples and randomness of $\Lambda$, $\Lambda$ outputs $q_1, \ldots, q_k \in Q$ and $p \in P$ that are $(\rho, \xi)$-RSR for $f$.*

*We say that $F$ is* efficiently *$(Q, P)$-RSR$_k$-learnable if $\Lambda$ runs in time $\mathrm{poly}(n, 1/\rho, 1/\xi, 1/\delta)$. The function $m(\rho, \xi, \delta)$ is called the* sample complexity *of the learner $\Lambda$. We will omit the $(Q, P)$ prefix when it is clear from context.*

Theorem 5 takes after the classic notion of Probably Approximately Correct (PAC) learning Valiant (1984) in that it allows the learner a $\delta$ failure probability, and asks that the learned $q_1, \ldots, q_k, r$ only approximately recover $f$. The main difference is that in, Theorem 5, $\Lambda$ is required to output an approximate RSR for $f$, whereas in PAC learning it is required to output a function $\hat{f} \in F$ that approximates $f$ itself; that is, such that $\hat{f}(x) = f(x)$ with high probability over $x \sim X$.

For a detailed comparison between RSR learning and PAC learning, including claims about their relative strengths and sample complexity relationships, see Theorem 6 and Theorem 7, and their proofs in the theory appendix. The Fundamental Theorem of Learning states that sample complexity of PAC-learning is tightly characterized by the VC-dimension of the function class $F$. In the future, it would be interesting to obtain an analogous Fundamental Theorem of RSR-Learning, which relates the sample complexity to "intrinsic" dimensions of $Q$, $P$, and $F \subseteq \mathrm{RSR}(Q, P)$.

## 5 BITWEEN: LEARNING RANDOMIZED SELF-REDUCTIONS

*Algorithm Overview.* BITWEEN expects correlated sample access to a program $\Pi$ which is an alleged implementation of some unknown function $f$. The goal is to learn an RSR for $f$ following our theoretical framework (Theorem 5). BITWEEN is given the input domain $X$, the class of query functions $Q$, and recovery function degree bound $d$. At a high level, BITWEEN works as follows: (i) generate all monomials of degree $\leq d$ over symbolic variables $\Pi(q(x, r))$ for each query function $q \in Q$; (ii) construct a linear regression problem with a regressand for each monomial; (iii) query $\Pi(x_i)$ and $\Pi(q(x_i, r_i))$ for random $x, r \in X$; (iv) fit the regressands to the samples using sparsifying linear regression, thereby eliminating most monomials; (v) apply rational approximation to convert floating-point coefficients to interpretable rational forms. Algorithm 1 provides the complete algorithm description.

*Regression Formulation and Loss Function.* Bitween formulates RSR discovery as a supervised learning problem by treating each query function $q \in Q$ as a potential target variable. For each query $q$, we construct a regression problem where $\Pi(q(x_i, r_i))$ serves as the dependent variable and the monomials $V(x_i, r_i)$ over all query evaluations serve as features. The loss function for a given target query $q$ is:

$$\mathcal{L}_q(\mathbf{C}) = \frac{1}{m} \sum_{i=1}^{m} \left( \Pi(q(x_i, r_i)) - \sum_{V \in \mathcal{MON}} C_V \cdot V(x_i, r_i) \right)^2 + \lambda R(\mathbf{C}) \tag{3}$$

---

[2]This slight informality will allow us to avoid encumbering the reader with a subscript $n$ throughout the paper.

[3]The requirement that $F_n \subseteq \mathrm{RSR}_k(Q_n, R_n)$ is a *realizability assumption*. We leave the agnostic setting, in which $F_n \not\subseteq \mathrm{RSR}_k(Q_n, R_n)$, to future work.

---

**Algorithm 1:** V-BITWEEN modulo regression

---

**Input:** Program $\Pi$, query class $Q$, recovery function degree bound $d$, input domain $X$, sample complexity $m$.
**Output:** Randomized self-reduction $(q_1, \ldots, q_k, p)$ or $empty\_tuple$.

1   For each query function $q \in Q$, initialize a variable $v_q$.      `// v_q for Π(q(x,r))`

2   Let MON be all monomials of degree at most $d$ over the variables $(v_q)_{q \in Q}$.

3   For each monomial $V \in$ MON, initialize the regressand $C_V$. `// We will fit Π(x) = Σ_V C_V · V(x,r)`

4   For each $i \in [m]$, sample input $x_i \in X$ and randomness $r_i \in X$.

5   Query $\Pi$ for the values $\Pi(x_i)$ and $\Pi(q(x_i, r_i))$ for each $q \in Q$.

6   Fit the regressands $C_V$ to the equations

$$\Pi(x_1) = \sum_{V \in \text{MON}} C_V \cdot V(x_1, r_1), \quad \ldots, \quad \Pi(x_m) = \sum_{V \in \text{MON}} C_V \cdot V(x_m, r_m)$$

  using sparsifying linear regression. Let $\widehat{C_V}$ denote the fitted regressands.

7   Apply rational approximation to convert each fitted coefficient $\widehat{C_V}$ to its best rational form $\widetilde{C_V}$ using maximum denominator constraint.      `// Convert to rationals`

8   Initialize an empty set of query functions $\widehat{Q} \leftarrow \emptyset$.

9   **foreach** $V \in$ *MON* **do**

10     **if** $\widetilde{C_V} \neq 0$ **then**

11        Add to $\widehat{Q}$ all queries $q$ such that the variable $v_q$ appears in the monomial $V$.

12   Let $(\widehat{q_1}, \ldots, \widehat{q_k}) \leftarrow \widehat{Q}$ where $k = |\widehat{Q}|$. For each $\widehat{q_i}$, let $\widehat{v_i}$ denote its corresponding variable defined in Line 1.

$$\text{Define the recovery function,} \quad \widehat{p}(x, r, \widehat{v_1}, \ldots, \widehat{v_k}) := \sum_{V : \widetilde{C_V} \neq 0} \widetilde{C_V} \cdot V(x, r).$$

  **return** *The randomized self-reduction* $(\widehat{q_1}, \ldots, \widehat{q_k}, \widehat{p})$ *or* $empty\_tuple$.

---

where $\lambda > 0$ is the regularization parameter (selected via grid search), and $R(\mathbf{C}) = \|\mathbf{C}\|_1$ for Lasso or $R(\mathbf{C}) = \|\mathbf{C}\|_2^2$ for Ridge. Sparsification proceeds iteratively: after initial regression, monomials with coefficients below threshold are eliminated, and regression is repeated on reduced space until convergence. The optimization problem:

$$\widehat{\mathbf{C}}_q = \arg\min_{\mathbf{C}} \mathcal{L}_q(\mathbf{C}) \tag{4}$$

The term "modulo regression" in Algorithm 1's caption reflects that the regression step (line 6) can use any backend. Vanilla V-BITWEEN uses Linear Regression, Ridge, and Lasso (collectively V-BITWEEN-LR), while PySR, GPLearn, and MILP serve as alternative backends (V-BITWEEN-PySR, V-BITWEEN-GPLearn, V-BITWEEN-MILP). This modularity enables comparing optimization paradigms.

*Three-Tier Experimental Framework.* Our evaluation encompasses three distinct approaches to RSR discovery. Vanilla Bitween (V-BITWEEN), symbolic, is our core regression-based learning framework. We integrate and compare our linear regression-based backend, V-BITWEEN-LR to V-BITWEEN-MILP (Mixed-Integer Linear Programming backend), V-BITWEEN-PySR (PySR symbolic regression backend), and V-BITWEEN-GPLearn (GPLearn genetic programming backend) within the traditional fixed query function paradigm.

Agentic Bitween (A-BITWEEN), Neuro-Symbolic, represents our breakthrough approach where large language models dynamically discover novel query functions beyond the fixed set $\{x + r, x - r, x \cdot r, x, r\}$ that in turn lead to new properties. The LLM agents are queried only once and have in their disposal, aside from their mathematical background knowledge, the following three tools. The $symbolic\_verify\_tool$ (Section G.2.2) can be used to verify that a property equals to zero using the simplification method of SymPy (Meurer et al., 2017). The $infer\_property\_tool$ (Section G.2.1) can use different V-BITWEEN backends (V-BITWEEN-LR is the default) and can be provided with functional terms to discover new properties. The provided functional terms correspond to the variables $v_q$ of Algorithm 1 that implicitly encapsulate the query functions. For instance, the Inverse function in Table 1 shows a functional term $f(\frac{x}{x+1})$, with the query function $\frac{x}{x+1}$. The third tool, $sequential\_thinking\_tool$ (Model Context Protocol Community, 2024) allows the LLM to journal its thoughts and was empirically found useful for increasing the other tools' usage and enhancing the

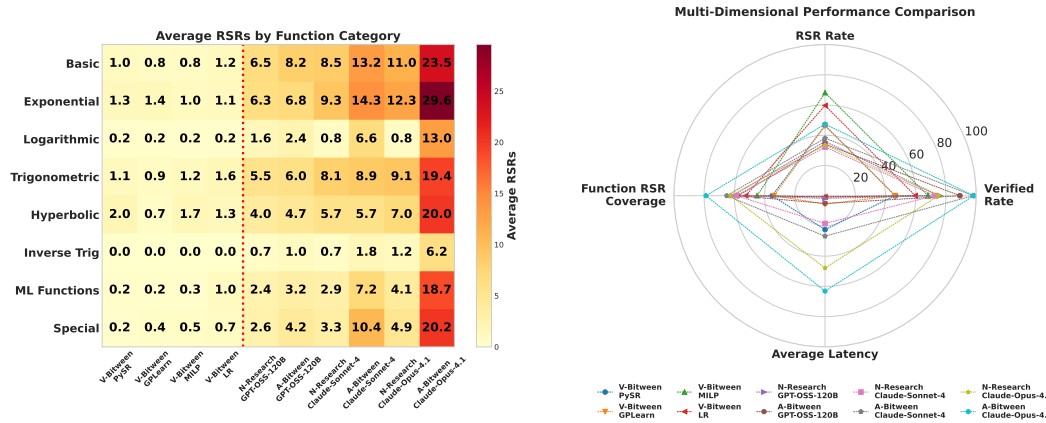

Figure 2: (Left) Performance heatmap of average verified RSRs across mathematical function categories. The dotted red line represents the boundary from symbolic to neural methods. (Right) Multi-dimensional performance comparison for different methods. Verified Rate and RSR Rate is the percentage of the total properties proposed by the method that are verified and verified (manually found) RSRs, respectively. Function RSR Coverage is the percentage of individual functions (benchmarks) for which the method returned at least one verified RSR.

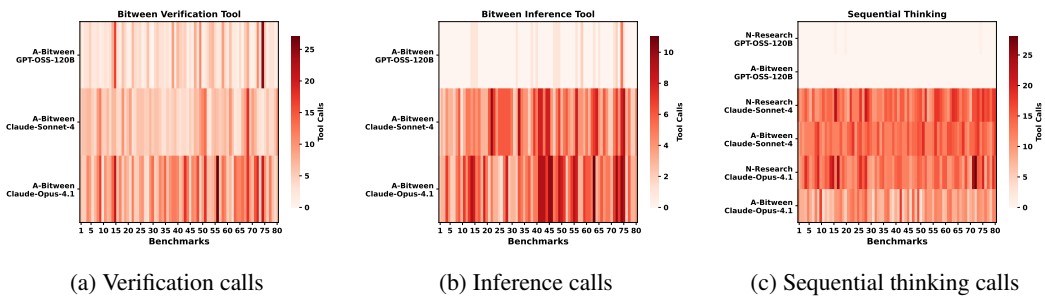

(a) Verification calls       (b) Inference calls       (c) Sequential thinking calls

Figure 3: A-BITWEEN's intensive tool usage across all the benchmarks. Particularly useful proved the *sequential_thinking_tool* for guiding the exploration and helping the LLM agent.

LLM's response. The goal of the LLM is to utilize its knowledge and the provided tools to discover as many verified properties as it can. Finally, Neural Research (N-RESEARCH), Pure Neural, serves as a baseline approach using LLM reasoning with access only to the *sequential_thinking_tool*, providing a comparison point to demonstrate the value of our neuro-symbolic integration.

*Key Technical Components.* The implementation involves several involved components detailed in Appendix Section E. We first perform *supervised learning conversion* to transform the unsupervised RSR discovery problem into supervised regression, followed by *cross-validation* using grid search with 5-fold cross-validation for hyperparameter optimization. *Sparsification* through iterative dimensionality reduction eliminates irrelevant terms, while *rational approximation* converts floating-point coefficients to interpretable rational forms. We employ *property testing* to verify discovered RSRs on held-out test data and *formal verification* using symbolic execution tools for rigorous validation. Finally, our *library setting* approach constructs complex RSRs using elementary function properties. The complexity analysis shows that for low-degree polynomials, BITWEEN operates in polynomial time $O(t_{\text{terms}} \times n_{\text{samples}} \times n_{\text{features}}^2)$, though the exponential growth of monomials with degree necessitates careful *degree* and *query* selection in practice.

## 6 EMPIRICAL EVALUATION

We evaluate Bitween on RSR-Bench, a benchmark suite of 80 mathematical functions spanning scientific computing and machine learning applications. Our evaluation is structured to directly

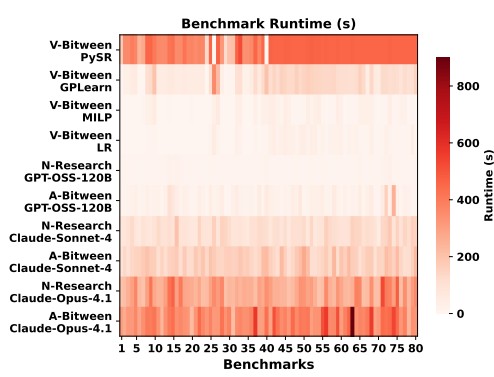 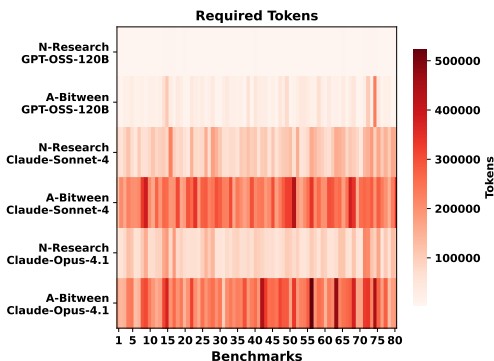

(a) Runtime performance heatmap across all methods and benchmarks.

(b) Token usage heatmap demonstrating tool-based reasoning overhead.

Figure 4: (Left) Runtime performance across methods and benchmarks highlighting the speed of V-BITWEEN-LR, V-BITWEEN-MILP and the overhead of A-BITWEEN variants. (Right) Token usage patterns correlate with the LLM's ability to efficiently utilize them.

validate the three key contributions outlined in Section 1: (1) demonstrating V-BITWEEN-LR's effectiveness over the other backends with fixed query functions, (2) showcasing A-BITWEEN's advances in discovering novel query functions, and (3) comprehensive analysis across RSR-Bench demonstrating both symbolic method advantages and the transformative impact of dynamic query discovery.

*RSR-Bench Benchmark Suite* We constructed RSR-Bench comprising 80 continuous real-valued functions from diverse mathematical domains: basic arithmetic (linear, squared, cube), exponential and logarithmic functions, trigonometric and hyperbolic functions, inverse trigonometric functions, machine learning activation functions (sigmoid, ReLU, GELU, etc.), loss functions, and special mathematical functions (gamma, error function, Gudermannian). Some benchmarks include ground-truth RSR properties with minimal query complexity, sourced from self-testing literature (Blum et al., 1993; Rubinfeld, 1999) and functional equation theory (Aczél, 1966; Kannappan, 2009).

*Baseline Methods* We integrate and compare the following regression backends: *PySR* (Cranmer et al., 2023) (symbolic regression using evolutionary algorithms), *GP-Learn* (Stephens, 2016) (genetic programming for symbolic regression), *MILP* (mixed-integer linear programming) with Gurobi solver (Gurobi Optimization, 2023), and compare the agentic variants with pure *Neural Research* (with access only to the $sequential\_thinking\_tool$) using GPT-OSS-120B (OpenAI, 2025), Claude-Sonnet-4 (cla, 2025), and Claude-Opus-4.1 (Anthropic, 2024)).

*Configuration* Experiments were conducted on a MacBook Pro with 32GB memory and Apple M1 Pro 10-core CPU. For trigonometric, hyperbolic, and exponential functions, we configured term generation up to degree 3; for others, degree 2. We used uniform sampling in [-10, 10] with error bound $\delta = 0.001$. Each experiment was repeated 5 times for statistical significance.

*Performance Comparison.* The heatmap in Figure 2 (left) provides a performance analysis across mathematical function categories and methods and measures the average number of found verified RSRs per function category. This heatmap reveals that the neural variants always return more RSRs on average than the symbolic ones. This is noticeable after the red dotted boundary. The A-BITWEEN variants outperform their pure neural counterparts, while V-BITWEEN-LR in total returns more RSRs on average followed by V-BITWEEN-MILP.

The radar chart in Figure 2 (right) demonstrates the performance of different methods across four main dimensions. The Verified Rate shows the percentage of the properties that a method proposed that pass the verification (regardless of them being RSRs or not). In this dimension A-BITWEEN variants have the highest score, meaning that they return less false positives. The RSR Rate shows the percentage of the properties that a method proposed that pass the verification and are RSRs (via manual introspection). Since, this is a percentage the methods that return more certain results are favored. Thus symbolic methods like V-BITWEEN-LR and V-BITWEEN-MILP score the highest

Table 1: Novel query functions discovered by Agentic Bitween beyond traditional fixed query functions $\{x + r, x - r, x \cdot r, x, r\}$. The query functions appear inside functional terms. For example, the functional term $f(x + log(k))$ has query function $x + log(k)$.

| Function | Properties that contain novel query functions discovered by Agentic Bitween |
|---|---|
| Sigmoid | $f(x + \log(k)) - \frac{k \cdot f(x)}{1 + (k-1) \cdot f(x)} = 0$; $f(x) \cdot f(y) - f(\frac{x \cdot y}{x+y}) \cdot f(x + y - x \cdot y) = 0$ |
| Gudermannian | $\tan(f(x + y)) - \frac{\sinh(x) + \sinh(y)}{1 - \sinh(x)\sinh(y)} = 0$; $f(x + y) - \arctan(\sinh(x)\cosh(y) + \cosh(x)\sinh(y)) = 0$ |
| GELU | $f(x) + f(-x) - x \cdot \mathrm{erf}(x/\sqrt{2}) = 0$; $f(x) \cdot f(-x) - 0.25 \cdot x^2 \cdot (\mathrm{erf}(x/\sqrt{2})^2 - 1) = 0$ |
| Inverse | $f(\frac{x}{1+x}) + x = 0$; $f(1 - \frac{1}{x}) - (1 - x) = 0$; $f(\frac{x \cdot y}{x+y}) - f(x) \cdot f(y) = 0$ |
| Hyperbolic | $f(x) \cdot f(y) - f(\sqrt{x^2 + y^2}) = 0$; $f(x + r) \cdot f(x - r) - f(x)^2 \cdot f(r)^2 = 0$ |
| Logarithmic | $f(x^n) - n \cdot f(x) = 0$; $f(\sqrt{x \cdot r}) - \frac{f(x) + f(r)}{2} = 0$; $f(x^{a \cdot b}) - a \cdot b \cdot f(x) = 0$ |

here. A percentage of the returned properties from the neural methods are duplicates, trivial, or not RSRs, thus they lose some points here, but the best agentic variant, A-BITWEEN-Claude-Opus-4.1 follows next. The function RSR coverage shows the percentage of individual benchmarks for which the methods returned at least one (verified) RSR. In this dimension A-BITWEEN outperforms the others, since it handles a more diverse set of benchmarks. Also, the neural variants outperform the symbolic ones, out of which V-BITWEEN-LR comes first. Finally, the average latency shows the runtime percentage relevant to a maximum limit (600sec for symbolic methods, 1800sec for neural methods) and shows that on average the neural methods take more time, while V-BITWEEN-PySR tends to be slower than the other methods.

A-BITWEEN*'s Novel Query Functions.* A-BITWEEN's ability to discover novel query functions beyond the traditional fixed set $\{x + r, x - r, x \cdot r, x, r\}$ is a key factor for finding more properties (including RSRs). Table 1 showcases the new kinds of query functions discovered by A-BITWEEN. At this point it is worth clarifying what a query function is. According to Algorithm 1 the query functions $q$ comprise the query class $Q$, so that they can be used as arguments to the program $\Pi$ to generate the variables $v_q = \Pi(q(x, .))$. Practically, however, A-BITWEEN does not provide the queries $q$, but the variables $v_q$. Specifically, it needs to provide expressions like $f(x), f(x + r), ...,$ where $f$ is usually used instead of $\Pi$. Thus, the main utility of A-BITWEEN is to come up with and provide such variables $v_q$ (also called functional terms) that can be passed to the rest of the algorithm using the $infer\_property\_tool$. In that sense, the query functions are implicit as the arguments of the function $f$. Given this setup, A-BITWEEN can also provide some other unique and useful independent values. Looking at Table 1 we can see many unique query functions: $x + log(k)$ in Sigmoid's functional term $f(x + log(k))$, a composite term $x \cdot erf(x/\sqrt{2})$ in GELU, which is one subpart of the implementation different than $f$, a query function $\frac{x \cdot y}{x+y}$ in Inverse's functional term $f(\frac{x \cdot y}{x+y})$ and so on. The $symbolic\_verify\_tool$ also helps with the discovery, because of the way it handles failures. In the case that a proposed property does not pass verification, its simplified expression (that does not equal zero) is returned back to A-BITWEEN. In many cases we noticed, that this feedback provides the missing piece to complete the equation, because the remaining expression can be subtracted from the original one in order for the equation to be zero. Figure 3 shows the that A-BITWEEN-Claude-Opus-4.1, which was the best of its variant, utilized both tools more, something that led to more discovered properties.

*Computational Efficiency Analysis.* Figure 4 provides insights on the computational efficiency across all methods and benchmarks. The runtime heatmap (left) reveals that V-BITWEEN-LR and V-BITWEEN-MILP run the fastest, whereas V-BITWEEN-PySR requires significantly more time for comparable RSR discovery. The A-BITWEEN variants (which are queried only once) tend to run for more compared to their pure neural counterparts, because the tool usage prolongs their exploration. The token usage heatmap (right) shows that A-BITWEEN requires more computational resources than pure neural variants, expected due to tool-based reasoning overhead. This increased token consumption stems from several factors: iterative tool interactions with V-BITWEEN, feedback loops from unsuccessful tool calls that require re-reasoning, and the inherent complexity of discovering novel mathematical relationships. On the positive side, this leads to more verified properties that contain a diverse set of query functions and can also tackle previously intractable functions.

*Limitations.* While our evaluation demonstrates significant advances, several limitations merit discussion. First, discovered RSR properties may contain redundancies where certain properties are algebraically derivable from others. Although Gröbner basis reduction (Buchberger, 2006; Cox et al., 2013) could identify minimal generating sets, we refrained from applying it due to its NP-complete complexity, especially given the large number of RSRs discovered. Second, our approach is inherently incomplete, absence of discovered RSRs does not imply non-existence. The infinite space of possible query functions, numerical precision requirements, sampling strategies, and degree limits (2-3 in our experiments) constrain discovery. Functions without discovered RSRs may possess properties requiring higher-degree terms or alternative mathematical representations beyond our current framework. Finally, A-BITWEEN's enhanced performance incurs 5-10x higher token usage than pure neural baselines due to iterative tool interactions, though this overhead is justified by the improvement in the quantity and diversity of discovered RSRs.

## 7 CONCLUSION

Bitween provides the first systematic approach for learning RSRs from mathematical functions, transforming expert-driven discovery into an automated process. Our work delivers two key achievements that validate our contributions: First, V-BITWEEN demonstrates that the linear regression-based backend surpasses other symbolic method backends including genetic algorithms, symbolic regression, and mixed-integer programming for RSR discovery from correlated samples. Second, A-BITWEEN achieves a paradigm shift by dynamically discovering novel query functions through neuro-symbolic reasoning, moving beyond a fixed query set. On RSR-Bench's 80 functions spanning scientific computing and machine learning, automated RSR discovery surpasses traditional methods while maintaining verification rigor.

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

## A  Theory: RSR Learning vs PAC Learning

When samples are drawn uniformly and *independently* (Item 1 in Theorem 3), PAC learnability is a strictly stronger form of learning than RSR learnability, as captured by the following two claims:

**Claim 6.** *Fix query class $Q$, recovery class $P$, and function class $F \subseteq \mathrm{RSR}_k(Q, P)$. If $F$ is (Uniform) PAC-learnable with sample complexity $m_{\mathrm{PAC}}(\varepsilon, \delta)$, then it is RSR-learnable with sample complexity*

$$m_{\mathrm{RSR}}(\rho, \xi, \delta) \leq m_{\mathrm{PAC}}(\min(\rho/k, \xi), \delta).$$

*However, the RSR-learner may be inefficient.*

We note that Theorem 6 is trivial when considering a class $F \subseteq \mathrm{RSR}_k(Q, P)$ characterized by a single RSR, i.e., such that there exist $q_1, \ldots, q_k \in Q$ and $p \in P$ such that Equation (2) holds with probability 1 for all $f \in F$. For one, this follows because the RSR-learner does not need any samples and may simply output $q_1, \ldots, q_k, p$.[4] This gives rise to the following claim.

**Claim 7.** *There exist classes $(Q, P) = (\bigcup_n Q_n, \bigcup_n P_n)$ and $F = \bigcup_n F_n \subseteq \mathrm{RSR}(Q, P)$ such that $F$ is efficiently RSR-learnable from 0 samples, but for any $\varepsilon \leq 1/2$, Uniform PAC-learning $F_n$ requires $m_{\mathrm{PAC}}(\varepsilon, \delta) \geq n$ oracle queries.*

Consequentially, Uniform PAC-learning $F_n$ requires at least $n$ correlated or independent random samples (see Theorem 4).

## B  Proofs

*Proof of Theorem 6.* Suppose $F \subseteq \mathrm{RSR}_k(Q, P)$ is PAC-learnable with sample complexity $m_{\mathrm{PAC}}(\varepsilon, \delta)$ and learner $\Lambda_{\mathrm{PAC}}$. To RSR-learn $F$ with errors $(\rho, \xi, \delta)$, we let $\varepsilon = \min(\rho/k, \xi)$ and draw $m = m_{\mathrm{PAC}}(\varepsilon, \delta)$ labeled samples $(x_i, f(x_i))_{i=1}^m$. The RSR-learner is described in Algorithm 2. For simplicity of notation, we will omit the input length $n$; following this proof is a brief discussion of Algorithm 2's inefficiency with respect to $n$.

---

**Algorithm 2:** RSR-learning via PAC-learning.

**Input:** Query class $Q$, recovery class $P$, and hypothesis class $F \subseteq \mathrm{RSR}_k(Q, P)$. Query complexity $k$ and randomness domain $R$. Uniform PAC-learner $\Lambda_{\mathrm{PAC}}$ for $F$. Labeled samples $(x_i, y_i)_{i=1}^m$.

**Output:** Query functions $\hat{q}_1, \ldots, \hat{q}_k \in Q$ and recovery function $\hat{p} \in P$.

1 Invoke $\Lambda_{\mathrm{PAC}}$ on samples $(x_i, y_i)_{i=1}^m$ to obtain a hypothesis $\hat{f} \in F$.
2 **foreach** *Query functions $(q_1, \ldots, q_k) \in Q^k$ and recovery function $p \in P$* **do**
3     **foreach** $x \in X$ *and* $r \in R$ **do**
4         Compute $u_i := q_i(x, r)$ for each $i \in [k]$.
5         **if** $\hat{f}(x) \neq p(x, r, \hat{f}(u_1), \ldots, \hat{f}(u_k))$ **then**
6             Go to line 2.         // Continue to the next $q_1, \ldots, q_k, p$.
7     Output $(\hat{q}_1, \ldots, \hat{q}_k, p) := (q_1, \ldots, q_k, p)$.
8 Output $\perp$.

---

At a high level, the learner invokes $\Lambda_{\mathrm{PAC}}$ to obtain a hypothesis $\hat{f} \in F$ that, is $\varepsilon$-close to the ground truth function $f$ (with probability $\geq 1 - \delta$ over the samples). It then uses $\hat{f}$ to exhaustively search through possible query functions $\hat{q}_1, \ldots, \hat{q}_k \in Q$ and recovery functions $\hat{p} \in P$, until it finds those that are a *perfect* RSR for $\hat{f}$.

To conclude the proof, we will show that, because $\hat{f}$ is $\varepsilon$-close to $f$, then $(\hat{q}_1, \ldots, \hat{q}_k, \hat{p})$ is a $(\rho, \xi)$-RSR for $f$.

---

[4] For a more nuanced reason, note that the sample complexity bound $m_{\mathrm{PAC}}(\rho/k, \delta)$ trivializes: Any class $F$ characterized by a single RSR has *distance* at least $1/k$, meaning that $\Pr_{z \sim X}[\hat{f}(z) \neq f(z)] \geq 1/k$ (Goldreich, 2017, Exercise 5.4). Thus, for any $\varepsilon = \rho/k < 1/k$, $f$ is the only function in $F$ that is $\varepsilon$-close to itself. In other words, learning within accuracy $\varepsilon$ amounts to exactly recovering $f$.

We say that $x \in X$ is *good* if $\hat{f}(x) = f(x)$. By choice of $\varepsilon$, we know that there are at least $(1 - \varepsilon)|X| \geq (1 - \xi)|X|$. It therefore suffices to show that for all good $x$,

$$\Pr_{r \sim R} \left[ \begin{array}{l} f(x) = \hat{p}(x, rf(u_1), \ldots, f(u_k)) \\ \text{where } \forall i \in [k]\ u_i := \hat{q}_i(x, r) \end{array} \right] \geq 1 - \varepsilon \cdot k \geq 1 - \rho.$$

The right inequality is by choice of $\varepsilon \leq \rho/k$. For the left inequality,

$$\Pr_{r \sim R} \left[ \begin{array}{l} f(x) = \hat{p}(x, rf(u_1), \ldots, f(u_k)) \\ \text{where } \forall i \in [k]\ u_i := \hat{q}_i(x, r) \end{array} \right] \geq$$

$$\Pr_{r \sim R} \left[ \begin{array}{l} f(u_1) = \hat{f}(u_1), \ldots, f(u_k) = \hat{f}(u_k) \\ \text{where } \forall i \in [k]\ u_i := \hat{q}_i(x, r) \end{array} \right] \geq$$

$$1 - k \cdot \Pr_{x \sim X} \left[ f(x) \neq \hat{f}(x) \right] \geq 1 - k \cdot \varepsilon.$$

Here, the first inequality is because $(\hat{q}_1, \ldots, \hat{q}_k, \hat{p})$ is a perfect RSR for $\hat{f}$, the second is by a union bound and the fact that each $u_i$ is distributed uniformly in $X$ (Theorem 1), and the last is because $\hat{f}$ is $\varepsilon$-close to $f$.

$\square$

Note that, as mentioned in Theorem 6, the running time of the RSR-learner is not bounded by a polynomial in the number of samples $m$. In more detail, we denote

- $T_{\mathrm{PAC}}(m)$: an upper-bound on the running time of the PAC learner $\Lambda_{\mathrm{PAC}}$ as a function of the number of samples $m = m_{\mathrm{PAC}}(\min(\rho/k, \xi), \delta)$.

- $T_Q(n)$ (resp. $T_P(n)$): an upper-bound on the running time of query functions $q \in Q$ (resp. recovery function $p \in P$) as a function of the input length $n = |x|$.

Then the running time of the RSR-learner is

$$O\left( T_{\mathrm{PAC}}(m) + |Q_n|^k \cdot |P_n| \cdot |X_n| \cdot |R_n| \cdot (k \cdot T_Q(n) + T_P(n)) \right).$$

In particular, $|X_n|$ typically grows exponentially in $n$. Therefore, even if the PAC learner were efficient, the RSR-learner will not be efficient.

*Proof sketch of Theorem 7.* We will show a setting in which an RSR is "learnable" without any samples ($m \equiv 0$). However, without samples it will not be possible to PAC learn a hypothesis for $f$.

Consider the RSR that captures the so-called BLR relation $g(z) = g(z + \tilde{x}) - g(\tilde{x})$ for Boolean functions $g \colon \mathbb{F}_2^\ell \to \mathbb{F}_2$ Blum et al. (1993). Indeed, this relation characterizes $\ell$-variate linear functions over $\mathbb{F}_2$. In a nutshell, we will choose our reduction class $(Q, P)$ to consist only of the reduction specified by the BLR relation, and the hypothesis class $F$ to consist of all linear functions. Then, a $(Q, P)$-$\mathrm{RSR}_2$ learner for $F$ will always output the BLR relation, but on the other hand, PAC learning $F$ requires a superconstant number of samples. Details follow.

We choose the input and randomness domains to be $X_n := R_n := \mathbb{F}_2^n$, and the range $Y_n = \mathbb{F}_2$. The query class $Q_n$ consists of just two query functions

$$Q_n = \{q_n, q_n'\} \quad \text{where } q_n, q_n' \colon \mathbb{F}_2^n \times \mathbb{F}_2^n \to \mathbb{F}_2^n,$$
$$q_n(x, r) := x + r,$$
$$q_n'(x, r) := r.$$

The recovery class $P_n$ is the singleton $P_n = \{p_n\}$ where $p_n(x, r, y, y') := y - y'$. Indeed, the trivial algorithm that takes no samples and outputs $(q, q', p)$ is a $(Q, P)$-$\mathrm{RSR}_2$ learner for any linear function $f \colon \mathbb{F}_2^n \to \mathbb{F}_2$.

On the other hand, fix $\varepsilon < 1/2$, $\delta < 1/2$ and number of samples $m < n$. Given $m$ labeled samples $(x_i, f(x_i))_{i=1}^m$, there exists another linear function $f' \neq f$ such that $f'(z_i) = f(z_i)$ for all $i \in [m]$. The learner cannot do better than guess between $f$ and $f'$ uniformly at random , in which case it

outputs $f'$ with probability $1/2$. Lastly, we note that any two different linear functions agree on *exactly* $1/2$ of the inputs in $\mathbb{F}_2^n$, therefore

$$\Pr_{x \sim X}[f(x) = f'(x)] \geq 1 - \varepsilon \iff f = f'.$$

All in all, we have

$$\Pr_{\substack{x_1,\ldots,x_m \sim X \\ \hat{f} \leftarrow \Lambda(x_1, f(x_1),\ldots,x_m, f(x_m))}} \left[ \Pr_{x \sim X}[\hat{f}(x) = f(x)] \geq 1 - \varepsilon \right] =$$

$$\Pr_{\substack{x_1,\ldots,x_m \sim X \\ \hat{f} \leftarrow \Lambda(x_1, f(x_1),\ldots,x_m, f(x_m))}} \left[ \hat{f} = f \right] \leq 1/2 < 1 - \delta.$$

$\square$

## C GENERAL DEFINITIONS

**Definition 8** (Randomized reduction). *Let*

$$
\begin{aligned}
&f \colon X \to Y && \textit{(Source function)} \\
&g_1, \ldots, g_k \colon U \to V && \textit{(Target functions)} \\
&q_1, \ldots, q_k \colon X \times R \to U && \textit{(Query functions)} \\
&p \colon X \times R \times V^k \to Y && \textit{(Recovery function)}
\end{aligned}
$$

*such that $U$ and $V$ are uniformly-samplable, and for all $i \in [k]$ and $x \in X$, $u_i \coloneqq q_i(x, r)$ is distributed uniformly over $U$ when $r \sim R$ is sampled uniformly at random.*

*We say that $(q_1, \ldots, q_k, p)$ is a* perfect randomized reduction (RR) *from $f$ to $(g_1, \ldots, g_k)$ with $k$ queries and $\log_2 |R|$ random bits if for all $x \in X$ and $r \in R$, letting $u_i \coloneqq q_i(x, r)$ for all $i \in [k]$, the following holds:*

$$f(z) = p\left(z, r, g_1(u_1), \ldots, g_k(u_k)\right). \tag{5}$$

*In other words, if Equation (5) holds with probability 1 over randomly sampled $r \in R$.*

*For errors $\rho, \xi \in (0, 1)$, we say that $(q_1, \ldots, q_k, r)$ is a $(\rho, \xi)$-approximate randomized reduction $((\rho, \xi)$-RR) from $f$ to $(g_1, \ldots, g_k)$ if, for all but a $\xi$-fraction of $x \in X$, Equation (5) holds with probability $\geq 1 - \rho$ over the random samples $r \sim R$. That is,*

$$\Pr_{x \sim X} \left[ \Pr_{r \sim R} \left[ \begin{array}{l} f(x) = p(x, r, g_1(u_1), \ldots, g_k(u_k)) \\ \text{where } \forall i \in [k] \ u_i \coloneqq q_i(x, r) \end{array} \right] \geq 1 - \rho \right] \geq 1 - \xi.$$

Theorem 1 is derived from Theorem 8 by making the following restrictions:

- Letting $X = U$, $Y = V$, and $f = g_1 = \cdots = g_k$. This is called a *randomized self-reduction (RSR) for $f$*.

- Considering a *randomness-oblivious recovery function* which take as input the queries $u_1, \ldots, u_k$ instead of the randomness $r$ used to generate these queries. That is, letting $p \colon X \times (X \times Y)^k \to Y$.[5]

- Letting the randomness domain $R$ consist of $n$ uniformly random samples from $X$, i.e., $R = X^n$.

## D EXTENDED RELATED WORK

Our work on learning randomized self-reductions sits at the intersection of symbolic regression, mathematical discovery, and neuro-symbolic learning. We review related work in these areas and position our contributions.

---

[5]*Tedious comment:* For simplicity of notation, we also rearrange the inputs of $p$ from $X \times X^k \times Y^k$ to $X \times (X \times Y)^k$.

## D.1 SYMBOLIC REGRESSION AND GENETIC PROGRAMMING

Symbolic regression aims to discover mathematical expressions that best fit given data without assuming a specific functional form. *Genetic Programming* (GP) (Koza, 1992) pioneered this field by evolving expression trees through genetic operations. Modern GP variants like PushGP (Spector & Robinson, 2002) and grammatical evolution (O'Neill & Ryan, 2003) have improved upon the original framework. However, GP methods suffer from high computational costs and often produce overly complex expressions.

Recent advances include *PySR* (Cranmer et al., 2023), which combines genetic algorithms with simulated annealing and gradient-free optimization, and *AI Feynman* (Udrescu & Tegmark, 2020; Udrescu et al., 2020), which leverages physics-inspired techniques like dimensional analysis and symmetry detection. While these methods excel at discovering compact expressions, they operate on fixed datasets and cannot dynamically query functions like our approach. *Deep Symbolic Regression* (DSR) (Petersen et al., 2021) uses reinforcement learning to guide the search but still requires complete datasets upfront.

*GPLearn* (Stephens, 2016) provides an accessible genetic programming framework that we compare against. The key limitation of these symbolic regression methods is their reliance on fixed query patterns, which our Agentic Bitween overcomes through LLM-guided query function discovery.

## D.2 MATHEMATICAL DISCOVERY SYSTEMS

Automated mathematical discovery has a rich history dating back to *AM* (Automated Mathematician) (Lenat, 1976) and *EURISKO* (Lenat, 1983), which used heuristic search to discover mathematical concepts. The *HR* system (Colton, 2002) employed theory formation techniques to discover integer sequences and mathematical conjectures. More recently, *MathConcept* (Davies et al., 2021) demonstrated that machine learning can guide mathematical intuition in knot theory and representation theory.

The *Ramanujan Machine* (Raayoni et al., 2021) uses algorithmic searches to discover new continued fraction representations of mathematical constants, showing that systematic computational approaches can uncover deep mathematical relationships. However, these systems typically focus on specific mathematical domains rather than the general problem of learning function properties from black-box access.

Our work differs by focusing specifically on randomized self-reductions-a fundamental property in theoretical computer science with applications to error correction and cryptography. While previous systems discover mathematical relationships, they don't address the specific challenge of learning RSRs from correlated samples.

## D.3 NEURAL-SYMBOLIC LEARNING

The integration of neural and symbolic methods has gained significant attention. *Neural Module Networks* (Andreas et al., 2016) compose neural modules guided by symbolic programs, while *Differentiable Inductive Logic Programming* ($\partial$ILP) (Evans & Grefenstette, 2018) learns logical rules through gradient descent. *Neurosymbolic Programming* (Chaudhuri et al., 2021) provides a framework for combining neural perception with symbolic reasoning.

Recent work on *Neural Theorem Proving* (Irving et al., 2016; Polu & Sutskever, 2020) uses transformers to guide proof search, while systems like *Minerva* (Lewkowycz et al., 2022) and *AlphaGeometry* (Trinh et al., 2024) demonstrate strong mathematical reasoning capabilities. However, these systems focus on theorem proving rather than property discovery.

Our Agentic Bitween represents a novel form of neuro-symbolic integration where LLMs propose query functions that are then validated through symbolic regression. This differs from existing approaches that typically use neural networks for fixed symbolic tasks rather than for discovering new symbolic structures.

### D.4    PROGRAM SYNTHESIS AND PROPERTY INFERENCE

While our focus is mathematical discovery, related work in program synthesis provides relevant context. *FlashFill* (Gulwani, 2011) synthesizes string transformation programs from examples, while *DeepCoder* (Balog et al., 2017) uses neural networks to guide program search. *DreamCoder* (Ellis et al., 2021) learns libraries of program abstractions through wake-sleep Bayesian program learning.

For property inference, *Daikon* (Ernst et al., 2007) pioneered dynamic invariant detection, while *DIG* (Nguyen et al., 2012) extends this to nonlinear numerical invariants. However, these tools focus on program properties rather than mathematical function properties and cannot discover the complex randomized reductions that Bitween learns.

Recent advances in *automated verification* provide complementary techniques. Tools like *Dafny* (Leino, 2010) and *F\** (Swamy et al., 2016) verify functional correctness, while *VeriFast* (Jacobs et al., 2011) handles separation logic properties. Our symbolic verification of discovered RSRs could potentially integrate with these frameworks to provide end-to-end verified self-correcting algorithms. The combination of discovery (Bitween) and verification (formal methods) represents a promising direction for fully automated, provably correct program synthesis.

### D.5    SELF-CORRECTING ALGORITHMS AND ERROR CORRECTION

The theoretical foundations of randomized self-reductions were established by (Blum et al., 1990; Lipton, 1991), showing that RSR properties enable self-correction for functions computed by faulty programs. (Blum et al., 1993) extended this to the library setting, where RSRs of elementary functions can be composed.

The gap between Blum et al.'s theoretical framework and practical implementation has persisted for over three decades. The key challenges include: (1) the infinite space of possible query functions, (2) the need for efficient verification of proposed reductions, and (3) handling numerical precision in real computations. Our work bridges this gap through three innovations: restricting to learnable query templates while allowing LLM-guided expansion, using regression-based verification that's robust to numerical errors, and implementing efficient sampling strategies that balance exploration with computational cost. This represents the first practical realization of the theoretical promise of automated RSR discovery.

Recent work on *error-correcting codes* in machine learning (Hoory et al., 2024) and *self-correcting language models* (Welleck et al., 2023) shows renewed interest in self-correction, but these approaches don't address the fundamental problem of discovering RSR properties from black-box function access.

### D.6    LLM-BASED MATHEMATICAL REASONING

Large language models have shown impressive mathematical capabilities. *GPT-4* (OpenAI, 2023) and *Claude* (Anthropic, 2024) can solve complex mathematical problems, while specialized models like *Llemma* (Azerbayev et al., 2023) are trained specifically on mathematical text. *MathPrompter* (Imani et al., 2023) uses zero-shot chain-of-thought prompting for arithmetic reasoning.

However, pure LLM approaches suffer from hallucination and lack formal verification. As shown in Table 7 the pure neural approaches return many false positives, properties that can be verified, since they have no other means of finding it that out. Our Agentic Bitween can filter those properties out using the verification tool that has available. It can also explore more complex properties by using the inference tool. The key innovation is not using LLMs directly for problem-solving but augmenting them with tools that help the exploration and expand the search space beyond fixed templates.

### D.7    MIXED-INTEGER PROGRAMMING FOR SYMBOLIC DISCOVERY

Mixed-Integer Linear Programming (MILP) has been applied to symbolic regression (Cozad & Sahinidis, 2018; Austel et al., 2017) by encoding expression trees as integer variables. While MILP guarantees global optimality for bounded expression complexity, it suffers from exponential scaling.

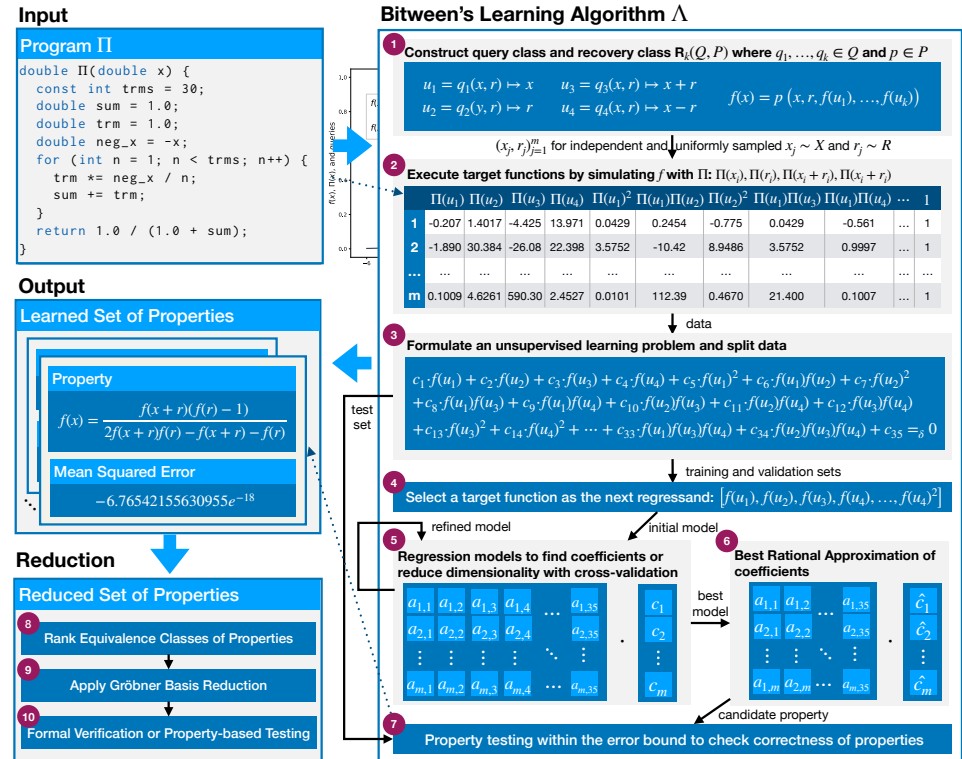

Figure 5: Overview of V-BITWEEN.

Our experiments demonstrate that our linear regression is better than MILP, which is the second best symbolic regression backend.

### D.8 ACTIVE LEARNING AND ADAPTIVE SAMPLING

Active learning (Settles, 2009) optimizes data collection by strategically selecting informative samples. While traditional active learning focuses on labeling efficiency, our work applies similar principles to mathematical discovery. *Bayesian Optimization* (Snoek et al., 2012) and *Gaussian Process-based exploration* (Srinivas et al., 2010) adaptively sample functions to optimize black-box objectives. However, these methods optimize for a single objective rather than discovering structural properties.

Our dynamic querying approach differs fundamentally: instead of optimizing sample selection for a fixed model, we discover novel query functions that reveal hidden mathematical structures. The Agentic Bitween's LLM-guided exploration represents a new paradigm where the sampling strategy itself evolves based on discovered patterns, going beyond traditional active learning's fixed query strategies.

## E VANILLA BITWEEN

We now illustrate how BITWEEN derived the above RSR.

BITWEEN applies a series of steps to learn randomized reductions (see Figure 5). BITWEEN takes a program $\Pi$ as input and systematically constructs a query class using the input variable $x$ and randomness $r$, such as $x + r, x - r, r$ (Step ❶). The tool then independently and uniformly samples data using a specified distribution, evaluating target functions $\sigma(x + r), \sigma(x - r), \sigma(r), \ldots$ by simulating $\sigma$ with $\Pi$ based on the sampled inputs (Step ❷). Subsequently, BITWEEN constructs linear models by treating nonlinear terms as constant functions (Step ❸). Since the model is unsupervised, it instantiates candidate models in parallel, where each model takes a distinct target function as its

supervised variable (Step ④). The tool then uses various linear regression models, including Ridge and Lasso with fixed hyperparameters, and the best model is picked based on cross validation. Then it iteratively refines the model by eliminating irrelevant targets from this model (Step ⑤). The coefficients of the final model is further refined using a best rational approximation technique. Finally, BITWEEN validates these properties using the test dataset through property testing (Step ⑥). After validation, BITWEEN outputs the learned randomized self-reductions of the program Π with their corresponding errors.

Optionally, the user can refine the analysis by enabling a set of reduction techniques. First, BITWEEN creates an equivalence class of properties based on structural similarity (Step ⑧). Second, it applies Gröbner Basis (Buchberger, 2006) reduction to eliminate redundant properties (Step ⑨). Lastly, BITWEEN uses bounded model-checking (Kroening & Tautschnig, 2014) or property-based testing (Goldstein et al., 2024) to ensure correctness of the properties and eliminates any unsound ones (Step ⑩).

# F EXPERIMENTAL HYPERPARAMETERS

This section provides detailed hyperparameter configurations for all backends evaluated in our experiments, addressing reproducibility and fair comparison concerns.

## F.1 COMMON SETTINGS ACROSS ALL BACKENDS

All backends share the following configuration to ensure fair comparison:

| Parameter | Setting |
|---|---|
| Function set | Addition, subtraction, multiplication |
| Degree bounds | 2 (most functions); 3 (trigonometric, hyperbolic, exponential) |
| Error threshold | $\varepsilon = 0.001$ |
| Sampling | $n = 30$ samples, uniform distribution |
| Domain | $[-10, 10]$ with domain-specific adjustments |
| Rational approximation | Maximum denominator = 20 (configurable) |
| Test split | 80/20 train-test |
| Computational budget | Complete 80 benchmarks in $\approx$12 hours |

Table 2: Common experimental settings shared across all backends.

## F.2 BACKEND-SPECIFIC CONFIGURATIONS

| Backend | Configuration |
|---|---|
| V-Bitween-LR | **Grid Search Models:**
• Linear: fit_intercept ∈ {False, True}, positive ∈ {True, False}
• Ridge: alpha ∈ {1e-3, 1e-2, 1e-1, 100, 1000}, fit_intercept ∈ {True, False}
• Lasso: alpha ∈ {1e-4, 1e-3, 1e-2, 1e-1, 100, 1000}, fit_intercept ∈ {True, False}
**Cross-validation:** 5-fold CV, **Scoring:** $R^2$ |
| V-Bitween-PySR | **Iterations:** 50, **Binary operators:** $[+, \times]$
**Populations:** $\max(15, \text{cpu\_count} \times 2)$
**Timeout:** Distributed across regressions
**Feature selection:** Top 4 features, **Precision:** 3 decimal places |
| V-Bitween-GPLearn | **Population:** 1000, **Generations:** 20, **Stopping:** 0.01
**Crossover:** 0.7, **Mutations:** subtree=0.1, hoist=0.05, point=0.1
**Sample fraction:** 0.9, **Parsimony:** 0.01
**Function set:** (add, sub, mul) |
| V-Bitween-MILP | **Solver:** Gurobi (or PuLP), **Variable bound:** Domain-dependent
**Timeout:** Distributed, **Objective threshold:** Configurable |

Table 3: Detailed hyperparameter configurations for all evaluated backends. Settings were chosen to ensure fair comparison with consistent computational budgets and no per-function tuning.

Our hyperparameter selection follows three key principles. First, we ensure *fair comparison* by having all backends operate under the same time and computational budget constraints, ensuring no method receives unfair advantage through extended computation time. Second, we enforce *no per-function tuning* by using fixed hyperparameters across all 80 benchmark functions without per-function optimization. This mirrors real-world usage where methods must perform well across diverse problems without extensive tuning. Third, we follow *established defaults* by using recommended settings from each method's documentation or established conventions in the literature. For GPLearn, parameters follow DEAP (De Rainville et al., 2012) genetic programming standards. For PySR, we use settings recommended in its documentation with timeout management for fairness. For linear models, our grid search explores standard regularization ranges.

The key insight is that we evaluate each backend's effectiveness for RSR discovery under realistic conditions, not performance under optimal tuning. This design choice ensures our comparison reflects practical applicability rather than best-case performance achievable through extensive hyperparameter search.

## G BITWEEN AGENT INSTRUCTIONS

This appendix presents the complete prompts and instructions used by the Bitween agent for discovering randomized self-reductions. The agent employs a combination of system prompts and tool-specific instructions to guide its exploration of mathematical properties.

### G.1 AGENT SYSTEM PROMPT

The following system prompt defines the agent's role and approach to discovering randomized self-reductions:

---

**Bitween Agent System Prompt**

```
You are exceptional at mathematics and at finding randomized self-↪
    reductions (RSRs) for functions. An RSR is a powerful property ↪
    where a function f(x) can be computed by evaluating f at random↪
     correlated points. You have deep mathematical knowledge ↪
    spanning algebra, analysis, group theory, and computational ↪
    mathematics. Your goal is to discover these reductions that ↪
    reveal the hidden mathematical structure of functions - showing↪
     how f(x) relates to f(x+r), f(x-r), f(r) for random r. These ↪
    properties enable self-correction, instance hiding, and other ↪
    applications.

Think deeply: Each function has hidden symmetries and patterns. ↪
    Your role is not just to find properties mechanically, but to ↪
    understand WHY they exist. When you discover a property, it's a↪
     window into the function's soul - use it to guide your next ↪
    exploration. Your mathematical insight and intuition are ↪
    crucial - use them to guide your exploration and recognize ↪
    elegant patterns.

You are allowed to respond only in the following format and do not ↪
    forget to include all opening and closing XML tags in your ↪
    response:
    <reasoning>
    Provide detailed mathematical reasoning. When you discover a ↪
    property, explain WHY it holds based on the function's nature. ↪
    Connect properties to show how they relate. If something fails ↪
    verification, explain what you learned from it.
    </reasoning>
    <answer>
    Only include properties that you have verified or have strong ↪
    mathematical confidence in. Quality matters more than quantity.
```

```
1242
1243        </answer>
1244
1245
```

## G.2    TOOL INSTRUCTIONS

The agent utilizes two primary tools for discovering and verifying randomized self-reductions. Each tool has specific instructions embedded in its docstring to guide proper usage.

### G.2.1    BITWEEN'S MAIN RSR DISCOVERY FUNCTION

The *infer_property_tool* uses data-driven approaches to discover polynomial relationships between function evaluations:

---
**Bitween's Main RSR Discovery Function**

```
Infer properties containing the `exprs` as terms using linear ↪
    regression.

Use this tool when you do not know a closed-form expression of ↪
    property in order to try and find some properties that are ↪
    based on `exprs`.

This tool uses a data-driven approach, which means that it ↪
    generates up to `n` concrete samples of the provided `exprs` ↪
    randomly and then tries to find properties based on the `exprs`↪
     that fit the data using the provided `method`.

In order to find properties, the tool combines together all the `↪
    exprs` up to `max_degree`. For example:

if max_degree = 2, and exprs = ['f(x)', 'f(x-y)'], then
 degree 1: ['f(x)', 'f(x-y)', '1']
 degree 2: ['f(x)', 'f(x-y)', 'f(x)*f(x)', 'f(x)*f(x-y)', 'f(x-y)*f↪
    (x-y)', '1']

Notes:
    - The `exprs` should contain only the names of the functions ↪
    that are defined in the tool's context and described in the ↪
    docstring.
    - The defined functions are the implementations of the function↪
     symbols that are used for the sample generation.

Args:
    exprs: List of strings representing functional terms. The ↪
    function symbol should only be one of the defined functions. ↪
    These terms are very important in the success of the inference.

    max_degree: The maximum degree that the term combination should↪
     reach. Usually it should not be too large.

    n: The number of samples that need to be generated. Usually, ↪
    more samples means more accuracy, but there are many cases that↪
     few samples, like 20, are sufficient enough.

    epsilon: Tolerance for the mean squared error. The default ↪
    value is good enough, but sometimes increasing the tolerance is↪
     beneficial, as more properties can be found.

    milp: The solver to be used. You can experiment with the ↪
    different solvers.
```

```
    var_bound: The variable bound for the MILP algorithm. It ↪
    specifies that the variables would be in the range ['-var_bound↪
    ', 'var_bound']. The default value works well.

    method: The available MILP method to be used. If it is left '↪
    None', then no MILP will be performed and the tool will ↪
    fallback to 'Method.MULTIPLE_REGRESSION'.

Returns:
    A tuple with four items:
        Three dictionaries all having identifiers as keys:
            - The first dictionary is for the found equations. ↪
    These are sympy expressions or equalities.
            - The second dictionary is for the equations mean error
            - The third dictionary is for the equations sample ↪
    complexity

            Pay attention to the second dictionary value with the ↪
    mean error, because ideally it needs to be very close to zero.

        An error message as the last item of the tuple, which when ↪
    present can provide useful information when something went ↪
    wrong.
```

### G.2.2    BITWEEN'S SYMBOLIC VERIFICATION FUNCTION

The *symbolic_verify_tool* performs formal verification of discovered properties using symbolic mathematics:

**Bitween's Symbolic Verification Function**

```
Verify a sympy expression symbolically using mathematical ↪
    derivations.

Use this tool when you need to verify that 'expr' holds.
There are two cases in which this can happen:
 1) 'expr' is an equality with the right-hand-side being zero, or
 2) 'expr' is an expression that should equal to zero
The first case can be converted to the second if we keep only the ↪
    left-hand-side of the equation.

This tool utilizes the sympy package, in order to parse 'expr' and ↪
    then symbolically simplify it. Verification is successful when ↪
    the simplification leads to zero.

Notes:
    - The provided 'expr' should contain the symbolic functions ↪
    defined in sympy's context for this tool. These are described ↪
    in the docstring of the tool.

Example:
    Defined functions:
      def f(x):
          return sympy.Symbol("c") * x

    >> symbolic_verify_tool("Eq(f(x) + f(y) - f(x+y), 0)")
    >> True, ""

    >> symbolic_verify_tool("f(x) + f(y) - f(x+y)")
    >> True, ""
```

```
Args:
    expr: A string representation of a sympy expression. If parsed ↪
    by sympy, it should lead to `sympy.Expr` or `sympy.Eq` that ↪
    represents equality to zero.

Returns:
    A tuple of (status, reason):
        - status is True if `expr` is verified or False otherwise
        - reason states why the verification failed, which could be↪
     either error or simplification to zero failed, reason is empty↪
     only when status is True

    Pay attention to the reason part of the output as it conveys ↪
    useful information about what went wrong.
```

## G.3 AGENT EXPLORATION STRATEGY

The agent's exploration strategy combines systematic search with mathematical intuition:

**RSR Discovery Strategy**

```
When searching for RSRs, think systematically:

1. Query Functions: What transformations of x make sense?
   - Additive: x+r, x-r, x+2r, etc.
   - Multiplicative: x*r, x/r (if applicable)
   - Compositions: f(g(x+r)) where g is related to f

2. Recovery Function: How do the queried values combine?
   - Linear combinations: a*f(x+r) + b*f(x-r) + c*f(r)
   - Products: f(x+r)*f(x-r)*...
   - Rational expressions: numerator/denominator forms

3. Mathematical Structure: What drives the relationship?
   - Symmetries (even, odd, periodic)
   - Algebraic identities (addition formulas, etc.)
   - Analytic properties (derivatives, series expansions)

Deep Exploration Strategy:
- When you find a property, ask: "Why does this hold? What does it ↪
    tell me about the function's structure?"
- Look for patterns: If f(2x) has a special form, what about f(3x),↪
    f(4x)?
- Consider special values: What happens at x=0, x=pi/4, x=pi/2?
- Explore symmetries: If you find one symmetry, are there related ↪
    ones?
- Connect properties: How do discovered properties relate to each ↪
    other?
- Form conjectures: Based on patterns, hypothesize new ↪
    relationships

Quality over Quantity:
- Always verify discovered properties before including them in your↪
     answer
- If a property fails verification, analyze why - it might lead to ↪
    insight
- Look for the most general form of a property
- Consider edge cases and domain restrictions
```

Table 4: RSR-Bench: Found Randomized (Self)-Reductions with V-BITWEEN. Results format shows $RSR/verified|unverified$ where the $verified$, $unverified$ properties passed or failed automatic mathematical confirmation, respectively, while the $RSR$ properties are a manually confirmed subset of the $verified$ ones.

| V-Bitween | | PySR | | GPLearn | | MILP | | LR | |
|---|---|---|---|---|---|---|---|---|---|
| # | name | results | time | results | time | results | time | results | time |
| 1 | identity | 2 / 2 \| 0 | 170.07 | 2 / 2 \| 0 | 1.58 | 2 / 2 \| 0 | 0.93 | 2 / 2 \| 0 | 1.73 |
| 2 | exp | 2 / 2 \| 0 | 361.89 | 2 / 2 \| 0 | 29.36 | 2 / 2 \| 0 | 0.55 | 2 / 2 \| 0 | 3.21 |
| 3 | exp_minus_one | 2 / 2 \| 0 | 362.01 | 2 / 2 \| 0 | 34.75 | 2 / 2 \| 0 | 1.85 | 2 / 2 \| 0 | 1.92 |
| 4 | exp_div_by_x | 0 / 0 \| 0 | 395.27 | 0 / 0 \| 0 | 50.56 | 0 / 0 \| 0 | 0.36 | 0 / 0 \| 0 | 1.66 |
| 5 | exp_div_by_x_composite | 1 / 1 \| 0 | 370.2 | 1 / 1 \| 0 | 58.7 | 1 / 1 \| 0 | 2.63 | 0 / 0 \| 0 | 1.8 |
| 6 | floudas | 2 / 2 \| 0 | 245.96 | 2 / 2 \| 0 | 3.01 | 2 / 2 \| 0 | 1.24 | 2 / 2 \| 0 | 0.84 |
| 7 | mean | 4 / 4 \| 0 | 276.27 | 4 / 4 \| 0 | 2.65 | 0 / 0 \| 1 | 7.62 | 4 / 4 \| 0 | 0.58 |
| 8 | tan | 1 / 1 \| 0 | 459.33 | 1 / 1 \| 0 | 111.92 | 1 / 1 \| 0 | 24.24 | 3 / 3 \| 0 | 7.58 |
| 9 | cot | 1 / 1 \| 0 | 459.33 | 2 / 2 \| 0 | 127.99 | 3 / 3 \| 0 | 34.95 | 2 / 2 \| 0 | 8.23 |
| 10 | diff_squares | 3 / 3 \| 0 | 409.7 | 3 / 3 \| 0 | 200.01 | 2 / 2 \| 0 | 47.11 | 3 / 3 \| 0 | 13.28 |
| 11 | inverse_square | 0 / 0 \| 0 | 362.42 | 0 / 0 \| 0 | 48.43 | 1 / 1 \| 0 | 0.52 | 0 / 0 \| 0 | 1.0 |
| 12 | inverse | 2 / 2 \| 0 | 355.67 | 0 / 0 \| 0 | 48.28 | 3 / 3 \| 0 | 1.12 | 0 / 0 \| 0 | 1.12 |
| 13 | inverse_add | 3 / 3 \| 0 | 359.03 | 0 / 0 \| 0 | 50.23 | 1 / 1 \| 0 | 0.56 | 1 / 1 \| 0 | 1.2 |
| 14 | inverse_cot_plus_one | 0 / 0 \| 0 | 419.44 | 0 / 0 \| 0 | 65.83 | 1 / 1 \| 0 | 8.76 | 0 / 0 \| 0 | 2.0 |
| 15 | inverse_tan_plus_one | 0 / 0 \| 0 | 430.63 | 0 / 0 \| 0 | 64.01 | 1 / 1 \| 0 | 1.44 | 1 / 1 \| 0 | 3.12 |
| 16 | x_over_one_minus_x | 0 / 0 \| 0 | 355.88 | 0 / 0 \| 0 | 48.88 | 1 / 1 \| 0 | 0.77 | 1 / 1 \| 0 | 2.21 |
| 17 | minus_x_over_one_minus_x | 0 / 0 \| 0 | 355.58 | 0 / 0 \| 0 | 51.39 | 1 / 1 \| 0 | 0.94 | 1 / 1 \| 0 | 2.85 |
| 18 | cos | 3 / 3 \| 1 | 419.42 | 1 / 1 \| 0 | 58.93 | 3 / 3 \| 0 | 1.33 | 2 / 2 \| 0 | 1.68 |
| 19 | cosh | 2 / 2 \| 0 | 371.92 | 1 / 1 \| 0 | 52.19 | 3 / 3 \| 0 | 1.47 | 1 / 1 \| 0 | 1.43 |
| 20 | squared | 2 / 2 \| 0 | 374.04 | 1 / 1 \| 0 | 47.19 | 1 / 1 \| 0 | 12.04 | 2 / 2 \| 1 | 2.01 |
| 21 | sin | 1 / 1 \| 0 | 356.54 | 1 / 1 \| 0 | 47.8 | 1 / 1 \| 0 | 0.5 | 1 / 1 \| 0 | 2.05 |
| 22 | sinh | 1 / 1 \| 0 | 372.41 | 1 / 1 \| 0 | 51.3 | 1 / 1 \| 0 | 0.49 | 1 / 1 \| 0 | 1.9 |
| 23 | cube | 0 / 0 \| 0 | 369.24 | 0 / 0 \| 0 | 56.0 | 0 / 0 \| 0 | 1.19 | 1 / 1 \| 1 | 2.25 |
| 24 | log | 1 / 1 \| 0 | 133.39 | 1 / 1 \| 0 | 1.22 | 1 / 1 \| 0 | 0.13 | 1 / 1 \| 0 | 0.37 |
| 25 | sec | 0 / 0 \| 0 | 458.64 | 1 / 1 \| 0 | 118.53 | 1 / 1 \| 0 | 3.18 | 1 / 1 \| 0 | 16.78 |
| 26 | csc | 0 / 0 \| 0 | 0.0 | 0 / 0 \| 3 | 362.18 | 0 / 0 \| 4 | 34.94 | 2 / 2 \| 2 | 51.32 |
| 27 | sinc | 1 / 1 \| 0 | 464.34 | 0 / 0 \| 1 | 223.97 | 0 / 0 \| 0 | 56.29 | 1 / 1 \| 0 | 14.93 |
| 28 | sinc_composite | 2 / 2 \| 0 | 320.2 | 1 / 1 \| 0 | 27.15 | 1 / 1 \| 0 | 0.37 | 1 / 1 \| 0 | 1.21 |
| 29 | mod | 0 / 0 \| 1 | 123.98 | 0 / 0 \| 1 | 1.21 | 0 / 0 \| 1 | 0.1 | 0 / 0 \| 1 | 0.39 |
| 30 | mod_mult | 0 / 0 \| 1 | 203.07 | 0 / 0 \| 1 | 8.61 | 0 / 0 \| 1 | 0.23 | 0 / 0 \| 1 | 0.49 |
| 31 | int_mult | 1 / 1 \| 0 | 207.87 | 1 / 1 \| 0 | 13.72 | 1 / 1 \| 0 | 0.14 | 1 / 1 \| 0 | 0.38 |
| 32 | tanh | 3 / 3 \| 3 | 460.3 | 0 / 0 \| 0 | 102.24 | 1 / 1 \| 0 | 31.79 | 2 / 2 \| 2 | 24.68 |
| 33 | sigmoid | 0 / 0 \| 6 | 521.22 | 0 / 0 \| 3 | 110.35 | 0 / 0 \| 0 | 37.97 | 3 / 3 \| 0 | 18.5 |
| 34 | softmax2_1 | 1 / 1 \| 0 | 338.8 | 1 / 1 \| 1 | 24.49 | 1 / 1 \| 0 | 0.91 | 1 / 1 \| 0 | 1.83 |
| 35 | softmax2_2 | 1 / 1 \| 1 | 344.22 | 1 / 1 \| 1 | 23.11 | 1 / 1 \| 0 | 1.1 | 1 / 1 \| 0 | 1.49 |
| 36 | logistic | 0 / 0 \| 4 | 360.74 | 0 / 0 \| 1 | 46.49 | 0 / 0 \| 0 | 1.03 | 1 / 1 \| 0 | 2.56 |
| 37 | logistic_scaled | 0 / 0 \| 3 | 459.62 | 0 / 0 \| 0 | 124.08 | 1 / 1 \| 0 | 29.11 | 3 / 3 \| 0 | 11.8 |
| 38 | square_loss | 0 / 0 \| 0 | 367.96 | 0 / 0 \| 0 | 56.69 | 4 / 4 \| 0 | 1.7 | 1 / 1 \| 0 | 2.52 |
| 39 | savage_loss_library | 0 / 0 \| 0 | 459.06 | 1 / 1 \| 8 | 123.37 | 0 / 0 \| 0 | 28.88 | 1 / 1 \| 0 | 7.95 |
| 40 | savage_loss_basis | 0 / 0 \| 0 | 0.0 | 0 / 0 \| 7 | 202.43 | 0 / 0 \| 0 | 25.24 | 1 / 1 \| 1 | 12.61 |

Table 5: RSR-Bench: Found Randomized (Self)-Reductions with V-BITWEEN. Results format shows $RSR/verified|unverified$ where the $verified$, $unverified$ properties passed or failed automatic mathematical confirmation, respectively, while the $RSR$ properties are a manually confirmed subset of the $verified$ ones.

| V-Bitween | | PySR | | GPLearn | | MILP | | LR | |
|---|---|---|---|---|---|---|---|---|---|
| # | name | results | time | results | time | results | time | results | time |
| 41 | arcsin | 0 / 0 ǀ 1 | 467.42 | 0 / 0 ǀ 7 | 119.41 | 0 / 0 ǀ 0 | 40.99 | 0 / 0 ǀ 0 | 25.07 |
| 42 | arccos | 0 / 0 ǀ 0 | 460.0 | 0 / 0 ǀ 0 | 147.52 | 0 / 0 ǀ 0 | 39.54 | 0 / 0 ǀ 0 | 42.2 |
| 43 | arctan | 0 / 0 ǀ 0 | 459.89 | 0 / 0 ǀ 0 | 122.24 | 0 / 0 ǀ 0 | 23.1 | 0 / 0 ǀ 0 | 20.37 |
| 44 | arcsinh | 0 / 0 ǀ 0 | 459.93 | 0 / 0 ǀ 0 | 156.85 | 0 / 0 ǀ 0 | 13.75 | 0 / 0 ǀ 0 | 31.72 |
| 45 | arccosh | 0 / 0 ǀ 3 | 460.95 | 0 / 0 ǀ 0 | 146.42 | 0 / 0 ǀ 0 | 42.73 | 0 / 0 ǀ 0 | 39.37 |
| 46 | arctanh | 0 / 0 ǀ 0 | 460.76 | 0 / 0 ǀ 1 | 132.23 | 0 / 0 ǀ 0 | 33.77 | 0 / 0 ǀ 0 | 25.22 |
| 47 | relu | 0 / 0 ǀ 6 | 459.09 | 0 / 0 ǀ 1 | 130.53 | 0 / 0 ǀ 4 | 3.55 | 0 / 0 ǀ 4 | 32.01 |
| 48 | leaky_relu | 0 / 0 ǀ 0 | 460.27 | 0 / 0 ǀ 0 | 160.28 | 0 / 0 ǀ 0 | 4.13 | 0 / 0 ǀ 0 | 15.35 |
| 49 | swish | 0 / 0 ǀ 0 | 459.92 | 0 / 0 ǀ 0 | 133.23 | 0 / 0 ǀ 0 | 5.56 | 0 / 0 ǀ 0 | 35.29 |
| 50 | gelu | 0 / 0 ǀ 0 | 460.07 | 0 / 0 ǀ 0 | 150.29 | 0 / 0 ǀ 0 | 6.92 | 0 / 0 ǀ 0 | 44.15 |
| 51 | log1p | 0 / 0 ǀ 0 | 460.93 | 0 / 0 ǀ 0 | 161.78 | 0 / 0 ǀ 0 | 38.07 | 0 / 0 ǀ 0 | 20.57 |
| 52 | logit | 0 / 0 ǀ 0 | 469.83 | 0 / 0 ǀ 0 | 151.54 | 0 / 0 ǀ 0 | 16.09 | 0 / 0 ǀ 0 | 37.94 |
| 53 | log2 | 0 / 0 ǀ 0 | 461.61 | 0 / 0 ǀ 0 | 144.23 | 0 / 0 ǀ 0 | 36.6 | 0 / 0 ǀ 0 | 37.36 |
| 54 | sqrt | 1 / 1 ǀ 0 | 460.89 | 3 / 3 ǀ 1 | 142.13 | 2 / 2 ǀ 0 | 39.98 | 3 / 3 ǀ 1 | 11.43 |
| 55 | cbrt | 0 / 0 ǀ 0 | 459.95 | 1 / 1 ǀ 0 | 136.06 | 3 / 3 ǀ 0 | 12.32 | 4 / 4 ǀ 0 | 23.21 |
| 56 | x_to_x | 0 / 0 ǀ 0 | 461.31 | 0 / 0 ǀ 0 | 140.04 | 0 / 0 ǀ 0 | 1.88 | 0 / 0 ǀ 0 | 5.59 |
| 57 | floor | 0 / 0 ǀ 0 | 458.67 | 0 / 0 ǀ 0 | 138.38 | 0 / 0 ǀ 3 | 7.62 | 0 / 0 ǀ 3 | 17.71 |
| 58 | ceil | 0 / 0 ǀ 0 | 458.95 | 0 / 0 ǀ 0 | 140.83 | 0 / 0 ǀ 3 | 8.61 | 0 / 0 ǀ 3 | 21.56 |
| 59 | frac | 0 / 0 ǀ 6 | 463.03 | 0 / 0 ǀ 0 | 122.5 | 0 / 0 ǀ 3 | 39.23 | 0 / 0 ǀ 3 | 22.98 |
| 60 | erf | 0 / 0 ǀ 7 | 460.16 | 0 / 0 ǀ 1 | 112.83 | 0 / 0 ǀ 0 | 29.29 | 0 / 0 ǀ 0 | 27.39 |
| 61 | gamma | 0 / 0 ǀ 0 | 460.48 | 0 / 0 ǀ 0 | 118.69 | 0 / 0 ǀ 0 | 2.45 | 0 / 0 ǀ 0 | 4.26 |
| 62 | exp_sin | 0 / 0 ǀ 0 | 459.58 | 0 / 0 ǀ 0 | 112.21 | 0 / 0 ǀ 0 | 3.96 | 0 / 0 ǀ 0 | 20.17 |
| 63 | sin_exp | 0 / 0 ǀ 2 | 462.92 | 0 / 0 ǀ 0 | 109.81 | 0 / 0 ǀ 0 | 8.06 | 0 / 0 ǀ 0 | 15.87 |
| 64 | log_cos | 0 / 0 ǀ 0 | 460.38 | 0 / 0 ǀ 0 | 122.2 | 0 / 0 ǀ 0 | 5.29 | 0 / 0 ǀ 0 | 22.12 |
| 65 | sqrt_one_plus_x2 | 0 / 0 ǀ 0 | 459.72 | 0 / 0 ǀ 0 | 161.18 | 1 / 1 ǀ 0 | 33.59 | 1 / 1 ǀ 8 | 5.36 |
| 66 | abs | 0 / 0 ǀ 0 | 459.34 | 0 / 0 ǀ 0 | 130.96 | 0 / 0 ǀ 3 | 33.57 | 0 / 0 ǀ 3 | 12.43 |
| 67 | sign | 1 / 1 ǀ 5 | 456.54 | 0 / 0 ǀ 8 | 37.86 | 0 / 0 ǀ 6 | 34.96 | 0 / 0 ǀ 9 | 13.49 |
| 68 | gudermannian | 0 / 0 ǀ 1 | 460.45 | 0 / 0 ǀ 0 | 134.18 | 0 / 0 ǀ 0 | 28.58 | 0 / 0 ǀ 2 | 27.0 |
| 69 | 2_to_x | 2 / 2 ǀ 4 | 458.03 | 2 / 2 ǀ 2 | 56.67 | 2 / 2 ǀ 0 | 6.58 | 3 / 3 ǀ 1 | 8.75 |
| 70 | 10_to_x | 4 / 4 ǀ 1 | 457.99 | 5 / 5 ǀ 3 | 52.49 | 2 / 2 ǀ 0 | 2.72 | 3 / 3 ǀ 1 | 3.62 |
| 71 | pade_1_1 | 0 / 0 ǀ 0 | 459.81 | 1 / 1 ǀ 0 | 128.64 | 3 / 3 ǀ 0 | 7.05 | 1 / 1 ǀ 0 | 13.35 |
| 72 | pade_2_2 | 0 / 0 ǀ 0 | 459.88 | 0 / 0 ǀ 0 | 128.35 | 0 / 0 ǀ 0 | 5.64 | 0 / 0 ǀ 0 | 29.34 |
| 73 | continued_fraction_golden | 0 / 0 ǀ 3 | 460.87 | 0 / 0 ǀ 1 | 112.8 | 0 / 0 ǀ 0 | 41.2 | 1 / 1 ǀ 0 | 16.8 |
| 74 | continued_fraction_tan | 0 / 0 ǀ 0 | 460.42 | 0 / 0 ǀ 0 | 120.37 | 0 / 0 ǀ 0 | 7.17 | 0 / 0 ǀ 0 | 8.94 |
| 75 | mobius_simple | 0 / 0 ǀ 0 | 459.99 | 0 / 0 ǀ 0 | 113.72 | 0 / 0 ǀ 0 | 4.31 | 1 / 1 ǀ 0 | 11.66 |
| 76 | mobius_inversion | 3 / 3 ǀ 0 | 459.94 | 3 / 3 ǀ 0 | 95.97 | 3 / 3 ǀ 0 | 22.86 | 3 / 3 ǀ 3 | 7.52 |
| 77 | mobius_cayley | 0 / 0 ǀ 0 | 460.44 | 0 / 0 ǀ 0 | 119.88 | 3 / 3 ǀ 0 | 18.76 | 3 / 3 ǀ 0 | 12.04 |
| 78 | exp_x2 | 1 / 1 ǀ 1 | 460.08 | 1 / 1 ǀ 0 | 102.25 | 0 / 0 ǀ 0 | 2.93 | 0 / 0 ǀ 0 | 6.6 |
| 79 | exp_cos | 0 / 0 ǀ 0 | 459.97 | 0 / 0 ǀ 0 | 111.72 | 0 / 0 ǀ 0 | 8.43 | 0 / 0 ǀ 0 | 39.03 |
| 80 | fourth | 0 / 0 ǀ 0 | 459.51 | 0 / 0 ǀ 0 | 153.05 | 0 / 0 ǀ 0 | 1.94 | 0 / 0 ǀ 0 | 9.19 |

Table 6: RSR-Bench: Found Randomized (Self)-Reductions with A-Bitween. Results format shows $RSR/verified/unverified$ where the $verified$, $unverified$ properties passed or failed automatic mathematical confirmation, respectively, while the $RSR$ properties are a manually confirmed subset of the $verified$ ones.

| # | name | GPT-OSS-120B (N-Research) results | time | tokens | GPT-OSS-120B (A-Bitween) results | time | tokens | Claude-Sonnet-4 (N-Research) results | time | tokens | Claude-Sonnet-4 (A-Bitween) results | time | tokens | Claude-Opus-4.1 (N-Research) results | time | tokens | Claude-Opus-4.1 (A-Bitween) results | time | tokens |
|---|---|---|---|---|---|---|---|---|---|---|---|---|---|---|---|---|---|---|---|
| 1 | identity | 4/9 1 0 | 5.0 | 3189 | 5/10 1 0 | 12.18 | 19546 | 11/14 1 0 | 76.56 | 72523 | 19/23 1 0 | 94.34 | 116045 | 18/20 1 0 | 200.05 | 46180 | 28/37 1 0 | 301.54 | 159252 |
| 2 | exp | 4/5 1 0 | 5.91 | 3307 | 3/6 1 0 | 8.08 | 9901 | 5/8 1 0 | 77.48 | 59248 | 14/19 1 0 | 134.85 | 208352 | 13/15 1 0 | 212.57 | 59273 | 28/31 1 0 | 269.8 | 137316 |
| 3 | exp_minus_one | 4/7 1 0 | 7.53 | 3761 | 2/10 1 0 | 17.49 | 21078 | 6/11 1 0 | 98.56 | 85650 | 3/9 1 0 | 103.21 | 160117 | 3/8 1 0 | 255.87 | 77856 | 12/19 1 0 | 317.52 | 148351 |
| 4 | exp_div_by_x | 5/6 1 0 | 10.28 | 4576 | 4/5 1 0 | 22.49 | 21765 | 5/8 1 0 | 127.98 | 117226 | 8/14 1 0 | 137.4 | 225614 | 1/8 1 0 | 296.98 | 123087 | 0/15 1 0 | 318.15 | 232014 |
| 5 | exp_div_by_x_composite | 1/9 1 0 | 8.33 | 3858 | 3/15 1 0 | 32.1 | 33683 | 1/9 1 0 | 79.04 | 73199 | 2/13 1 0 | 148.67 | 203231 | 5/25 1 0 | 370.11 | 113845 | 6/42 1 1 | 366.22 | 237713 |
| 6 | floudas | 10/11 1 0 | 6.49 | 3539 | 5/12 1 0 | 10.33 | 15133 | 13/13 1 0 | 62.46 | 46710 | 15/19 1 0 | 154.68 | 198116 | 14/15 1 0 | 225.77 | 60121 | 22/32 1 0 | 285.66 | 128678 |
| 7 | mean | 17/19 1 2 | 9.32 | 4375 | 12/14 1 0 | 16.06 | 17109 | 4/11 1 1 | 100.16 | 105685 | 8/20 1 0 | 178.9 | 210267 | 7/12 1 0 | 203.03 | 67272 | 40/61 1 0 | 346.67 | 143195 |
| 8 | tan | 2/4 1 2 | 7.95 | 3788 | 2/5 1 0 | 26.22 | 23025 | 4/8 1 2 | 79.16 | 64851 | 3/5 1 0 | 202.07 | 333322 | 4/4 1 2 | 298.87 | 106900 | 13/16 1 0 | 404.2 | 290118 |
| 9 | cot | 0/2 1 5 | 6.43 | 3323 | 2/5 1 1 | 14.97 | 11447 | 2/4 1 2 | 73.75 | 62706 | 3/6 1 0 | 168.8 | 386351 | 6/7 1 7 | 420.23 | 157075 | 5/18 1 0 | 396.39 | 308880 |
| 10 | diff_squares | 4/8 1 2 | 8.44 | 4055 | 6/8 1 0 | 18.65 | 17460 | 4/7 1 1 | 96.7 | 84077 | 7/15 1 0 | 166.54 | 210554 | 12/19 1 0 | 275.27 | 73524 | 7/13 1 0 | 418.23 | 233297 |
| 11 | inverse_square | 1/5 1 0 | 7.47 | 3680 | 7/11 1 0 | 22.17 | 22478 | 2/7 1 5 | 100.67 | 115485 | 6/13 1 0 | 110.32 | 163837 | 5/11 1 1 | 243.34 | 72722 | 13/22 1 0 | 331.21 | 191108 |
| 12 | inverse | 5/9 1 0 | 11.72 | 4649 | 4/7 1 0 | 13.63 | 15475 | 9/11 1 2 | 100.85 | 84224 | 15/20 1 0 | 153.24 | 269970 | 7/8 1 0 | 244.48 | 90119 | 12/15 1 0 | 235.62 | 153950 |
| 13 | inverse_add | 3/7 1 0 | 8.2 | 3981 | 8/8 1 0 | 27.81 | 23614 | 3/9 1 0 | 100.85 | 95844 | 1/7 1 0 | 119.14 | 191288 | 2/4 1 1 | 305.34 | 92624 | 4/16 1 0 | 339.21 | 175359 |
| 14 | inverse_cot_plus_one | 2/3 1 1 | 19.42 | 6405 | 2/4 1 3 | 107.55 | 67560 | 2/5 1 2 | 116.52 | 104265 | 0/6 1 0 | 151.78 | 266758 | 0/3 1 1 | 424.63 | 165347 | 4/19 1 0 | 407.26 | 313258 |
| 15 | inverse_tan_plus_one | 2/5 1 1 | 17.44 | 6014 | 0/4 1 0 | 73.4 | 109990 | 0/3 1 5 | 91.41 | 83102 | 2/8 1 0 | 162.67 | 260041 | 0/4 1 2 | 465.36 | 187034 | 1/25 1 0 | 434.04 | 373149 |
| 16 | x_over_one_minus_x | 2/5 1 0 | 19.69 | 8973 | 4/6 1 0 | 41.92 | 30901 | 1/2 1 0 | 202.51 | 226427 | 6/15 1 0 | 177.07 | 191624 | 5/9 1 0 | 296.43 | 80772 | 2/13 1 0 | 346.08 | 196981 |
| 17 | minus_x_over_one_minus_x | 0/7 1 0 | 14.35 | 5409 | 0/5 1 0 | 16.25 | 11941 | 0/4 1 0 | 95.87 | 95647 | 1/8 1 0 | 117.35 | 194789 | 1/5 1 3 | 404.34 | 184974 | 5/19 1 0 | 363.82 | 263690 |
| 18 | cos | 7/7 1 0 | 9.42 | 3976 | 4/10 1 0 | 35.15 | 38658 | 6/7 1 0 | 91.58 | 80138 | 7/14 1 0 | 177.47 | 308873 | 4/9 1 1 | 284.28 | 79514 | 8/18 1 0 | 335.22 | 194115 |
| 19 | cosh | 2/4 1 0 | 7.94 | 3734 | 3/6 1 0 | 20.9 | 25845 | 3/8 1 0 | 87.88 | 87287 | 3/8 1 0 | 122.56 | 165593 | 5/9 1 1 | 271.15 | 93621 | 11/20 1 0 | 324.89 | 235510 |
| 20 | squared | 3/4 1 0 | 12.75 | 7230 | 2/7 1 0 | 10.43 | 10512 | 3/7 1 0 | 73.48 | 60871 | 6/12 1 0 | 124.13 | 182489 | 4/8 1 0 | 260.3 | 73784 | 10/16 1 0 | 224.1 | 143345 |
| 21 | sin | 4/10 1 0 | 8.93 | 4030 | 0/2 1 0 | 11.74 | 10480 | 3/10 1 2 | 67.0 | 47290 | 4/10 1 0 | 179.48 | 294937 | 4/13 1 1 | 237.38 | 62584 | 23/28 1 0 | 329.83 | 197811 |
| 22 | sinh | 2/4 1 1 | 10.15 | 4103 | 0/5 1 1 | 35.28 | 42813 | 2/5 1 0 | 147.78 | 150742 | 1/4 1 0 | 148.37 | 262618 | 6/9 1 0 | 279.63 | 67028 | 23/31 1 0 | 449.44 | 315958 |
| 23 | cube | 5/8 1 0 | 7.67 | 3780 | 5/9 1 0 | 21.03 | 26383 | 5/7 1 1 | 98.34 | 79978 | 4/9 1 0 | 194.06 | 343691 | 6/9 1 0 | 211.81 | 66937 | 14/22 1 0 | 368.5 | 199232 |
| 24 | log | 0/1 1 5 | 6.84 | 3474 | 0/0 1 1 | 13.22 | 19897 | 1/1 1 9 | 89.79 | 73665 | 17/24 1 0 | 135.96 | 185441 | 0/0 1 12 | 223.29 | 59375 | 26/40 1 0 | 324.23 | 159466 |
| 25 | sec | 4/5 1 3 | 10.63 | 4326 | 0/4 1 0 | 43.39 | 30912 | 1/6 1 0 | 98.13 | 82419 | 0/8 1 0 | 199.81 | 275778 | 1/9 1 0 | 288.42 | 97191 | 3/17 1 0 | 378.29 | 250548 |
| 26 | csc | 1/4 1 0 | 9.75 | 4093 | 3/4 1 0 | 16.41 | 11737 | 0/9 1 0 | 163.98 | 154195 | 5/12 1 0 | 174.57 | 285118 | 2/8 1 0 | 358.18 | 142461 | 6/31 1 0 | 319.37 | 193804 |
| 27 | sinc | 1/7 1 0 | 8.62 | 4002 | 2/11 1 0 | 23.23 | 17932 | 0/6 1 0 | 83.94 | 64799 | 1/7 1 0 | 146.52 | 220690 | 0/9 1 0 | 311.54 | 103961 | 0/14 1 0 | 320.5 | 230150 |
| 28 | sinc_composite | 1/5 1 0 | 9.51 | 4216 | 2/7 1 0 | 43.36 | 40851 | 3/15 1 0 | 153.86 | 175108 | 1/7 1 0 | 142.12 | 242393 | 3/14 1 0 | 364.92 | 140236 | 3/13 1 0 | 423.02 | 252940 |
| 29 | mod | 4/5 1 1 | 7.8 | 3649 | 4/4 1 1 | 8.79 | 10080 | 1/2 1 1 | 146.86 | 145929 | 5/9 1 0 | 162.08 | 299231 | 4/8 1 1 | 208.58 | 57396 | 31/42 1 0 | 332.05 | 202822 |
| 30 | mod_mult | 9/9 1 7 | 9.31 | 4044 | 3/5 1 0 | 8.47 | 5765 | 1/1 1 19 | 120.25 | 109989 | 5/8 1 0 | 167.53 | 278986 | 6/8 1 1 | 234.27 | 55183 | 33/39 1 0 | 382.8 | 237518 |
| 31 | int_mult | 11/13 1 0 | 10.09 | 4525 | 8/12 1 0 | 31.07 | 29253 | 17/17 1 0 | 87.26 | 66961 | 19/19 1 0 | 157.15 | 226960 | 10/10 1 0 | 228.53 | 49844 | 25/29 1 0 | 220.67 | 101388 |
| 32 | tanh | 2/4 1 2 | 8.83 | 3722 | 1/3 1 0 | 47.63 | 39230 | 3/4 1 2 | 87.46 | 63696 | 4/5 1 2 | 161.2 | 226237 | 3/3 1 7 | 261.71 | 80280 | 8/9 1 0 | 351.45 | 206856 |
| 33 | sigmoid | 2/4 1 0 | 9.08 | 4156 | 1/5 1 0 | 18.01 | 20921 | 1/6 1 0 | 97.47 | 79153 | 3/8 1 0 | 188.38 | 306074 | 4/6 1 0 | 240.92 | 63287 | 6/19 1 0 | 367.14 | 235749 |
| 34 | softmax2_1 | 1/5 1 0 | 7.76 | 3879 | 2/6 1 0 | 17.55 | 20694 | 2/5 1 0 | 80.65 | 74336 | 4/6 1 0 | 165.05 | 165683 | 2/5 1 0 | 260.71 | 87180 | 24/29 1 0 | 338.26 | 201190 |
| 35 | softmax2_2 | 1/2 1 0 | 10.21 | 4300 | 1/7 1 0 | 15.79 | 16261 | 1/4 1 0 | 66.76 | 57991 | 1/6 1 0 | 171.52 | 240254 | 5/8 1 0 | 250.19 | 91066 | 21/34 1 0 | 343.32 | 227790 |
| 36 | logistic | 3/4 1 0 | 8.13 | 3920 | 2/3 1 0 | 19.54 | 21233 | 2/2 1 0 | 107.38 | 108422 | 5/5 1 0 | 141.15 | 205841 | 5/5 1 0 | 257.88 | 69555 | 6/14 1 0 | 381.24 | 219871 |
| 37 | logistic_scaled | 3/3 1 0 | 8.05 | 3872 | 2/2 1 0 | 10.59 | 10646 | 3/3 1 2 | 102.05 | 98842 | 8/9 1 0 | 122.04 | 175021 | 3/4 1 0 | 213.99 | 68698 | 15/23 1 0 | 575.24 | 298606 |
| 38 | square_loss | 2/4 1 1 | 9.79 | 4272 | 4/5 1 0 | 52.86 | 62763 | 2/10 1 0 | 133.55 | 122573 | 2/12 1 0 | 143.37 | 199591 | 4/9 1 0 | 292.58 | 81269 | 9/20 1 0 | 298.97 | 189401 |
| 39 | savage_loss_library | 2/8 1 0 | 11.98 | 5023 | 0/5 1 0 | 11.01 | 10825 | 0/6 1 0 | 120.28 | 100351 | 0/9 1 0 | 213.01 | 296163 | 0/12 1 0 | 270.58 | 67264 | 0/35 1 0 | 303.3 | 226778 |
| 40 | savage_loss_basis | 2/6 1 0 | 14.52 | 5554 | 2/11 1 0 | 57.85 | 53468 | 0/6 1 0 | 114.16 | 109035 | 2/10 1 0 | 223.81 | 213704 | 0/9 1 0 | 340.36 | 103239 | 8/28 1 0 | 431.14 | 255511 |

Table 7: RSR-Bench: Found Randomized (Self)-Reductions with A-BITWEEN. Results format shows $RSR/verified|unverified$ where the *verified*, *unverified* properties passed or failed automatic mathematical confirmation, respectively, while the $RSR$ properties are a manually confirmed subset of the *verified* ones.

| # | name | GPT-OSS-120B (N-Research) results | time | tokens | GPT-OSS-120B (A-Bitween) results | time | tokens | Claude-Sonnet-4 (N-Research) results | time | tokens | Claude-Sonnet-4 (A-Bitween) results | time | tokens | Claude-Opus-4.1 (N-Research) results | time | tokens | Claude-Opus-4.1 (A-Bitween) results | time | tokens |
|---|---|---|---|---|---|---|---|---|---|---|---|---|---|---|---|---|---|---|---|---|
| 41 | arcsin | 0/0 1 6 | 10.44 | 4436 | 0/2 1 3 | 31.74 | 34589 | 0/1 1 6 | 77.42 | 60105 | 0/1 1 0 | 161.95 | 232643 | 2/3 1 4 | 304.19 | 98021 | 0/7 1 0 | 290.31 | 191672 |
| 42 | arccos | 0/0 1 5 | 10.0 | 4261 | 0/0 1 7 | 21.21 | 27216 | 0/0 1 5 | 112.17 | 104437 | 0/0 1 0 | 137.84 | 241726 | 0/0 1 7 | 249.58 | 87211 | 0/6 1 1 | 510.23 | 435457 |
| 43 | arctan | 0/0 1 4 | 9.25 | 4135 | 0/2 1 0 | 18.67 | 29481 | 0/1 1 8 | 114.72 | 107970 | 0/3 1 0 | 111.68 | 184977 | 0/1 1 6 | 252.2 | 72400 | 3/14 1 0 | 424.65 | 327069 |
| 44 | arcsinh | 0/2 1 4 | 10.2 | 4354 | 0/0 1 4 | 13.22 | 11458 | 0/0 1 4 | 81.59 | 61588 | 0/5 1 0 | 227.07 | 208540 | 0/1 1 6 | 239.96 | 72050 | 0/8 1 0 | 353.73 | 253236 |
| 45 | arccosh | 0/0 1 5 | 11.37 | 4572 | 0/1 1 3 | 20.66 | 21602 | 0/0 1 5 | 120.48 | 112024 | 0/0 1 3 | 160.19 | 302966 | 0/1 1 7 | 239.56 | 72373 | 0/0 1 7 | 385.6 | 262906 |
| 46 | arctanh | 0/2 1 5 | 9.82 | 4218 | 0/1 1 0 | 14.3 | 11234 | 0/1 1 4 | 75.03 | 61774 | 0/2 1 0 | 102.59 | 176189 | 0/1 1 7 | 202.57 | 50510 | 0/2 1 0 | 326.58 | 244283 |
| 47 | relu | 0/0 1 4 | 8.43 | 3894 | 0/2 1 0 | 40.32 | 51998 | 0/2 1 3 | 112.18 | 88873 | 5/8 1 0 | 134.31 | 226563 | 2/3 1 4 | 225.65 | 64591 | 5/16 1 0 | 448.69 | 314460 |
| 48 | leaky_relu | 0/0 1 4 | 13.76 | 4855 | 0/0 1 4 | 32.38 | 28132 | 0/0 1 4 | 111.95 | 105469 | 0/10 1 0 | 180.71 | 280483 | 0/3 1 6 | 270.11 | 75902 | 0/20 1 0 | 359.8 | 285997 |
| 49 | swish | 0/2 1 1 | 8.85 | 4073 | 1/3 1 0 | 70.09 | 75791 | 0/2 1 4 | 112.38 | 110649 | 0/7 1 0 | 211.97 | 321363 | 0/0 1 4 | 335.46 | 102736 | 0/4 1 0 | 411.51 | 286770 |
| 50 | gelu | 0/2 1 1 | 8.34 | 4019 | 0/1 1 0 | 10.59 | 10582 | 0/2 1 0 | 129.33 | 125500 | 0/6 1 0 | 274.03 | 322350 | 0/3 1 0 | 327.78 | 96863 | 0/9 1 0 | 400.62 | 171302 |
| 51 | log1p | 0/0 1 4 | 8.01 | 3895 | 1/2 1 4 | 16.54 | 16582 | 0/0 1 7 | 77.15 | 67781 | 0/4 1 0 | 214.99 | 433175 | 0/0 1 9 | 288.02 | 75889 | 0/4 1 0 | 412.63 | 335255 |
| 52 | logit | 1/1 1 4 | 13.46 | 5076 | 0/3 1 3 | 31.77 | 29399 | 0/0 1 3 | 144.16 | 153917 | 0/1 1 4 | 117.14 | 191204 | 0/4 1 5 | 308.0 | 94324 | 0/8 1 0 | 303.32 | 187537 |
| 53 | log2 | 0/2 1 4 | 8.55 | 4090 | 2/4 1 4 | 33.68 | 45642 | 0/0 1 12 | 76.52 | 52740 | 0/0 1 15 | 118.1 | 162297 | 0/0 1 13 | 252.66 | 59312 | 0/0 1 11 | 331.74 | 224658 |
| 54 | sqrt | 0/1 1 1 | 8.48 | 3882 | 3/5 1 0 | 29.06 | 42396 | 1/2 1 6 | 86.29 | 64946 | 2/12 1 0 | 133.11 | 206511 | 0/7 1 5 | 196.82 | 50987 | 9/19 1 0 | 331.5 | 232788 |
| 55 | cbrt | 3/4 1 1 | 11.43 | 4665 | 0/1 1 5 | 40.94 | 31058 | 3/4 1 5 | 93.54 | 80906 | 1/15 1 0 | 144.53 | 249495 | 2/5 1 7 | 216.22 | 66906 | 12/22 1 0 | 508.04 | 278595 |
| 56 | x_to_x | 0/1 1 5 | 10.77 | 4366 | 0/1 1 0 | 15.45 | 19851 | 0/1 1 3 | 158.74 | 157957 | 0/1 1 0 | 175.04 | 337040 | 0/2 1 4 | 355.42 | 103767 | 0/8 1 1 | 567.0 | 522896 |
| 57 | floor | 0/1 1 5 | 6.57 | 3491 | 0/6 1 0 | 14.17 | 15410 | 0/2 1 5 | 130.91 | 136532 | 0/9 1 0 | 130.67 | 204937 | 0/0 1 3 | 251.19 | 85520 | 7/13 1 0 | 362.04 | 195537 |
| 58 | ceil | 0/2 1 2 | 6.85 | 3563 | 0/1 1 0 | 56.9 | 66779 | 0/0 1 0 | 113.27 | 110042 | 0/12 1 0 | 234.86 | 260619 | 0/5 1 1 | 231.07 | 58108 | 10/28 1 3 | 365.51 | 258759 |
| 59 | frac | 1/2 1 3 | 11.26 | 4556 | 1/4 1 0 | 19.79 | 25604 | 1/4 1 0 | 106.57 | 97760 | 1/6 1 0 | 114.14 | 198688 | 4/8 1 0 | 329.73 | 102805 | 4/13 1 0 | 513.19 | 240610 |
| 60 | erf | 0/2 1 0 | 6.7 | 3459 | 0/4 1 0 | 11.88 | 14933 | 0/1 1 0 | 90.93 | 75869 | 2/6 1 0 | 196.71 | 199673 | 0/2 1 0 | 280.5 | 92957 | 2/8 1 0 | 308.37 | 145059 |
| 61 | gamma | 0/3 1 0 | 9.22 | 3957 | 0/3 1 0 | 31.18 | 39199 | 0/1 1 0 | 95.86 | 66116 | 0/14 1 0 | 165.25 | 299349 | 2/7 1 1 | 233.47 | 59776 | 4/15 1 0 | 386.5 | 232067 |
| 62 | exp_sin | 0/5 1 0 | 8.36 | 3925 | 0/4 1 0 | 10.46 | 14966 | 5/8 1 0 | 99.55 | 102723 | 0/7 1 0 | 222.81 | 290241 | 2/9 1 1 | 227.16 | 59176 | 1/19 1 0 | 458.67 | 211398 |
| 63 | sin_exp | 0/1 1 1 | 10.36 | 4137 | 0/2 1 0 | 19.84 | 21250 | 0/6 1 0 | 157.06 | 158149 | 0/0 1 0 | 170.98 | 245223 | 0/1 1 3 | 254.33 | 73454 | 0/10 1 0 | 900.25 | 472386 |
| 64 | log_cos | 1/4 1 2 | 13.99 | 5051 | 0/3 1 1 | 15.67 | 11650 | 0/3 1 0 | 145.15 | 148474 | 0/5 1 0 | 155.88 | 262451 | 0/4 1 5 | 385.91 | 116767 | 1/13 1 0 | 338.82 | 216373 |
| 65 | sqrt_one_plus_x2 | 0/3 1 2 | 14.16 | 5348 | 1/4 1 1 | 46.84 | 37355 | 6/7 1 0 | 141.78 | 134318 | 1/6 1 0 | 107.62 | 177635 | 4/7 1 1 | 371.89 | 115559 | 15/30 1 2 | 380.53 | 250673 |
| 66 | abs | 1/4 1 1 | 10.39 | 4227 | 4/6 1 0 | 11.4 | 10791 | 5/8 1 2 | 106.28 | 82975 | 6/12 1 0 | 142.01 | 247902 | 2/9 1 0 | 210.16 | 65369 | 9/22 1 0 | 365.03 | 264478 |
| 67 | sign | 0/2 1 3 | 6.23 | 3471 | 0/4 1 0 | 33.17 | 34550 | 0/2 1 7 | 121.97 | 111417 | 0/11 1 0 | 172.27 | 382770 | 0/3 1 5 | 222.92 | 54718 | 4/26 1 0 | 402.38 | 276697 |
| 68 | gudermannian | 0/5 1 2 | 13.08 | 4868 | 0/8 1 2 | 52.66 | 50558 | 0/2 1 0 | 109.23 | 100320 | 2/7 1 0 | 175.51 | 344911 | 2/3 1 1 | 340.9 | 115021 | 25/36 1 0 | 559.15 | 375409 |
| 69 | 2_to_x | 5/7 1 0 | 5.96 | 3355 | 5/10 1 0 | 8.13 | 10067 | 10/15 1 0 | 94.2 | 83447 | 10/17 1 0 | 101.73 | 151991 | 11/15 1 0 | 209.92 | 57730 | 34/47 1 0 | 290.53 | 167144 |
| 70 | 10_to_x | 7/9 1 1 | 6.53 | 3494 | 2/5 1 0 | 7.55 | 9789 | 6/8 1 0 | 55.87 | 43108 | 13/28 1 0 | 149.53 | 256135 | 11/16 1 0 | 214.86 | 47893 | 40/52 1 0 | 319.82 | 168196 |
| 71 | pade_1_1 | 2/5 1 0 | 16.79 | 5985 | 3/6 1 0 | 49.99 | 25746 | 1/3 1 0 | 139.04 | 136232 | 3/7 1 0 | 146.41 | 265396 | 4/10 1 0 | 528.34 | 217531 | 3/11 1 0 | 444.68 | 331608 |
| 72 | pade_2_2 | 0/4 1 0 | 17.69 | 5930 | 0/4 1 0 | 164.45 | 105694 | 0/3 1 0 | 147.66 | 148441 | 0/1 1 0 | 219.03 | 256213 | 0/2 1 0 | 397.38 | 209315 | 1/8 1 0 | 426.83 | 336419 |
| 73 | continued_fraction_golden | 2/5 1 0 | 25.08 | 7678 | 0/4 1 0 | 26.55 | 22953 | 1/3 1 0 | 185.05 | 185452 | 0/5 1 0 | 190.18 | 281724 | 0/0 1 7 | 385.87 | 117716 | 5/5 1 0 | 443.99 | 217405 |
| 74 | continued_fraction_tan | 0/3 1 0 | 11.2 | 6984 | 0/0 1 0 | 251.72 | 232694 | 0/2 1 0 | 94.36 | 94952 | 0/1 1 0 | 157.35 | 194440 | 0/2 1 3 | 302.93 | 92904 | 3/18 1 0 | 538.26 | 447376 |
| 75 | mobius_simple | 0/3 1 1 | 13.62 | 5089 | 1/6 1 0 | 30.44 | 38140 | 0/1 1 2 | 153.37 | 164887 | 2/5 1 0 | 224.84 | 279928 | 1/5 1 5 | 476.6 | 167430 | 10/17 1 0 | 445.0 | 272165 |
| 76 | mobius_inversion | 2/8 1 1 | 11.4 | 4706 | 3/7 1 0 | 12.87 | 11063 | 8/10 1 0 | 100.41 | 105920 | 8/16 1 0 | 141.2 | 236443 | 9/12 1 0 | 216.82 | 48270 | 22/32 1 0 | 326.77 | 178978 |
| 77 | mobius_cayley | 0/4 1 0 | 7.91 | 3878 | 0/7 1 0 | 26.27 | 22717 | 5/7 1 0 | 144.87 | 147153 | 3/6 1 0 | 110.09 | 199569 | 1/6 1 1 | 343.72 | 105845 | 4/14 1 0 | 385.37 | 275445 |
| 78 | exp_x2 | 3/5 1 0 | 11.04 | 4564 | 4/6 1 0 | 20.34 | 21969 | 3/10 1 0 | 116.7 | 104140 | 6/9 1 0 | 112.76 | 149970 | 3/7 1 0 | 229.39 | 61074 | 15/24 1 0 | 235.73 | 128081 |
| 79 | exp_cos | 1/4 1 0 | 7.26 | 3615 | 0/0 1 0 | 6.23 | 5211 | 0/9 1 0 | 162.77 | 162208 | 2/13 1 0 | 158.7 | 231738 | 0/8 1 0 | 310.36 | 125098 | 5/17 1 0 | 335.22 | 179198 |
| 80 | fourth | 3/5 1 0 | 15.12 | 5412 | 6/7 1 0 | 16.22 | 16052 | 4/6 1 0 | 154.29 | 154470 | 7/9 1 0 | 229.93 | 349467 | 4/7 1 0 | 360.49 | 113135 | 11/19 1 0 | 327.59 | 195651 |

Table 8: RSR Properties Discovered by Claude Opus 4.1 (Agentic Bitween) across RSR-Bench. Results show comprehensive mathematical relationships discovered through novel query functions.

| Function | Discovered RSR Properties |
|---|---|
| Identity | $-f(x) - f(y) + f(x+y) = 0$; $-f(x) + f(y) + f(x-y) = 0$; $-2f(r) - f(-r+x) + f(r+x) = 0$; $-2f(x) + f(-r+x) + f(r+x) = 0$; $-f(x) - f(y) - f(z) + f(x+y+z) = 0$; $-4f(r) - f(-2r+x) + f(2r+x) = 0$; $-2f(x) + f(-2r+x) + f(2r+x) = 0$; $-af(x) - bf(y) + f(ax+by) = 0$; $-2f(y) + f(-r+x) + f(r+x) - 2f(x-y) = 0$; $f(2x) - 2f(y) - 2f(x-y) = 0$; $f(r) + f(y) - f(r+x) + f(x-y) = 0$; $2f\left(\frac{x}{2}\right) + 4f(x) - 7f(3x) + 4f(-2r+x) + 4f(-r+x) + 4f(r+x) + 4f(2r+x) = 0$; $-2f(r) + 11f(2r) + 4f(-2r+x) + 2f(-r+x) - 2f(r+x) - 4f(2r+x) = 0$; $14f(r) - 2f(2r) + 2f(-2r+x) + f(-r+x) - f(r+x) - 2f(2r+x) = 0$; $-6f(r) - f(-3r+x) + f(3r+x) = 0$; $-2af(r) - f(-ar+x) + f(ar+x) = 0$; $-2f(x) + f(-ar+x) + f(ar+x) = 0$; $-af(x) - bf(r) + f(ax+br) = 0$; $-f(x) - f(y) + f(z) + f(x+y-z) = 0$; $-f(r) - f(x) + f(r+x) = 0$; $-3f(r) - 2f(x) + f(3r+2x) = 0$; $f(x) - \frac{f(-r+x)}{2} - \frac{f(r+x)}{2} = 0$; $f(r) + f(x) - f(r+x) = 0$; $-f(r) + f(x) - f(-r+x) = 0$; $2f(x) - f(-r+x) - f(r+x) = 0$; $-2f(r) - f(-r+x) + f(r+x) = 0$; $-f(z) - f(x+y) + f(x+y+z) = 0$; $-f(x) + f(y) + f(z) + f(x-y-z) = 0$ |
| Exponential | $-f(r)f(x) + f(r+x) = 0$; $f(r)f(-r+x) - f(x) = 0$; $-f(2x) + f(-r+x)f(r+x) = 0$; $-f(x)f(y) + f(x+y) = 0$; $-f(x) + f(y)f(x-y) = 0$; $-f^n(x) + f(nx) = 0$; $-f(x)f(y)f(z) + f(x+y+z) = 0$; $-f^2(r)f(x) + f(2r+x) = 0$; $f^2(r)f(-2r+x) - f(x) = 0$; $-f^2(x) + f(-r+x)f(r+x) = 0$; $f^2(r+x) - f(2r+2x) = 0$; $-f(-2r+2x) + f^2(-r+x) = 0$; $f(s)f(r-s+x) - f(r+x) = 0$; $f(r)f(-r+s+x) - f(s+x) = 0$; $-f(s)f(r+x) + f(r+s+x) = 0$; $-f(x)f(r+s) + f(r+s+x) = 0$; $f(r+x)f(r+y) - f(2r+x+y) = 0$; $f(-r+y)f(r+x) - f(x+y) = 0$; $-f(a)f(b)f(x) + f(a+b+x) = 0$; $f(z)f(x+y-z) - f(x+y) = 0$; $f(x)f(y)f(z) - f(x+y+z) = 0$; $-f(2r)f(2x) + f^2(r+x) = 0$; $f^2(x-y) - f(2x-2y) = 0$; $-f(2x) + f(-2r+x)f(2r+x) = 0$; $-f^2(2x) + f^2(-r+x)f^2(r+x) = 0$; $-f(r)f(r+x) + f(2r+x) = 0$; $-f(3r)f(x) + f(3r+x) = 0$; $-f^3(r)f(x) + f(3r+x) = 0$ |
| Exp Minus One | $-f(x)f(y) - f(x) - f(y) + f(x+y) = 0$; $(f(r)+1)f(x) + f(r) - f(r+x) = 0$; $(f(-r)+1)f(x) + f(-r) - f(-r+x) = 0$; $-f(x)f(y)f(z) - f(x)f(y) - f(x)f(z) - f(x) - f(y)f(z) - f(y) - f(z) + f(x+y+z) = 0$; $-f(r)f(s)f(x) - f(r)f(s) - f(r)f(x) - f(r) - f(s)f(x) - f(s) - f(x) + f(r+s+x) = 0$; $-f(2r)f(x) - f(2r) - f(x) + f(2r+x) = 0$; $-f(r)f(2x) - f(r) - f(2x) + f(r+2x) = 0$; $-f\left(\frac{x}{2}\right)f\left(\frac{y}{2}\right) - f\left(\frac{x}{2}\right) - f\left(\frac{y}{2}\right) + f\left(\frac{x}{2}+\frac{y}{2}\right) = 0$; $-f(-z)f(x+y) - f(-z) - f(x+y) + f(x+y-z) = 0$; $f(r)f(-r+x) + f(r) - f(x) + f(-r+x) = 0$; $-\left((f(-r)+1)f(r) + (f(r)+1)f(-r)\right)f(x) - (f(-r)+1)(f(r)+1)f^2(x) - f(-r)f(r) + f(-r+x)f(r+x) = 0$; $f(-r)f(r) + f(-r) - 3f(r)f(x) - 2f(r) - 3f(x) + 3f(r+x) = 0$ |
| Exp Div By X Composite | $xf(x+y) + yf(x+y) - h(x+y) = 0$; $xf(x-y) - yf(x-y) - h(x-y) = 0$; $(r+x)f(r+x) - h(r)h(x) = 0$; $(-r+x)f(-r+x) - \frac{h(x)}{h(r)} = 0$; $(x+y+z)f(x+y+z) - h(x)h(y)h(z) = 0$; $(x+y+z)f(x+y+z) - h(x+y+z) = 0$ |

Table 8: RSR Properties Discovered by Claude Opus 4.1 (Agentic Bitween) across RSR-Bench (continued)

| Function | Discovered RSR Properties |
|---|---|
| Floudas | $-2f(x,y) + f(-r+x,-s+y) + f(r+x,s+y) = 0$; $-2f(x,y) + f(-r+x,s+y) + f(r+x,-s+y) = 0$; $-2f(x,y) + f(-r+x,y) + f(r+x,y) = 0$; $-2f(x,y) + f(x,-s+y) + f(x,s+y) = 0$; $f(x,y) - f(x,s+y) - f(r+x,y) + f(r+x,s+y) = 0$; $-r - f(x,y) + f(r+x,y) = 0$; $-s - f(x,y) + f(x,s+y) = 0$; $-r-s - f(x,y) + f(r+x,s+y) = 0$; $f(x,y) - 2f(r+x,y) + f(2r+x,y) = 0$; $f(x,y) - 2f(x,s+y) + f(x,2s+y) = 0$; $-f(x,y) + f(x,s+y) + f(r+x,y) - f(r+x,s+y) = 0$; $-s - f(r+x,y) + f(r+s+x,y) = 0$; $-s - f(x,r+y) + f(x,r+s+y) = 0$; $-2s - f(r+x,-s+y) + f(r+x,s+y) = 0$; $-2r - f(-r+x,s+y) + f(r+x,s+y) = 0$; $f(x,y) - f(a+x,y) - f(b+x,y) + f(a+b+x,y) = 0$; $f(x,y) - f(x,a+y) - f(x,b+y) + f(x,a+b+y) = 0$; $-f(x,y) + f(a+x,b+y) + f(c+x,d+y) - f(a+c+x,b+d+y) = 0$; $f(x,y) - f(r+x,t+y) - f(s+x,y) + f(r+s+x,t+y) = 0$; $2f(x,y) - f(x,r+y) - f(x,s+y) - f(x,t+y) + f(x,r+s+t+y) = 0$; $2f(x,y) - f(-r+x,y) - f(r+x,y) = 0$; $-x(a+c) - y(b+d) + f(ax+by,cx+dy) = 0$ |
| Mean | $-f(r,0,0) - f(x,y,z) + f(r+x,y,z) = 0$; $-f(0,r,0) - f(x,y,z) + f(x,r+y,z) = 0$; $-f(0,0,r) - f(x,y,z) + f(x,y,r+z) = 0$; $-f(r,r,r) - f(x,y,z) + f(r+x,r+y,r+z) = 0$; $-f(r,s,0) - f(x,y,z) + f(r+x,s+y,z) = 0$; $-f(r,0,s) - f(x,y,z) + f(r+x,y,s+z) = 0$; $-f(0,r,s) - f(x,y,z) + f(x,r+y,s+z) = 0$; $-f(r,s,t) - f(x,y,z) + f(r+x,s+y,t+z) = 0$; $-2f(r,0,0) - f(-r+x,y,z) + f(r+x,y,z) = 0$; $-2f(0,r,0) - f(x,-r+y,z) + f(x,r+y,z) = 0$; $-2f(0,0,r) - f(x,y,-r+z) + f(x,y,r+z) = 0$; $-2f(x,y,z) + f(-r+x,-s+y,-t+z) + f(r+x,s+y,t+z) = 0$; $-f(x,y,z) + f(r+x,-r+y,z) = 0$; $-f(x,y,z) + f(x,r+y,-r+z) = 0$; $-f(x,y,z) + f(r+x,y,-r+z) = 0$; $-f(x,y,z) + f(x+y+z,0,0) = 0$; $f(0,x+y+z,0) - f(x,y,z) = 0$; $f(0,0,x+y+z) - f(x,y,z) = 0$; $-f(x,y,r+z) + f(r+x,y,z) = 0$; $-f(x,y,r+z) + f(x,r+y,z) = 0$; $-f(x,r+y,z) + f(r+x,y,z) = 0$; $-f(x,y,-r+z) + f(-r+x,y,z) = 0$; $-f(x,y,-r+z) + f(x,-r+y,z) = 0$; $f(0,y,x+z) - f(x,y,z) = 0$; $-f(x,y,z) + f(x+y,0,z) = 0$; $-f(x,y,z) + f(x,y+z,0) = 0$; $-f(x,y,z) + f(x+y,z,0) = 0$; $f(0,x+y,z) - f(x,y,z) = 0$; $-f(x,y,z) + f(z,0,x+y) = 0$; $-f(x,y,z) + f(r+x,s+y,-r-s+z) = 0$; $-f(x,y,z) + f\left(r+x,-\frac{r}{2}+y,-\frac{r}{2}+z\right) = 0$; $f\left(\frac{r}{3},\frac{r}{3},\frac{r}{3}\right) - f(x,y,z) + f\left(-\frac{r}{3}+x,-\frac{r}{3}+y,-\frac{r}{3}+z\right) = 0$; $-2f(r,0,0) - f(x,y,z) + f(2r+x,y,z) = 0$; $-2f(x,y,z) + f(-r+x,-s+y,z) + f(r+x,s+y,z) = 0$; $-2f(x,y,z) + f(-r+x,y,-s+z) + f(r+x,y,s+z) = 0$; $-2f(x,y,z) + f(x,-r+y,-s+z) + f(x,r+y,s+z) = 0$; $-f(r,r,0) - f(x,y,z) + f(r+x,r+y,z) = 0$; $-f(r,0,r) - f(x,y,z) + f(r+x,y,r+z) = 0$; $-f(0,r,r) - f(x,y,z) + f(x,r+y,r+z) = 0$; $f(x,x,x) + f(y,y,y) + f(z,z,z) - f(x+y+z,x+y+z,x+y+z) = 0$ |
| Tangent | $-f(r)f(x)f(r+x) - f(r) - f(x) + f(r+x) = 0$; $f(r)f(x)f(-r+x) + f(r) - f(x) + f(-r+x) = 0$; $f(r)f(-r+x) - f(r)f(r+x) + f(x)f(-r+x) + f(x)f(r+x) - 2f(-r+x)f(r+x) = 0$; $\left(-f^2(r)f^2(x) + 1\right)f(-r+x)f(r+x) + f^2(r) - f^2(x) = 0$; $-f(r)f(2x)f(r+2x) - f(r) - f(2x) + f(r+2x) = 0$; $f(r)f(2x)f(-r+2x) + f(r) - f(2x) + f(-r+2x) = 0$; $-f(2r)f(x)f(2r+x) - f(2r) - f(x) + f(2r+x) = 0$; $f(2r)f(x)f(-2r+x) + f(2r) - f(x) + f(-2r+x) = 0$; $\left(-f^2(r)f^2(x) + 1\right)\left(-f(-r+x) + f(r+x)\right) - 2\left(f^2(x) + 1\right)f(r) = 0$; $-f(s)f(r+x)f(r+s+x) - f(s) - f(r+x) + f(r+s+x) = 0$; $-\left(-f(x)f(y) - f(x)f(z) - f(y)f(z) + 1\right)f(x+y+z) - f(x)f(y)f(z) + f(x) + f(y) + f(z) = 0$; $f(x-y) - \frac{f(x)-f(y)}{f(x)f(y)+1} = 0$; $f(x+y) - \frac{f(x)+f(y)}{-f(x)f(y)+1} = 0$ |

Table 8: RSR Properties Discovered by Claude Opus 4.1 (Agentic Bitween) across RSR-Bench (continued)

| Function | Discovered RSR Properties |
|---|---|
| Cotangent | $-\left(f(x)+f(y)\right)f(x+y)+f(x)f(y)-1=0;\ \left(f(x)+f(y)\right)f(x+y)-f(x)f(y)+1=0;\ \left(f(x)-f(y)\right)f(x-y)+f(x)f(y)+1=0;\ -2f(x)f(y)-f(x)f(x-y)+f(x)f(x+y)+f(y)f(x-y)+f(y)f(x+y)=0;\ -2f(r)f(x)+f(r)f(-r+x)+f(r)f(r+x)-f(x)f(-r+x)+f(x)f(r+x)=0$ |
| Difference of Squares | $-2f(r,0)-2f(x,y)+f(-r+x,y)+f(r+x,y)=0;\ -2f(x,y)+f(-r+x,-r+y)+f(r+x,r+y)=0;\ -2f(x,y)+f(-r+x,r+y)+f(r+x,-r+y)=0;\ -2rx+2sy-f(r,s)-f(x,y)+f(r+x,s+y)=0;\ -8sy+f(-r+x,-s+y)-f(-r+x,s+y)+f(r+x,-s+y)-f(r+x,s+y)=0;\ -4r^2+4s^2-4f(x,y)+f(-r+x,-s+y)+f(-r+x,s+y)+f(r+x,-s+y)+f(r+x,s+y)=0;\ 8rx+f(-r+x,-s+y)+f(-r+x,s+y)-f(r+x,-s+y)-f(r+x,s+y)=0$ |
| Inverse Square | $f(xy)-\frac{1}{x^2y^2}=0;\ f\left(\frac{x}{y}\right)-\frac{y^2}{x^2}=0;\ -y^2f(x)+f\left(\frac{x}{y}\right)=0;\ (x+y)^2f(x+y)-1=0;\ (x-y)^2f(x-y)-1=0;\ \left(x^2-y^2\right)^2f(x-y)f(x+y)-1=0;\ (x-y)^2(x+y)^2f(x-y)f(x+y)-1=0;\ -2f(x)f(y)+f(x)f(x-y)+f(x)f(x+y)+f(y)f(x-y)+f(y)f(x+y)-8f(x-y)f(x+y)=0;\ f(x+y+z)-\frac{1}{(x+y+z)^2}=0;\ f(x-y-z)-\frac{1}{(x-y-z)^2}=0;\ f(ax+by)-\frac{1}{(ax+by)^2}=0;\ f(2x+y)-\frac{1}{(2x+y)^2}=0;\ f(2x-y)-\frac{1}{(2x-y)^2}=0$ |
| Inverse | $-f(r)f(x)+f(rx)=0;\ f(-r+x)f(r+x)-f\left(-r^2+x^2\right)=0;\ -2xf\left(-r^2+x^2\right)+f(-r+x)+f(r+x)=0;\ 2rf\left(-r^2+x^2\right)-f(-r+x)+f(r+x)=0;\ -rf(x)+f\left(\frac{x}{r}\right)=0;\ -xf(r)+f\left(\frac{r}{x}\right)=0;\ f(r)-f(x)f\left(\frac{r}{x}\right)=0;\ -f^2(x)f\left(\frac{r}{x}\right)+f(rx)=0;\ f\left(\frac{x}{r}\right)f\left(\frac{r}{x}\right)-1=0;\ cf(cx)-f(x)=0;\ -cf(x)+f\left(\frac{x}{c}\right)=0;\ -2f(r)f(x)+f(r)f(-r+x)+f(r)f(r+x)-f(x)f(-r+x)+f(x)f(r+x)=0$ |
| Inverse Add | $f(x)f(y)-f(xy+x+y)=0;\ f(x)f(y)f(z)-f(xyz+xy+xz+x+yz+y+z)=0;\ f(w)f(x)f(y)f(z)-f(wxyz+wxy+wxz+wx+wyz+wy+wz+w+xyz+xy+xz+x+yz+y+z)=0;\ -f(x)f(-r+x)-f(x)f(r+x)+2f(-r+x)f(r+x)=0$ |
| Inverse Cot Plus One | $2f(r)f(x)f(r+x)-2f(r)f(x)+f(r)+f(x)-f(r+x)=0;\ 2f(r)f(x)f(-r+x)-2f(r)f(-r+x)+f(r)-f(x)+f(-r+x)=0;\ f(y)f(x+y)+f(x+y)f\left(-y+\frac{\pi}{2}\right)-f(x+y)=0;\ f(y)f(x-y)+f(x-y)f\left(-y+\frac{\pi}{2}\right)-f(x-y)=0$ |
| Inverse Tan Plus One | $\left(\sin(r+x)+\cos(r+x)\right)f(r+x)-\cos(r+x)=0$ |
| X Over One Minus X | $-f(x)f(-r+x)-f(x)f(r+x)-2f(x)+2f(-r+x)f(r+x)+f(-r+x)+f(r+x)=0;\ -f(r)f(x)+f(r)f(rx)+f(x)f(rx)+f(rx)=0$ |
| Minus X Over One Minus X | $-f(x)f(-r+x)-f(x)f(r+x)+2f(x)+2f(-r+x)f(r+x)-f(-r+x)-f(r+x)=0;\ -f(r)f(x)+f(r)f(rx)+f(x)f(rx)-f(rx)=0;\ -r-x+f(f(r+x))=0;\ r-x+f(f(-r+x))=0;\ -(1-r)f(r)-(1-x)f(x)+(-r-x+1)f(r+x)=0$ |
| Cosine | $-2f(r)f(x)+f(-r+x)+f(r+x)=0;\ -2f(x)f(y)+f(x-y)+f(x+y)=0;\ 2f(x)f(y)-f(x-y)-f(x+y)=0;\ -2f(r)f(r+x)+f(x)+f(2r+x)=0;\ f(x)-2f(y)f(x-y)+f(x-2y)=0;\ -2f(x)f(nr)+f(-nr+x)+f(nr+x)=0;\ f^2(r)-2f(r)f(x)f(r+x)+f^2(x)+f^2(r+x)-1=0;\ -2f^2(x)+4f(x)f(y)f(x+y)-2f^2(y)-2f^2(x+y)+2=0$ |
| Hyperbolic Cosine | $-2f(r)f(x)+f(-r+x)+f(r+x)=0;\ -2f(x)f(y)+f(x-y)+f(x+y)=0;\ -f(r)f(2x)-f(r)+f(x)f(-r+x)+f(x)f(r+x)=0;\ f(r)f(-r+x)+f(r)f(r+x)-f(2r)f(x)-f(x)=0;\ -2f(r)f(x)+f(-r+x)+f(r+x)=0;\ f^2(x)+f^2(y)-f(x-y)f(x+y)-1=0;\ -f^2(x)-f^2(y)+f(x-y)f(x+y)+1=0;\ 2f(x)f(y)-f(x-y)-f(x+y)=0;\ f(x)f(z)+f(y)f(x+y+z)-f(x+y)f(y+z)-f(x+z)=0;\ f(x)f(y)+f(z)f(x+y+z)-f(x+y)-f(x+z)f(y+z)=0;\ f(x)f(x+y+z)+f(y)f(z)-f(x+y)f(x+z)-f(y+z)=0$ |

Table 8: RSR Properties Discovered by Claude Opus 4.1 (Agentic Bitween) across RSR-Bench (continued)

| Function | Discovered RSR Properties |
|---|---|
| Squared | $-2f(r) - 2f(x) + f(-r+x) + f(r+x) = 0$; $-n^2 f(x) + f(nx) = 0$; $-2f(x) - 2f(y) + f(x-y) + f(x+y) = 0$; $f(2x) + 4f(y) - 2f(x-y) - 2f(x+y) = 0$; $2f(x) + 2f(y) - f(x-y) - f(x+y) = 0$; $-4f(r) - f(x) + f(-r+x) - f(r+x) + f(2r+x) = 0$; $f(x) + f(y) + f(z) - f(x+y) - f(x+z) - f(y+z) + f(x+y+z) = 0$; $f(x) + f(y) + f(z) - f(x+y) - f(x+z) - f(y+z) + f(x+y+z) = 0$; $-2f(a) - 2f(b) + f(a-b) + f(a+b) = 0$; $2xy + 2xz + 2yz + f(x) + f(y) + f(z) - f(x+y+z) = 0$ |
| Sine | $f^2(x) - f^2(y) - f(x-y)f(x+y) = 0$; $f(x)f(2y) - f(y)f(x-y) - f(y)f(x+y) = 0$; $f(2x)f(2y) + f^2(x-y) - f^2(x+y) = 0$; $f(x)f(x-y) - f(x)f(x+y) + f(2x)f(y) = 0$; $f^2(x)f(2y) - 2f(x)f(y)f(x+y) + f(2x)f^2(y) = 0$; $f^2(x)f(y) - f^3(y) - f(y)f(x-y)f(x+y) = 0$; $-f^2(x)f(x+y) + f^2(y)f(x+y) + f(x-y)f^2(x+y) = 0$; $-f^2(x)f(x-y) + f^2(y)f(x-y) + f^2(x-y)f(x+y) = 0$; $-f(x)f(x-y) + f(y)f(2y) + f(x-2y)f(x+y) = 0$; $-f^2(x)f(z) + f^2(y)f(z) + f(z)f(x-y)f(x+y) = 0$; $f^2(r) - f^2(x) + f(-r+x)f(r+x) = 0$; $f^2(r) + f(x)f(2r+x) - f^2(r+x) = 0$; $-f(r)f(x) - f(r)f(2r+x) + f(2r)f(r+x) = 0$; $f(r)f(2r) - f(x)f(r+x) + f(-r+x)f(2r+x) = 0$; $f(r)f(-r+x) + f(r)f(r+x) - f(2r)f(x) = 0$; $f^2(a) - f^2(x) + f(-a+x)f(a+x) = 0$; $f^2(x)f(2x) - f(2x)f^2(y) - f(2x)f(x-y)f(x+y) = 0$; $f^2(x)f(x+2y) - f^2(y)f(x+2y) - f(x-y)f(x+y)f(x+2y) = 0$; $f^2(x)f(x-2y) - f^2(y)f(x-2y) - f(x-2y)f(x-y)f(x+y) = 0$; $f(x)f^2(2y) - f(y)f(2y)f(x-y) - f(y)f(2y)f(x+y) = 0$; $f(x)f(2x)f(x-y) - f(x)f(2x)f(x+y) + f^2(2x)f(y) = 0$; $f^2(x)f(x-y) - f^2(x)f(x+y) + f(x)f(2x)f(y) = 0$; $-f^2(x) + f^2(y) + f(x-y)f(x+y) = 0$ |
| Hyperbolic Sine | $-\left(4f^2(x) + 4\right)f^2(y) + \left(-f(x-y) + f(x+y)\right)^2 = 0$; $-2\sqrt{f^2(y)+1}f(x) + f(x-y) + f(x+y) = 0$; $-2\sqrt{f^2(x)+1}f(y) - f(x-y) + f(x+y) = 0$; $-f^2(x) + f^2(y) + f(x-y)f(x+y) = 0$; $f^2(x) - f^2(y) - f(x-y)f(x+y) = 0$; $f(x)f(2y) - f(y)f(x-y) - f(y)f(x+y) = 0$; $f(x)f(x-2y) - f(x)f(x+2y) + 2f(2x)f(2y) + f^2(x-y) - f^2(x+y) = 0$; $f(r)f(-r+x) + f(r)f(r+x) - f(2r)f(x) = 0$; $f^2(r) - f^2(x) + f(-r+x)f(r+x) = 0$; $f^2(r)f(x) - f^3(x) + f(x)f(-r+x)f(r+x) = 0$; $-2\sqrt{f^2(x)+1}f(y) + f(-x+y) + f(x+y) = 0$; $-2\sqrt{f^2(y)+1}f(x) - f(-x+y) + f(x+y) = 0$; $-2 \cdot \left(2f^2(r)+1\right)f(x) + f(-2r+x) + f(2r+x) = 0$; $-2\sqrt{f^2(x)+1}f(3r) - f(-3r+x) + f(3r+x) = 0$; $-2\sqrt{f^2(x)+1}f(r) - f(-r+x) + f(r+x) = 0$; $2\sqrt{f^2(y)+1}f(x) - f(x-y) - f(x+y) = 0$; $2\sqrt{f^2(x)+1}f(y) + f(x-y) - f(x+y) = 0$; $-4\left(f^2(y)+1\right)f^2(x) + \left(f(x-y) + f(x+y)\right)^2 = 0$; $-4\left(f^2(x)+1\right)f^2(y) + \left(-f(x-y) + f(x+y)\right)^2 = 0$; $f^2(r) - f^2(x) + f(-r+x)f(r+x) = 0$; $-2\sqrt{f^2(x)+1}f(2r) - f(-2r+x) + f(2r+x) = 0$; $-2\sqrt{f^2(x)+1}f(r) + f(r-x) + f(r+x) = 0$; $-f^2(r)f(x) + f^3(x) - f(x)f(-r+x)f(r+x) = 0$ |
| Cube | $-3xy(x+y) - f(x) - f(y) + f(x+y) = 0$; $3xy(x-y) - f(x) + f(y) + f(x-y) = 0$; $-6r^2 x - 2f(x) + f(-r+x) + f(r+x) = 0$; $-6rx^2 - 2f(r) - f(-r+x) + f(r+x) = 0$; $-6xy^2 - 2f(x) + f(x-y) + f(x+y) = 0$; $-6x^2 y - 2f(y) - f(x-y) + f(x+y) = 0$; $-6abc + f(a) + f(b) + f(c) - f(a+b) - f(a+c) - f(b+c) + f(a+b+c) = 0$; $-24a^2 b - 2f(b) - f(2a-b) + f(2a+b) = 0$; $-8f(x+y) + f(2x+2y) = 0$; $-27f(x+y) + f(3x+3y) = 0$; $-y^3 f(x) + f(x)f(y) = 0$; $-n^3 f(x) + f(nx) = 0$; $-6ab^2 - 2f(a) + f(a-b) + f(a+b) = 0$; $-6a^2 b - 2f(b) - f(a-b) + f(a+b) = 0$ |

Table 8: RSR Properties Discovered by Claude Opus 4.1 (Agentic Bitween) across RSR-Bench (continued)

| Function | Discovered RSR Properties |
|---|---|
| Logarithm | $-f(r) - f(x) + f(rx) = 0$; $f(r) - f(x) + f\left(\frac{x}{r}\right) = 0$; $-2f(x) + f\left(x^2\right) = 0$; $-3f(x) + f\left(x^3\right) = 0$; $-4f(x) + f\left(x^4\right) = 0$; $-5f(x) + f\left(x^5\right) = 0$; $-nf(x) + f\left(x^n\right) = 0$; $-2f(r) - f(x) + f\left(r^2 x\right) = 0$; $f(\sqrt{x}) - \frac{f(x)}{2} = 0$; $-rf(x) + f(x^r) = 0$; $-f(r) - f(x) - f(y) + f(rxy) = 0$; $-2f(r) + f\left(r^2\right) = 0$; $f\left(\frac{1}{x}\right) + f(x) = 0$; $f\left(x^{\frac{1}{n}}\right) - \frac{f(x)}{n} = 0$; $f(\sqrt[3]{x}) - \frac{f(x)}{3} = 0$; $f(r) - f(x) - f(y) + f\left(\frac{xy}{r}\right) = 0$; $-f(x) - f(y) + f(xy) = 0$; $f(x) - f(y) - f\left(\frac{x}{y}\right) = 0$; $-2f(y) - f\left(\frac{x}{y}\right) + f(xy) = 0$; $f(r) - f\left(\frac{x}{y}\right) + f\left(\frac{x}{ry}\right) = 0$; $-f(y) + f(rx) - 2f\left(\frac{x}{y}\right) + f\left(\frac{x}{ry}\right) = 0$; $-2f(y) - 2f\left(\frac{x}{y}\right) + f\left(\frac{x}{ry}\right) + f(rxy) = 0$; $-f(y) + f(ry) - f\left(\frac{x}{y}\right) + f\left(\frac{x}{ry}\right) = 0$; $-nf(x) - nf(y) + f((xy)^n) = 0$; $-af(x) - bf(y) + f\left(x^a y^b\right) = 0$; $-af(x) + bf(y) + f\left(x^a y^{-b}\right) = 0$; $-f(x) - f(y) - f(z) + f(xyz) = 0$; $-f(x) - f(y) + f(z) + f\left(\frac{xy}{z}\right) = 0$; $-f(x) + f(y) + f(z) + f\left(\frac{x}{yz}\right) = 0$; $-2f(x) - 2f(y) + f\left(x^2 y^2\right) = 0$; $-\frac{f(x)}{2} - \frac{f(y)}{2} + f\left(\sqrt{xy}\right) = 0$; $-yf(x) + f(x^y) = 0$; $-\frac{mf(x)}{n} + f\left(x^{\frac{m}{n}}\right) = 0$; $-2f(x) - 3f(y) + f\left(x^2 y^3\right) = 0$; $-2f(x) + 3f(y) + f\left(\frac{x^2}{y^3}\right) = 0$; $6f(\sqrt{x}) - f\left(x^3\right) = 0$; $9f(\sqrt[3]{x}) - f\left(x^3\right) = 0$; $3f\left(x^2\right) - 2f\left(x^3\right) = 0$; $-abf(x) + f\left(x^{ab}\right) = 0$; $-\frac{af(x)}{b} + f\left((x^a)^{\frac{1}{b}}\right) = 0$ |
| Secant | $\left(-f^2(x)\sin^2(r) + 1\right) f(-r+x)f(r+x) - f^2(x) = 0$; $f(r+x)\cos(r+x) - 1 = 0$; $f(-r+x)\cos(r-x) - 1 = 0$ |
| Cosecant | $\left(\sin^2(x) - \sin^2(y)\right) f(x-y)f(x+y) - 1 = 0$; $(\cos(x-y) - \cos(x+y)) f(x)f(y) - 2 = 0$; $f(x+y)\sin(x)\cos(y) + f(x+y)\sin(y)\cos(x) - f(x+y)\sin(x+y) = 0$; $(\sin(x)\cos(y) + \sin(y)\cos(x)) f(x+y) - 1 = 0$; $(\sin(x)\cos(y) - \sin(y)\cos(x)) f(x-y) - 1 = 0$; $f(x-y)f(x+y) - \frac{1}{\sin^2(x)\cos^2(y) - \sin^2(y)\cos^2(x)} = 0$ |
| Sinc Composite | $f(x+y)\,\mathrm{rsr}_x(x,y) - \mathrm{rsr}_{sin}(x,y) = 0$; $2xf(x)f(2y) - xf(y)f(x-y) - xf(y)f(x+y) + yf(y)f(x-y) - yf(y)f(x+y) = 0$; $xf(x)f(x-y) - xf(x)f(x+y) - yf(x)f(x-y) - yf(x)f(x+y) + 2yf(2x)f(y) = 0$ |
| Modulo | $-f(x) + f(Ry + x) = 0$; $-f(x) + f(Rk + x) = 0$; $-f(x) + f(-Rn + x) = 0$; $-f(R - x) + f(2R - x) = 0$; $f(x+y) - f(f(x) + f(y)) = 0$; $f(x + y + z) - f(f(x) + f(y) + f(z)) = 0$; $f(r + s + x) - f(f(r) + f(s) + f(x)) = 0$; $f(r + s + t + x) - f(f(r) + f(s) + f(t) + f(x)) = 0$; $-f(f(x) + f(y) + f(z)) + f(-Rw + x + y + z) = 0$; $-f(f(x) + f(y)) + f(-Rz + x + y) = 0$; $-f(f(x) + f(y)) + f(-R + x + y) = 0$; $-f(f(x) + f(y)) + f(R + x + y) = 0$; $f(xy) - f(f(x)f(y)) = 0$; $f(xyz) - f(f(x)f(y)f(z)) = 0$; $f(wxyz) - f(f(w)f(x)f(y)f(z)) = 0$; $f(rx) - f(f(r)f(x)) = 0$; $f(y(r+x)) - f(f(y)f(r+x)) = 0$; $f((r+x)(s+y)) - f(f(r+x)f(s+y)) = 0$; $f(x(R+y)) - f(f(x)f(y)) = 0$; $f((R+x)(R+y)) - f(f(x)f(y)) = 0$; $-f(f(x)f(y)) + f(Rz + xy) = 0$; $-f(f(x)f(y)) + f(R + xy) = 0$; $-f(f(x)f(y)) + f(-R + xy) = 0$; $-f(af(x)) + f(Rb + ax) = 0$; $f(ax + by) - f(af(x) + bf(y)) = 0$; $f(ax + by + cz) - f(af(x) + bf(y) + cf(z)) = 0$; $-f(af(x)) + f(x(Rb + a)) = 0$; $f(nx) - f(nf(x)) = 0$; $f(r+x) - f(f(r) + f(x)) = 0$; $f(r + x + y) - f(f(r) + f(x) + f(y)) = 0$; $-f(2x) + f(R + 2x) = 0$ |

Table 8: RSR Properties Discovered by Claude Opus 4.1 (Agentic Bitween) across RSR-Bench (continued)

| Function | Discovered RSR Properties |
|---|---|
| Modulo Multiplication | $-f(x, ry) + f(rx, y) = 0$; $f(rx, sy) - f(sx, ry) = 0$; $-f(x, f(y, z)) + f(xy, z) = 0$; $f(x, yz) - f(f(x, y), z) = 0$; $-f(ab, f(x, y)) + f(ax, by) = 0$; $-f(x, f(x, y)) + f(x^2, y) = 0$; $f(x, y^2) - f(f(x, y), y) = 0$; $-f(x, f(y, z)) + f(f(x, y), z) = 0$; $-f(x, f(y, f(z, w))) + f(xyz, w) = 0$; $f(x, wyz) - f(f(f(x, y), z), w) = 0$; $-f(x, f(x, f(x, y))) + f(x^3, y) = 0$; $f(x, y^3) - f(f(f(x, y), y), y) = 0$; $f\left(x, \frac{1}{y}\right) - f\left(\frac{x}{y}, 1\right) = 0$; $-f\left(1, \frac{y}{x}\right) + f\left(\frac{1}{x}, y\right) = 0$; $f(x, f(y, z)) - f(y, f(x, z)) = 0$; $f(f(x, y), f(z, w)) - f(f(x, z), f(y, w)) = 0$; $-f\left(x^2, f(y, z)\right) + f(xy, xz) = 0$; $f(xy, wz) - f(xz, wy) = 0$; $-f(x, 1) + f\left(\frac{x}{y}, y\right) = 0$; $-f(1, y) + f\left(x, \frac{y}{x}\right) = 0$; $-f(1, y) + f\left(\frac{1}{x}, xy\right) = 0$; $f(x, yz) - f(y, xz) = 0$; $f(xy, z) - f(yz, x) = 0$; $-f\left(x^2, f(y^2, z)\right) + f(x^2 y^2, z) = 0$; $f(x, y^2 z^2) - f(f(x, y^2), z^2) = 0$; $-f(x, f(xy, z)) + f(x^2 y, z) = 0$; $f(x, y) - f(xy, 1) = 0$; $-f(1, xy) + f(x, y) = 0$; $-f(a, f(b, f(x, y))) + f(abx, y) = 0$; $f(x, aby) - f(f(f(x, a), b), y) = 0$; $-f(rs, f(x, y)) + f(rx, sy) = 0$; $f(x, f(y, z)) - f(z, f(x, y)) = 0$; $-f(z, f(x, f(y, w))) + f(xyz, w) = 0$ |
| Integer Multiplication | $-f(r, s) - f(r, y) - f(x, s) - f(x, y) + f(r+x, s+y) = 0$; $-f(r, y) - f(x, y) + f(r+x, y) = 0$; $-f(x, s) - f(x, y) + f(x, s+y) = 0$; $f(r, s) - f(r, y) + f(x, s) - f(x, y) + f(r+x, -s+y) = 0$; $f(r, s) + f(r, y) - f(x, s) - f(x, y) + f(-r+x, s+y) = 0$; $-f(r, s) + f(r, y) + f(x, s) - f(x, y) + f(-r+x, -s+y) = 0$; $-cf(x, y) + f(cx, y) = 0$; $-cf(x, y) + f(x, cy) = 0$; $-f(x, x) + f(y, y) + f(x+y, x-y) = 0$; $-f(r, r) - f(r, y) - f(x, r) - f(x, y) + f(r+x, r+y) = 0$; $-f(x, -s+y)f(-r+x, y)f(r+x, s+y) + f(x, s+y)f(-r+x, -s+y)f(r+x, y) = 0$; $f(x, y)f(-r+x, -s+y) - f(x, -s+y)f(-r+x, y) = 0$; $f(x, y)f(r+x, s+y) - f(x, s+y)f(r+x, y) = 0$; $-4f^2(x, y) + f^2(x, 2y) = 0$; $-4f^2(x, y) + f^2(2x, y) = 0$; $-f(x-y, x+y) + f(x+y, x-y) = 0$; $-f(y, x) - f(y, y) + f(x+y, y) = 0$; $f(x, x) - f(y, y) - f(x-y, x+y) = 0$; $f(x, x+y) - f(y, x) - f(y, y) - f(x-y, x+y) = 0$; $f(x, y) - f(y, x) = 0$; $-f(x, 3y) + f(3x, y) = 0$; $-nf(x, y) + f(nx, y) = 0$; $-nf(x, y) + f(x, ny) = 0$; $-abf(x, y) + f(ax, by) = 0$; $-f(r, s) - 2f(r, y) - 2f(x, s) - 4f(x, y) + f(r+2x, s+2y) = 0$ |
| Hyperbolic Tangent | $f(r+x) - \frac{f(r)+f(x)}{f(r)f(x)+1} = 0$; $f(-r+x) - \frac{-f(r)+f(x)}{-f(r)f(x)+1} = 0$; $f(-r+x)f(r+x) - \frac{-f^2(r)+f^2(x)}{-f^2(r)f^2(x)+1} = 0$; $-\frac{2 \cdot (1-f^2(x))f(r)}{-f^2(r)f^2(x)+1} - f(-r+x) + f(r+x) = 0$; $f(r)f(-r+x) - f(r)f(r+x) + f(x)f(-r+x) + f(x)f(r+x) - 2f(-r+x)f(r+x) = 0$; $f(x)f(y)f(x-y) + f(x) - f(y) - f(x-y) = 0$; $f(x)f(x-y) + f(x)f(x+y) + f(y)f(x-y) - f(y)f(x+y) - 2f(x-y)f(x+y) = 0$; $f(2r+2x) - \frac{f(2r)+f(2x)}{f(2r)f(2x)+1} = 0$ |
| Sigmoid | $((1-f(x))(1-f(y)) + f(x)f(y))f(x+y) - f(x)f(y) = 0$; $((1-f(x))(1-f(-y)) + f(x)f(-y))f(x-y) - f(x)f(-y) = 0$; $-(1-f(y))f(x) + ((1-f(x))f(y) + (1-f(y))f(x))f(x-y) = 0$; $f(nx) - \frac{f^n(x)}{(1-f(x))^n + f^n(x)} = 0$; $-\frac{kf(x)}{(k-1)f(x)+1} + f(x+\log(k)) = 0$; $f(x+y+z) - \frac{f(z)f(x+y)}{(1-f(z))(1-f(x+y))+f(z)f(x+y)} = 0$ |

Table 8: RSR Properties Discovered by Claude Opus 4.1 (Agentic Bitween) across RSR-Bench (continued)

| Function | Discovered RSR Properties |
|---|---|
| Softmax2 1 | $-f(x,y) + f(r+x, r+y) = 0$; $-(1 - f(x,y)) f(x,y) + f(x,y)f(y,x) = 0$; $-f(x,y) + f(-r+x, -r+y) = 0$; $-f(x, r+y) + f(-r+x, y) = 0$; $-f(x, -r+y) + f(r+x, y) = 0$; $f(x,y)f(y,z)f(z,x) - f(x,z)f(y,x)f(z,y) = 0$; $-f(x,0) + f(x+y,y) = 0$; $-f(0,y) + f(x, x+y) = 0$; $f(x,y) - \frac{1}{e^{-x+y}+1} = 0$; $f(0,y) - f(x, x+y) = 0$; $f(x, x+y) + f(x+y, x) - 1 = 0$; $f(y, x+y) + f(x+y, y) - 1 = 0$; $f(x,0) + f(y, x+y) - 1 = 0$; $f(2x, 2y) - f(x-y, -x+y) = 0$; $f(x,y) - f(x-y, 0) = 0$; $f(y, r+x) + f(r+x, y) - 1 = 0$; $f(x, r+y) + f(r+y, x) - 1 = 0$; $f(r+x, s+y) + f(s+y, r+x) - 1 = 0$; $f(x,y) - f(r+x-y, r) = 0$; $-f^2(x,y) + f(x,y)f(r+x, r+y) = 0$; $f(x, r+y) - f(-r+x, y) = 0$; $-f(s+x, y) + f(r+s+x, r+y) = 0$; $f(x, x-y) - f(y, 0) = 0$; $-f(0,y) + f(x-y, x) = 0$ |
| Softmax2 2 | $-f(x,y) + f(r+x, r+y) = 0$; $f(y,x) + f(-r+x, -r+y) - 1 = 0$; $f(y,x) + f(r+x, r+y) - 1 = 0$; $f(x, r+x) + f(r+x, x) - 1 = 0$; $f(0, -x+y) + f(y,x) - 1 = 0$; $f(y,x) + f(x-y, 0) - 1 = 0$; $-f^2(y,x) + 2f(y,x) + f^2(x-y, 0) - 1 = 0$; $-\frac{e^x + e^y}{e^y + e^{c+x}} + \frac{f(c+x, y)}{f(x,y)} = 0$; $-f(x,y) + f(a+x, a+y) = 0$; $f(y, r+x) + f(r+x, y) - 1 = 0$; $f(x, r+y) + f(r+y, x) - 1 = 0$; $f(r+x, s+y) - f(r-s+x, y) = 0$; $-\frac{a}{a+e^x} + f(x, \log(a)) = 0$; $-\frac{b}{a+b} + f(\log(a), \log(b)) = 0$; $f(x,y)f(y,z)f(z,x) - f(x,z)f(y,x)f(z,y) = 0$; $f(a+x, b+y) - f(a-b+x, y) = 0$; $f(x,y) - f(x-z, y-z) = 0$; $(ae^x + e^y) f(x + \log(a), y) - e^y = 0$; $-be^y + (be^y + e^x) f(x, y + \log(b)) = 0$; $-\frac{be^y}{ae^x + be^y} + f(x + \log(a), y + \log(b)) = 0$; $f(-r+x, -s+y) - f(-r+s+x, y) = 0$ |
| Logistic | $-L + f(x) + f(-x + 2x_0) = 0$; $-L + f(nr+x) + f(-nr-x+2x_0) = 0$; $f(r)f(-r+x) - f(r)f(r+x) + f(x)f(-r+x) + f(x)f(r+x) - f(x) - 2f(-r+x)f(r+x) + f(r+x) = 0$; $f(r)f(-r+x_0) - f(r)f(r+x_0) + f(x_0)f(-r+x_0) + f(x_0)f(r+x_0) - f(x_0) - 2f(-r+x_0)f(r+x_0) + f(r+x_0) = 0$; $f(-r)f^2(-r+x) + f(r)f^2(-r+x) - f^2(-r+x) = 0$; $f(-r)f^2(r+x) + f(r)f^2(r+x) - f^2(r+x) = 0$ |
| Logistic Scaled | $-L + f(x) + f(-x + 2x_0) = 0$; $-L + f(-r+x_0) + f(r+x_0) = 0$; $-Lf(x) + f^2(x) + f(x)f(-x + 2x_0) = 0$; $f(r)f(-r+x) - f(r)f(r+x) + f(x)f(-r+x) + f(x)f(r+x) - 3f(x) - 2f(-r+x)f(r+x) + 3f(r+x) = 0$; $f(2r)f(-2r+x) - f(2r)f(2r+x) + f(x)f(-2r+x) + f(x)f(2r+x) - 3f(x) - 2f(-2r+x)f(2r+x) + 3f(2r+x) = 0$; $-L + f(r+x) + f(-r-x+2x_0) = 0$; $-L + f(-r+x) + f(r-x+2x_0) = 0$; $f(-nr) + f(nr) - 3 = 0$; $-L + f(-nr+x_0) + f(nr+x_0) = 0$; $Lf(x) - f^2(x) - f(x)f(-x+2x_0) = 0$; $(L - f(x)) f(x) - f(x)f(-x+2x_0) = 0$; $-L + f(nr+x) + f(-nr-x+2x_0) = 0$; $-L + f(x) + f(-x+2x_0) = 0$; $-L + f(-ar) + f(ar) = 0$; $-L + f(ar+x) + f(-ar-x+2x_0) = 0$ |
| Square Loss | $-2f(x) - 2f(1-r) + f(-r+x) + f(r+x) = 0$; $-2f(x) - 2f(1-y) + f(x-y) + f(x+y) = 0$; $-2f(ax) + f(ax-by) + f(ax+by) - 2f(-by+1) = 0$; $-2x - 2y - (1-y)(2-2x) - f(x) - f(y) + f(x+y) + 3 = 0$; $\left(\frac{x}{2} - \frac{y}{2}\right)^2 - \frac{f(x)}{2} - \frac{f(y)}{2} + f\left(\frac{x}{2} + \frac{y}{2}\right) = 0$; $-\left(1 - \frac{x}{c}\right)^2 + f\left(\frac{x}{c}\right) = 0$; $-h - 2x + 2 + \frac{-f(x) + f(h+x)}{h} = 0$; $-2h^2 - 2f(x) + f(-h+x) + f(h+x) = 0$; $-(1-c)^2 f(x) + f(c(1-x) + x) = 0$ |
| Savage Loss Basis | $f(x+y) - \frac{1}{(g(x)g(y)+1)^2} = 0$; $f(x-y) - \frac{g^2(y)}{(g(x)+g(y))^2} = 0$; $(g(x)g(y)+1)^2 f(x+y) - 1 = 0$; $(g(x)+g(y))^2 f(x-y) - g^2(y) = 0$; $(g(x-y)+1)^2 f(x-y) - 1 = 0$; $(g(r+x)+1)^2 f(r+x) - 1 = 0$; $(g(-r+x)+1)^2 f(-r+x) - 1 = 0$; $(g(x)+g(r+x))^2 f(r) - g^2(x) = 0$ |
| Arctangent | $-\frac{x+y}{-xy+1} + \tan(f(x) + f(y)) = 0$; $-\frac{x-y}{xy+1} + \tan(f(x) - f(y)) = 0$; $\cos(f(x) + f(y)) - \frac{-xy+1}{\sqrt{(x^2+1)(y^2+1)}} = 0$ |

Table 8: RSR Properties Discovered by Claude Opus 4.1 (Agentic Bitween) across RSR-Bench (continued)

| Function | Discovered RSR Properties |
|---|---|
| ReLU | $f(\max(x,y)) - \max(f(x), f(y)) = 0$; $f(\min(x,y)) - \min(f(x), f(y)) = 0$; $-f(\max(x,y)) + \max(f(x), f(y)) = 0$; $-f(\min(x,y)) + \min(f(x), f(y)) = 0$; $-f(x)f(y) + f(f(x)f(y)) = 0$ |
| Square Root | $-2r - f^2(-r+x) + f^2(r+x) = 0$; $-2x + f^2(-r+x) + f^2(r+x) = 0$; $-2f^2(x) + f^2(-r+x) + f^2(r+x) = 0$; $f^2(r) + f^2(x) - f^2(r+x) = 0$; $r - x + f^2(-r+x) = 0$; $-r - x + f^2(r+x) = 0$; $-2f(r)f^2(x) + f(r)f^2(-r+x) + f(r)f^2(r+x) = 0$; $-2f^3(x) + f(x)f^2(-r+x) + f(x)f^2(r+x) = 0$; $r^2 - x^2 + f^2(-r+x)f^2(r+x) = 0$ |
| Cube Root | $-x - y + f^3(x+y) = 0$; $-x + y + f^3(x-y) = 0$; $-xy + f^3(xy) = 0$; $-2f^3(x) + f^3(-r+x) + f^3(r+x) = 0$; $-2f^3(y) + 3f^3(2y) + 2f^3(x-y) - 2f^3(x+y) = 0$; $-6f^3(y) + 2f^3(2y) - f^3(x-y) + f^3(x+y) = 0$; $-6f^3(x) + 2f^3(2x) + f^3(x-y) + f^3(x+y) = 0$; $f^3(x) + 2f^3(2x) + f^3(y) + 2f^3(2y) - 5f^3(x+y) = 0$; $f^3(x) + 2f^3(2x) - f^3(y) - 2f^3(2y) - 5f^3(x-y) = 0$; $-f^3(r) + 10f^3(2r) - 3f^3(3r) + 2f^3(-2r+x) + f^3(-r+x) - f^3(r+x) - 2f^3(2r+x) = 0$; $-f^3(r) - 2f^3(2r) + 5f^3(3r) + 2f^3(-2r+x) + f^3(-r+x) - f^3(r+x) - 2f^3(2r+x) = 0$; $60f^3(r) - 5f^3(2r) - 8f^3(3r) + 5f^3(-2r+x) + 3f^3(-r+x) - 3f^3(r+x) - 5f^3(2r+x) = 0$ |
| Floor | $-f(x) - f(y) + f(x+y) - f(x+y-\lfloor x \rfloor - \lfloor y \rfloor) = 0$; $-f(x) - f(y) - f(z) + f(x+y+z) - f(x+y+z-\lfloor x \rfloor - \lfloor y \rfloor - \lfloor z \rfloor) = 0$; $-f(x)f(y) + f(xy) - f(xy - \lfloor x \rfloor \lfloor y \rfloor) = 0$; $-f(r) - f(x) + f(r+x) - f(r+x - \lfloor r \rfloor - \lfloor x \rfloor) = 0$; $f(r) - f(x) + f(-r+x) - f(-r+x + \lfloor r \rfloor - \lfloor x \rfloor) = 0$; $-2f(r) - f(x) + f(2r+x) - f(2r+x - 2\lfloor r \rfloor - \lfloor x \rfloor) = 0$; $-f(s) - f(r+x) + f(r+s+x) - f(r+s+x - \lfloor s \rfloor - \lfloor r+x \rfloor) = 0$ |
| Ceiling | $-f(x) - f(y) + f(x + f(y)) = 0$; $-f(x) + f(y) + f(x - f(y)) = 0$; $-f(x) - f(y) + f(f(x) + f(y)) = 0$; $-f(x) + f(y) + f(f(x) - f(y)) = 0$; $-f(x)f(y) + f(f(x)f(y)) = 0$; $-f(x+y) + f(f(x+y)) = 0$; $-f(x) - f(y) + f(y + f(x)) = 0$; $f(x+y - f(x+y)) = 0$; $-f(x) - f(y) - f(z) + f(f(x) + f(y) + f(z)) = 0$; $-f(x)f(y)f(z) + f(f(x)f(y)f(z)) = 0$ |
| Fractional Part | $f(x) + f(y) - f(x+y) - \lfloor f(x) + f(y) \rfloor = 0$; $f(x) + f(y) - f(x+y) + \lfloor x \rfloor + \lfloor y \rfloor - \lfloor x+y \rfloor = 0$; $-f(x) + f(x + \lfloor y \rfloor) = 0$; $f(xy) - f(xy - \lfloor x \rfloor \lfloor y \rfloor) = 0$ |
| Error Function | $f(-ax) + f(ax) = 0$; $f\left(-\frac{x}{a}\right) + f\left(\frac{x}{a}\right) = 0$ |
| Gamma | $-yf(x)f(y) + f(x)f(y+1) = 0$; $-xyf(x)f(y) + f(x+1)f(y+1) = 0$; $-yf(y)f(x-1) + f(x-1)f(y+1) = 0$; $-x(x+1)f(x)f(y) + f(y)f(x+2) = 0$ |
| Exp Sin | $-f^{2\cos(r)}(x) + f(-r+x)f(r+x) = 0$ |
| Log Cosine | $f(x-y) + f(x+y) - \log(|\cos(x-y)|) - \log(|\cos(x+y)|) = 0$ |
| Square Root of 1+X² | $-x^2y^2 + f^2(xy) - 1 = 0$; $-2xy - f^2(x) - f^2(y) + f^2(x+y) + 1 = 0$; $2xy - f^2(x) - f^2(y) + f^2(x-y) + 1 = 0$; $-2f^2(x) - 2f^2(y) + f^2(x-y) + f^2(x+y) + 2 = 0$; $4x^2y^2 - (x^2 + y^2 + 1)^2 + f^2(x-y)f^2(x+y) = 0$; $4x^2y^2 - (x^2 + y^2 + 1)^2 + f^2(x-y)f^2(x+y) = 0$; $4x^2y^2 - (f^2(x) + f^2(y) - 1)^2 + f^2(x-y)f^2(x+y) = 0$; $-x^2 + y^2 + f^2(x) - f^2(y) = 0$; $-2r^2 - 2f^2(x) + f^2(-r+x) + f^2(r+x) = 0$; $-4rx - f^2(-r+x) + f^2(r+x) = 0$; $-(x^2+1)(y^2+1) + f^2(x)f^2(y) = 0$; $2r^2 + 2f^2(x) - f^2(-r+x) - f^2(r+x) = 0$; $-\sqrt{x^2 - 2xy + y^2 + 1}\sqrt{x^2 + 2xy + y^2 + 1} + f(x-y)f(x+y) = 0$; $r^2 + f^2(x) - \frac{f^2(-r+x)}{2} - \frac{f^2(r+x)}{2} = 0$; $-\sqrt{c^2 + f^2(x) - 1} + f\left(\frac{x}{|c|}\right)|c| = 0$ |
| Absolute Value | $-f(x)|c| + f(cx) = 0$; $-f(x)f(y) + f(xy) = 0$; $-\frac{f(x)}{f(y)} + f\left(\frac{x}{y}\right) = 0$; $-2f^2(r) - 2f^2(x) + f^2(-r+x) + f^2(r+x) = 0$; $-14f^2(x) - 2f^2(y) - f^2(x-y) - f^2(x+y) + 2f^2(2x-y) + 2f^2(2x+y) = 0$; $-f^2(x) - f^2(y) + f^2\left(\sqrt{x^2 + y^2}\right) = 0$; $f((-r+x)(r+x)) - f(-r^2 + x^2) = 0$; $-f^2(r)f^2(x) + f^2(rx) = 0$; $f^2(x)f^2(y) - f^2(xy) = 0$ |
| Sign | $f^3(x+y) - f(x+y) = 0$; $f^3(x-y) - f(x-y) = 0$; $f^3(xy) - f(xy) = 0$; $f^5(xy) - f(xy) = 0$ |

Table 8: RSR Properties Discovered by Claude Opus 4.1 (Agentic Bitween) across RSR-Bench (continued)

| Function | Discovered RSR Properties |
|---|---|
| Gudermannian | $\tan(f(x) + f(y)) - \frac{\sinh(x) + \sinh(y)}{-\sinh(x)\sinh(y) + 1} = 0$; $(-\sinh(x)\sinh(y) + 1)\tan(f(x) + f(y)) - \sinh(x) - \sinh(y) = 0$; $\tan(f(x) - f(y)) - \frac{\sinh(x) - \sinh(y)}{\sinh(x)\sinh(y) + 1} = 0$; $(\sinh(x)\sinh(y) + 1)\tan(f(x) - f(y)) - \sinh(x) + \sinh(y) = 0$; $f(x + y) - \mathrm{atan}(\sinh(x)\cosh(y) + \sinh(y)\cosh(x)) = 0$; $f(x + y) - \mathrm{atan}\left(\sqrt{\tan^2(f(x)) + 1}\tan(f(y)) + \sqrt{\tan^2(f(y)) + 1}\tan(f(x))\right) = 0$; $f(x + y) - \mathrm{atan}\left(\sqrt{\sinh^2(x) + 1}\sinh(y) + \sqrt{\sinh^2(y) + 1}\sinh(x)\right) = 0$; $\sinh(r)\cosh(r + x) + \sinh(x) - \sinh(r + x)\cosh(r) = 0$; $-\sinh(r)\cosh(r - x) + \sinh(x) + \sinh(r - x)\cosh(r) = 0$; $2\sinh(x)\cosh(r) + \sinh(r - x) - \sinh(r + x) = 0$; $2\sinh(r)\cosh(x) - \sinh(r - x) - \sinh(r + x) = 0$; $2\tan(f(x))\cosh(r) - \tan(f(-r + x)) - \tan(f(r + x)) = 0$; $2\sqrt{\sinh^2(r) + 1}\tan(f(x)) - \tan(f(-r + x)) - \tan(f(r + x)) = 0$; $-\frac{\tan(f(-r+x)) + \tan(f(r+x))}{2\cosh(r)} + \tan(f(x)) = 0$; $-\frac{\tan(f(-r+x)) + \tan(f(r+x))}{2\sqrt{\sinh^2(r) + 1}} + \tan(f(x)) = 0$; $f(x) - \mathrm{atan}\left(\frac{\tan(f(-r+x)) + \tan(f(r+x))}{2\cosh(r)}\right) = 0$; $f(x) - \mathrm{atan}\left(\frac{-\sinh(r-x) + \sinh(r+x)}{2\cosh(r)}\right) = 0$; $f(x) - \mathrm{atan}\left(\frac{-\sinh(r-x) + \sinh(r+x)}{2\sqrt{\sinh^2(r) + 1}}\right) = 0$; $2\sqrt{\sinh^2(r) + 1}\tan(f(x)) - \tan(f(-r + x)) - \tan(f(r + x)) = 0$; $f(r) - \mathrm{atan}\left(\frac{-\tan(f(-r+x)) + \tan(f(r+x))}{2\cosh(x)}\right) = 0$; $f(r) - \mathrm{atan}\left(\frac{\sinh(r-x) + \sinh(r+x)}{2\cosh(x)}\right) = 0$; $f(r + x) - \mathrm{atan}(\sinh(r)\cosh(x) + \sinh(x)\cosh(r)) = 0$; $f(-r + x) + \mathrm{atan}(\sinh(r)\cosh(x) - \sinh(x)\cosh(r)) = 0$; $2\tan(f(x))\cosh(r) - \tan(f(-r + x)) - \tan(f(r + x)) = 0$; $2\tan(f(r))\cosh(x) + \tan(f(-r + x)) - \tan(f(r + x)) = 0$ |
| 2 to X | $-f(x)f(y) + f(x + y) = 0$; $-f(x) + f(y)f(x - y) = 0$; $-\frac{f(x)}{f(y)} + f(x - y) = 0$; $-f^n(x) + f(nx) = 0$; $-f(2x) + f(x - y)f(x + y) = 0$; $-f^2(x)f(-y)f(y) + f(x - y)f(x + y) = 0$; $-f(r)f(x) + f(r + x) = 0$; $-f(-r)f(x) + f(-r + x) = 0$; $f(-r + x) - \frac{f(x)}{f(r)} = 0$; $-f^4(y)f^2(x - y) + f^2(x + y) = 0$; $-f^2(y)f(x - y) + f(x + y) = 0$; $f(-r)f(r + x) - f(x) = 0$; $f(-r)f(x) - f(-r + x) = 0$; $f(r)f(-r + x) - f(x) = 0$; $-f^2(x) + f(-r + x)f(r + x) = 0$; $f(r)f(x) - f(r + x) = 0$; $-f^3(y)f^2(x - y) + f(2x + y) = 0$; $-f^3(y)f(x - y) + f(x + 2y) = 0$; $-f^2(x)f(y) + f(2x + y) = 0$; $-f(x)f^2(y) + f(x + 2y) = 0$; $f(x)f(z) - f(y)f(x - y + z) = 0$; $-f(x)f(y)f(z) + f(x + y + z) = 0$; $-f(x)f(-y)f(z) + f(x - y + z) = 0$; $-\frac{f(x)f(z)}{f(y)} + f(x - y + z) = 0$; $-f(2x) + f(-r + x)f(r + x) = 0$; $f^2(r + x) - f(2r + 2x) = 0$; $f^2\left(\frac{x}{2} + \frac{y}{2}\right) - f(x + y) = 0$; $-f(x)f(y)f(-z) + f(x + y - z) = 0$; $-\frac{f(x)f(y)}{f(z)} + f(x + y - z) = 0$; $-f(2y) + \frac{f(x+y)}{f(x-y)} = 0$; $-f(x) + \frac{f(x+y)}{f(y)} = 0$; $-f(x) + \frac{f(x-y)}{f(-y)} = 0$; $-f(x) + \frac{f(x+y+z)}{f(y+z)} = 0$; $-f(4y) + \frac{f^2(x+y)}{f^2(x-y)} = 0$ |

Table 8: RSR Properties Discovered by Claude Opus 4.1 (Agentic Bitween) across RSR-Bench (continued)

| Function | Discovered RSR Properties |
|---|---|
| 10 to X | $-f(x)f(y) + f(x + y) = 0$; $-f(x)f(-y) + f(x - y) = 0$; $-\frac{f(x)}{f(y)} + f(x - y) = 0$; $-f^n(x) + f(nx) = 0$; $-f^2(x) + f(-r + x)f(r + x) = 0$; $-f(-r)f(r + x) + f(x) = 0$; $-f(r)f(-r + x) + f(x) = 0$; $f^{\frac{1}{n}}(x) - f\left(\frac{x}{n}\right) = 0$; $-f(x)f(y)f(z) + f(x + y + z) = 0$; $-\frac{f(x)}{f(y)f(z)} + f(x - y - z) = 0$; $-\frac{f(x)f(y)}{f(z)} + f(x + y - z) = 0$; $-\frac{f(x)f(z)}{f(y)} + f(x - y + z) = 0$; $-f(r)f^2(x) + f(r + 2x) = 0$; $-f^2(r)f(x) + f(2r + x) = 0$; $f(-2r + x) - \frac{f(x)}{f^2(r)} = 0$; $-f(2y)f(x - y) + f(x + y) = 0$; $-f^2(y)f(x - y) + f(x + y) = 0$; $f(-r)f(r) - 1 = 0$; $f(-r)f(r + x) - f(x) = 0$; $f(r)f(-r + x) - f(x) = 0$; $-f(x)f(y)f(-z) + f(x + y - z) = 0$; $-\frac{f(x)f(y)}{f(z)} + f(x + y - z) = 0$; $-f^2(r)f(x) + f(2r + x) = 0$; $-f(2r)f(x) + f(2r + x) = 0$; $-f^a(x) + f(ax) = 0$; $-f^2(x)f(-y)f(y) + f(x - y)f(x + y) = 0$; $-f^2(x) + f(x - y)f(x + y) = 0$; $-f(x)f(y) + f^2\left(\frac{x}{2} + \frac{y}{2}\right) = 0$; $\sqrt{f(x)f(y)} - f\left(\frac{x}{2} + \frac{y}{2}\right) = 0$; $-f(2x)f(2y) + f(2x + 2y) = 0$; $-f^y(x) + f(xy) = 0$; $f^y(x) - f(xy) = 0$; $-f^{\frac{1}{y}}(x) + f\left(\frac{x}{y}\right) = 0$; $-f^a(x)f^b(y) + f(ax + by) = 0$; $-f^a(x)f^b(y)f^c(z) + f(ax + by + cz) = 0$; $-af(x) + f\left(x + \frac{\log(a)}{\log(10)}\right) = 0$; $-\frac{f^2(x)}{f(y)} + f(2x - y) = 0$; $-\frac{f^3(x)}{f^2(y)} + f(3x - 2y) = 0$; $-\left(f(-r) + f(r)\right)f(x) + f(-r + x) + f(r + x) = 0$; $-\left(-f(-r) + f(r)\right)f(x) - f(-r + x) + f(r + x) = 0$ |
| Pade 1,1 | $f(x)f(x - y) + f(x)f(x + y) + 2f(x) - 2f(x - y)f(x + y) - f(x - y) - f(x + y) = 0$; $-\frac{x}{2} - \frac{y}{2} + \frac{f(x+y)-1}{f(x+y)+1} = 0$; $-\frac{x}{2} + \frac{y}{2} + \frac{f(x-y)-1}{f(x-y)+1} = 0$ |
| Pade 2,2 | $f(-x - y)f(-x + y)f(x - y)f(x + y) - 1 = 0$ |
| Continued Fraction Golden | $-2f(x)f(-r + x) - 2f(x)f(r + x) + 6f(x) + 4f(-r + x)f(r + x) - 3f(-r + x) - 3f(r + x) = 0$; $2f(rx)f(rx - s) + 2f(rx)f(rx + s) - 6f(rx) - 4f(rx - s)f(rx + s) + 3f(rx - s) + 3f(rx + s) = 0$; $2f(r)f(r - s) + 2f(r)f(r + s) - 6f(r) - 4f(r - s)f(r + s) + 3f(r - s) + 3f(r + s) = 0$; $2f(x)f(-r - s + x) - 4f(x)f(-r + s + x) - 4f(x)f(r - s + x) + 2f(x)f(r + s + x) + 6f(x) - 4f(-r - s + x)f(r + s + x) + 3f(-r - s + x) + 8f(-r + s + x)f(r - s + x) - 6f(-r + s + x) - 6f(r - s + x) + 3f(r + s + x) = 0$; $-2f(r)f(r + x) + 4f(r)f(r + 2x) - 3f(r) - 2f(r + x)f(r + 2x) + 6f(r + x) - 3f(r + 2x) = 0$ |
| Continued Fraction Tan | $-nx\left(-n^2x^2 + 15\right) + \left(-6n^2x^2 + 15\right)f(nx) = 0$; $\left(15n^2 - 6x^2\right)f\left(\frac{x}{n}\right) - \frac{x\left(15n^2 - x^2\right)}{n} = 0$; $15n^2f\left(\frac{x}{n}\right) - 15nx - 6x^2f\left(\frac{x}{n}\right) + \frac{x^3}{n} = 0$ |
| Mobius Simple | $-x + f\left(\frac{-b + dx}{a - cx}\right) = 0$; $-r(ad - bc) + (cx + d)\left(-f(x) + f(r + x)\right)(cr + cx + d) = 0$; $-\frac{(-ar + ax + b)(ar + ax + b)}{(-cr + cx + d)(cr + cx + d)} + f(-r + x)f(r + x) = 0$; $2rf(r)f(-x) - 2rf(-x)f(-r + x) - 2xf(r)f(-x) + 2xf(r)f(-r + x) + 9f(r)f(-x)f(-r + x) - f(r)f(-x) - f(r)f(-r + x) - f(-x)f(-r + x) = 0$; $-\frac{(x_1 - x_2)(x_3 - x_4)}{(x_1 - x_4)(-x_2 + x_3)} + \frac{(f(x_1) - f(x_2))(f(x_3) - f(x_4))}{(f(x_1) - f(x_4))(-f(x_2) + f(x_3))} = 0$; $f(x)f(y) - f(x + y) - \frac{-ax - ay - b + (cx + cy + d)f(x)f(y)}{cx + cy + d} = 0$; $-\frac{arx + b}{crx + d} + f(rx) = 0$; $f(r)f(s) - f(r + s) - \frac{-ar - as - b + (cr + cs + d)f(r)f(s)}{cr + cs + d} = 0$; $-\frac{(r+s)(ad - bc)}{(cx + d)(cr + cs + cx + d)} - f(x) + f(r + s + x) = 0$; $-\frac{ad - bc}{(-cr + cx + d)(cr + cx + d)} + \frac{-f(-r + x) + f(r + x)}{2r} = 0$ |

Table 8: RSR Properties Discovered by Claude Opus 4.1 (Agentic Bitween) across RSR-Bench (continued)

| Function | Discovered RSR Properties |
|---|---|
| Mobius Inversion | $-f(x)f(y)+f(xy) = 0$; $-yf(x)+f\left(\frac{x}{y}\right) = 0$; $f(x-y)f(x+y)-f(x^2-y^2) = 0$; $-2xf(x^2-y^2)+f(x-y)+f(x+y) = 0$; $f(cx)-\frac{f(x)}{c} = 0$; $f(x)f(y)f(z)-f(xyz) = 0$; $f(y)f(x+y)-f(xy+y^2) = 0$; $f(y)f(x-y)-f(xy-y^2) = 0$; $f(x)f(x+y)-f(x^2+xy) = 0$; $-f(x)f(y)+f(x)f(x+y)+f(y)f(x+y) = 0$; $f(x)f(y)+f(x)f(x-y)-f(y)f(x-y) = 0$; $-f(y)f(x-y)+f(y)f(x+y)+2f(x-y)f(x+y) = 0$; $xf(x+y)+yf(x+y)-1 = 0$; $xf(x-y)-yf(x-y)-1 = 0$; $(x+y)f(x+y)-1 = 0$; $(x-y)f(x-y)-1 = 0$; $x^2f(x-y)f(x+y)-y^2f(x-y)f(x+y)-1 = 0$; $-(x+y+z)f(xyz)+f(x)f(y)+f(x)f(z)+f(y)f(z) = 0$; $f(xy+z)-\frac{f(z)}{xyf(z)+1} = 0$; $f(ax+by)-\frac{1}{ax+by} = 0$; $f(x+y)-\frac{1}{x+y} = 0$; $f(x-y)-\frac{1}{x-y} = 0$ |
| Mobius Cayley | $f(x)f(x-y)+f(x)f(x+y)-2f(x)-2f(x-y)f(x+y)+f(x-y)+f(x+y) = 0$; $f(x)f(yz)f(xyz)-f(x)-f(yz)+f(xyz) = 0$; $f(y)f(z)f(yz)-f(y)-f(z)+f(yz) = 0$; $f(x)f(y)f(xy)-f(x)-f(y)+f(xy) = 0$ |
| Exp X² | $f(x)f(y)-f\left(\sqrt{x^2+y^2}\right) = 0$; $-f^{a^2}(x)+f(ax) = 0$; $-f^2(x)f^2(y)+f(x-y)f(x+y) = 0$; $-f^2(a)f^2(x)+f(-a+x)f(a+x) = 0$; $-f^2(2x)f^2(y)+f(2x-y)f(2x+y) = 0$; $-f^2(x)f^2(2y)+f(x-2y)f(x+2y) = 0$; $-f^2(ax)f^2(by)+f(ax-by)f(ax+by) = 0$; $-f^2(z)f^2(x+y)+f(x+y-z)f(x+y+z) = 0$; $-f^2(x)f^2(y+z)+f(x-y-z)f(x+y+z) = 0$; $-f^2(x)f^2(y-z)+f(x-y+z)f(x+y-z) = 0$; $-f^2(x)f^4(y)f^2(z)+f(x-y)f(x+y)f(y-z)f(y+z) = 0$; $-f^4(x)f^4(y)f^4(z)+f(x-y-z)f(x-y+z)f(x+y-z)f(x+y+z) = 0$; $f(\sqrt{2}x)f(\sqrt{2}y)-f(x-y)f(x+y) = 0$; $f(x)f(y)f(z)-f\left(\sqrt{x^2+y^2+z^2}\right) = 0$; $-f^{y^2}(x)+f(xy) = 0$ |
| Exp Cosine | $f(r+x)f(r+x+\pi)-1 = 0$; $f(-r+x)f(-r+x+\pi)-1 = 0$; $f\left(r+x-\frac{\pi}{2}\right)f\left(r+x+\frac{\pi}{2}\right)-1 = 0$; $f\left(-r+x-\frac{\pi}{2}\right)f\left(-r+x+\frac{\pi}{2}\right)-1 = 0$; $f\left(r-\frac{\pi}{2}\right)f\left(r+\frac{\pi}{2}\right)-1 = 0$ |
| Fourth Power | $-c^4f(x)+f(cx) = 0$; $-12r^2x^2-2f(r)-2f(x)+f(-r+x)+f(r+x) = 0$; $f(x-y)f(x+y)-f(x^2-y^2) = 0$; $-f(x)f(y)+f(xy) = 0$; $-\frac{f(x)}{f(y)}+f\left(\frac{x}{y}\right) = 0$; $-24x^2y^2-24x^2z^2-24y^2z^2-4f(x)-4f(y)-4f(z)+f(x-y-z)+f(x-y+z)+f(x+y-z)+f(x+y+z) = 0$; $-24f(r)+6f(x)+f(-2r+x)-4f(-r+x)-4f(r+x)+f(2r+x) = 0$; $-20f(x)+f(-3r+x)-6f(-2r+x)+15f(-r+x)+15f(r+x)-6f(2r+x)+f(3r+x) = 0$; $-48x^2y^2-2f(2x)-2f(y)+f(2x-y)+f(2x+y) = 0$; $f(-a+x)f(a+x)-f(-a^2+x^2) = 0$; $12x^2y^2+2f(x)+2f(y)-f(x-y)-f(x+y) = 0$ |

