# OpenReview forum: "Learning Randomized Reductions"
_ICLR.cc/2026/Conference — ICLR 2026 Conference Desk Rejected Submission_

### Official Review · Reviewer_u98M · 2025-10-23

**Soundness:** 2
**Presentation:** 3
**Contribution:** 2
**Rating:** 4
**Confidence:** 4

**Summary:**

This work proposes Bitween, an automated framework for BSRs from mathematical functions. Specifically, the authors claim that the vanilla Bitween can outperform traditional symbolic regression methods under fixed query functions. Moreover, agentic Bitween can leverage LLMs to dynamically discover novel query functions. To justify the effectiveness of the proposed method, the experiments are conducted on RSR-Bench.

**Strengths:**

1.The paper is in general well written and the method is easy to understand.

2.The combination of linear regression and symbolic verification is practical and well-motivated.

**Weaknesses:**

1.The query function generation process of agentic Bitween is opaque, and the convergence behavior is not guaranteed.

2.The experimental design focused on quantity and lacked in-depth analysis. For example, there is a lack of in-depth discussion on whether the designed RSRs have mathematical meaning.

3.Authors argue that the proposed method outperforms sophisticated methods like GP and SR. The interpretability is the merit of GP and SR, the author should analyze this point.

4.The study mainly focuses on the performance in comparisons. However, the expression size and the search cost also need consideration, especially in practical terms.

5.More up-to-date peer competitors should be compared, such as RAG-SR [1], ParFam [2], and Metasymnet [3].

6.The authors argue that BSRs are important for complexity theory and cryptography. Thus, are there any downstream applications or case studies demonstrating practical value of the proposed method?

*References:*

[1] Zhang H, Chen Q, Banzhaf W, et al. RAG-SR: Retrieval-augmented generation for neural symbolic regression[C]//The Thirteenth International Conference on Learning Representations. 2025.

[2] Scholl P, Bieker K, Hauger H, et al. ParFam--(Neural Guided) Symbolic Regression via Continuous Global Optimization[C]//The Thirteenth International Conference on Learning Representations. 2025.

[3] Li Y, Li W, Yu L, et al. Metasymnet: A tree-like symbol network with adaptive architecture and activation functions[C]//Proceedings of the AAAI Conference on Artificial Intelligence. 2025, 39(25): 27081-27089.

**Questions:**

Please see the Weaknesses part.

---

> ### Author Response · Authors · 2025-11-18
>
> **Q1: Agentic Bitween Architecture and Convergence**
>
> We augmented Implementation section (blue highlighted) clarifying Agentic Bitween's architecture.
>
> A property is a SymPy expression that equals zero. Example: `−f(x) − f(y) + f(x+y) = 0`. A-Bitween has three tools: (1) `infer_property_tool`: uses `V-Bitween` providing functional terms (variables `v_q` in Algorithm 1) implicitly encoding query functions. Example: LLM provides `f(x/(1+x))`, containing query function `x/(1+x)`, (2) `symbolic_verify_tool`: verifies properties using SymPy simplification, it cannot distinguish RSR vs non-RSR, but verifies zero-equality, (3) `sequential_thinking_tool` that helps with LLM reasoning, empirically increasing tool usage and discovery.
>
> Convergence is not guaranteed. We removed "converge" from line L049.
>
> **Q2&3: Mathematical Meaning and Interpretability**
>
> We do not compete with GP/SR; they serve as backends within our framework (see Appendix "Experimental Hyperparameters"). As explained in revised Related Work (Section 3, blue), we demonstrated that our linear regression backend (`V-Bitween-LR`) outperforms alternative backends `V-Bitween-GPLearn`, `V-Bitween-PySR`, and `V-Bitween-MILP` for RSR discovery.
>
> RSRs have clear mathematical meaning enabling practical applications. Bitween produces interpretable rational expressions (e.g., `f(x) = (1/2)·f(x+r) + (3/5)·f(r)`) similar to GP/SR output, using bounded denominators (max=20) for readability. Our interpretability matches GP/SR: _exact symbolic expressions with rational coefficients_.
>
> Each RSR enables: (1) Self-correcting programs, (2) Instance hiding protocols, (3) Physical attack countermeasures (see Q6). RSRs with fewer query functions preferred for efficienct computation.
>
> **Q4: Expression Size and Search Cost**
>
> Efficiency analysis (Figure 4, Evaluation section) shows V-Bitween-LR achieves needs less time than alternative backends. RSR properties are compact rational expressions with bounded denominators (max=20 documented in appendix), keeping expression size manageable and interpretable.
>
> Computational costs documented in appendix "Experimental Hyperparameters": polynomial time `O(t_terms × n_samples × n_features²)` for low-degree polynomials. Monomial growth exponential with degree, necessitating careful degree selection (2-3 in experiments).
>
> **Q5: Recent Competitors (RAG-SR, ParFam, MetaSymNet)**
>
> We thank the reviewer. We revised Related Work (Section 3, blue) clarifying critical distinction.
>
> Bitween uses SR methods as computational backends, not competitors. Task differs fundamentally: we seek RSR properties for *known* functions, not approximate unknown functions.
>
> Backend architecture:
> - Vanilla Bitween (`V-Bitween`) employs SR methods (PySR, GPLearn, MILP, LR) as backends within our regression-based learning framework to discover polynomial relationships among correlated query evaluations
> - Agentic Bitween (`A-Bitween`) uses Vanilla Bitween's inference and verification tools to propose and verify properties with novel query functions
>
> We cited RAG-SR, ParFam, MetaSymNet-concurrent developments potentially integrable as alternative backends. Not compared because: (1) concurrent publication (ICLR/AAAI 2025), (2) traditional SR backends often timeout or produce approximations insufficient for formal verification (require exact rational coefficients). Effectiveness for RSR coefficient fitting remains to be evaluated; we commit to future exploration.
>
> **Q6: Downstream Applications**
>
> Discovered RSRs enable concrete applications:
>
> 1. Instance hiding protocols (Goldwasser-Micali): Sigmoid RSR enables secure computation where weak device computes `σ(x)` via powerful servers computing `σ(r)` and `σ(x+r)` without revealing `x`. Demonstrated in our motivating example.
>
> 2. Self-correcting programs: RSRs transform programs correct on most inputs to correct on all inputs with high probability (Blum et al.), enabling fault tolerance computation.
>
> 3. Complexity theory: Applications in average-case complexity reductions and interactive proof systems (foundational work by Goldwasser, Blum, Rubinfeld cited in Introduction).
>
> 4. Physical attack protection (Erata et al. 2024, arXiv:2405.05193v1): The authors showed that RSR properties systematically protect cryptographic implementations against power side-channel and fault injection attacks. Applications: RSA-CRT signatures and Kyber (post-quantum) key generation. Results: power leakage reduced by 2 orders of magnitude, fault reduction 95.4% average (modular multiplication 99.4%, Kyber 97.7%). Algorithm-independent, black-box protection at operation level.

---

### Official Review · Reviewer_FY2m · 2025-11-01

**Soundness:** 2
**Presentation:** 1
**Contribution:** 2
**Rating:** 2
**Confidence:** 3

**Summary:**

The paper proposes Bitween, an algorithm for learning randomized self-reductions (RSRs). The authors evaluate their method on a suite of 80 scientific and machine learning functions, demonstrating that Bitween can successfully learn RSRs for a significant portion of these functions. The authors also introduce Agentic Bitween, which integrates large language models (LLMs) into the Bitween framework to enhance RSR discovery. Experimental results indicate that Agentic Bitween outperforms all shown baselines and solves some cases previously believed intractable.

**Strengths:**

1. The problem is well-presented, well-motivated, and establishes a valuable connection to the well-known field of PAC learning.
2. The authors dedicate significant effort to building a theoretical foundation, stating assumptions and providing formal proofs.
3. The core problem of learning RSRs is interesting and holds potential for broader applications.

**Weaknesses:**

1. The theoretical foundation is riddled with numerous formatting and referencing errors. These mistakes significantly detract from the readability and undermine the mathematical rigor, which is a key claimed contribution. Examples include citation errors (e.g., L983), vague self-references (e.g., Section 2 referring to itself in L88), and, most critically, incorrect cross-references within the theoretical sections. The text refers to "Theorem 5" when it means "Definition 5", and similar errors occur on L766 (confusing Theorem and Claim) and L775 (confusing Theorem and Remark). Such mistakes make the formalisms difficult to follow. While a diligent reader might eventually decipher the intended meaning, the burden of clarity is on the authors.

2. It is seriously confusing what you refer to as "V-Bitween" versus baselines like PySR, GPLearn, and MILP. At one point, such as L071 (in the contributions), you treat them as separate methods, while in the experiments section, you say that V-Bitween uses these baselines as backends and Figure 2 names them with the "V-Bitween-" prefix. Is Vanilla Bitween the same thing as V-Bitween? This needs to be clarified.

3. Some of the results are badly reported/presented. Examples include:
* Section 6 reports results in text form (e.g., the contribution 1 paragraph) that would be much better suited for tables or plots.
* Figure 2 (left) reports "Average Verified RSRs by Function Category": I assume that means you average the number of RSRs found and verified for each function over the respective category. This is a bit strange to me. Why do you attempt multiple RSRs per function? Shouldn't one be enough? Wouldn't it make more sense to report the fraction of functions for which at least one RSR was found and verified?
* Why are the RSR Rate and Verification Rate reported separately in Figure 2 (right)? Shouldn't they be the same, since only verified RSRs should count?
* What is Function Coverage in Figure 2 (right)? This is not clearly defined in the text. How do you measure it? Is it coverage over individual functions or function categories?
* In Section 6, you say "Figure 3 demonstrates that this breakthrough stems from Agentic Bitween’s intensive use of verification and inference tools across all function categories." Looking at Figure 3, I see no evidence for this claim. It merely plots a heat map of "tool calls" between different A-Bitween variants (different LLMs). It does not correlate tool usage with performance at all.

4. Table 1 lists "Novel Query **Functions** Discovered by Agentic Bitween," but the entries appear to be equations rather than functions. Maybe I'm missing something, but this is rather confusing.

5. The paper claims Agentic-Bitween as its breakthrough contribution, yet it is hardly described in the main text other than saying that it uses an LLM to generate RSRs then verifies them symbolically. At the very least, an outline diagram of the architecture or a more detailed description of its operation should be provided in the main paper.

6. L406-408: "Larger models exhibit increased reasoning depth, with more sophisticated analysis leading to higher token usage but correspondingly better RSR **discovery quality**." What does "RSR discovery quality" mean here? Are some RSRs of higher quality than others? If so, how is this measured? This is not discussed anywhere in the paper.

7. L411-413: "The tool-based reasoning approach, while more expensive, enables systematic exploration of novel mathematical relationships **impossible** with pure neural reasoning." This is a very strong claim that requires proper backing. Why is it impossible with pure neural reasoning?

## Minor Issues
1. Why are all citations rendered as text citations (e.g., \citet{})? This appears to be a formatting error.

2. Vague references like "in Section 2" (e.g., L88 & L103) are not very helpful. Please refer to the specific labels for equations, figures, and subsections.

3. The citation in L344 for Claude 4 points to a paper about Claude 3.

4. Broken citations/references to F* and VeriFast (L983).

**Questions:**

I already raised several questions in the weaknesses section. Below are some additional ones:
1. Is the Bitween algorithm (Algorithm 1) sound? I cannot find a clear failure case in the algorithm definition. Does it simply return the same function as the input if no RSR is found?
2. Could you please explain, in one place, a proper outline of Agentic-Bitween? How is it structured? My current understanding is that it uses queries to generate potential query functions for the main algorithm. Is this correct? What symbolic reasoning tools does the LLM have access to, and how are they integrated?
3. I see the fixed query function set $\\{x + r, x − r, x · r, x, r\\}$ includes $x$, doesn't this defeat the purpose of the reduction since it is meant to derive $f(x)$ without directly evaluating $f$ at $x$?

---

> ### Author Response · Authors · 2025-11-18
>
> ## Presentation Issues - Fixed
>
> **Theoretical formatting/referencing errors**
>
> We corrected all identified errors: citation errors (L983), vague self-references (L88, L103), incorrect cross-references (Theorem vs Definition, Theorem vs Claim, Theorem vs Remark). All section references now use specific labels.
>
> **Minor Issues - All Fixed**
>
> Fixed all four issues listed in "Minor Issues" section: All `\citet{}` citation formatting corrected throughout paper, Vague "Section 2" references (L88, L103) replaced with specific equation/figure/subsection labels, Claude 4 citation (L344) corrected to proper reference, F* and VeriFast citations (L983) fixed and included.
>
> ## V-Bitween Terminology - Clarified Throughout
>
> **V-Bitween vs baselines confusion**
>
> We clarified throughout the paper (Abstract, Introduction, Related Work, Implementation, Evaluation: all blue). Vanilla Bitween = V-Bitween. V-Bitween uses different backends (hence suffixes): `V-Bitween-LR` (linear regression, our primary method), `V-Bitween-PySR`, `V-Bitween-GPLearn`, `V-Bitween-MILP`. All integrated in our framework; we compare backends for RSR discovery with fixed query functions. See Implementation section's "Three-Tier Framework" for details.
>
> ## Agentic Bitween Architecture - Added to Paper
>
> **Agentic architecture description**
>
> `Vanilla Bitween` proves approach works, outperforms backends, provides inference and verification engine. `Agentic Bitween` shows possibilities without query constraints, reveals query discovery as bottleneck, validates modular architecture using V-Bitween as tool.
>
> Added detailed description in Implementation (L300-313, blue). `A-Bitween` has three tools: (1) `infer_property_tool`: wrapper over `V-Bitween` taking functional terms encoding query functions (e.g., `f(x/(1+x))` with query `x/(1+x)`), (2) `symbolic_verify_tool`: verifies zero-equality using SymPy, (3) `sequential_thinking_tool`: empirically found to help LLM reasoning and RSR discovery.
>
> **Table 1 functional terms vs functions**
>
> Clarified in paper. Query functions appear inside functional terms. Example: `f(x+log(k))` has query function `q(x,k)=x+log(k)`. `A-Bitween` provides complete functional terms to `V-Bitween`; query functions implicit. Table shows complete properties (equations=0) to give context for implicit query functions.
>
> ## Results Presentation
>
> **Text vs tables for results**
>
> Updated Section 6. Results now better reflected in heatmap and radar. Text describes what appears in plots, avoiding redundancy.
>
> **Multiple RSRs per function**
>
> Multiple RSRs beneficial from application perspective. For power side-channel analysis: first-order attacks use `f(x) = f(x+r) - f(r)`, but second-order attacks require `f(x) = f(x+r+s) - f(r) - f(s)` (more robust). For complex non-linear functions, multiple RSRs provide defense-in-depth. Tables 4-7 in appendix show different RSR counts with varied success rates.
>
> **RSR Rate vs Verification Rate**
>
> Updated Figure 2. For `V-Bitween` variants, they're identical (fixed query set yields RSR if successful). For `A-Bitween`, verification rate includes non-RSR properties (properties with only f(x), tautologies, duplicates)-manually filtered. Verification rate shows volume of returned properties; RSR Rate shows verified RSRs after manual filtering.
>
> **Function Coverage definition**
>
> Updated Figure 2. Term now "Function RSR Coverage": percentage of individual functions (benchmarks) for which method returned at least one verified RSR.
>
> **Figure 3 tool usage correlation**
>
> Clarified in paper. Figure 3 shows Bitween tool usage increase across `A-Bitween` variants. This correlates with average verified RSRs in Figure 2 heatmap. Higher tool usage → more RSR discoveries, demonstrating tool-based reasoning's value.
>
> ## Technical Clarifications
>
> **"RSR discovery quality" (L406-408)**
>
> Removed misleading claim. We empirically showed more diverse query functions and properties. RSR quality can be justified application-specific (e.g., cryptography: fewer terms = better computational efficiency).
>
> **"Impossible with pure neural reasoning" (L411-413)**
>
> Removed unjustified claim. Tool-based approach has advantage: (1) one tool call verifies correctness, (2) avoids in-context verification (often fails). Tables 6-7 (appendix) show pure neural (`D-Research`) variants produce more false positives than tool-based (`A-Bitween`), especially on harder benchmarks [41-60].
>
> **Algorithm soundness**
>
> Corrected Algorithm 1: returns empty tuple if no RSR found (not the input function).
>
> **Query function set includes x**
>
> For learning, we sample from all query functions including `f(x), f(r), f(x+r)` to perform regression. Mathematically, property `f(x) = f(x+r) - f(r)` means we can compute `f(x)` from `f(x+r)` and `f(r)`. During learning (regression), we don't know which query functions are dependent, thus we need samples from all terms and construct hypothesis space.

---

### Official Review · Reviewer_gpS3 · 2025-11-01

**Soundness:** 3
**Presentation:** 2
**Contribution:** 2
**Rating:** 4
**Confidence:** 3

**Summary:**

This study proposes a method for learning RSR for mathematical functions. It introduces Vanillan Bitween, a linear-regression–based learner and A-Bitween, an agentic LLM-driven variant that can discover new query functions beyond the classic ones. An RSR-Bench with 80 functions is proposed and used for evaluation of the method.

**Strengths:**

The paper introduces an interesting framework for RSRs, which potentially could be highly impactful.  The proposed Bitween combines sparse linear regression with formal verification, and is further extended to the neuro-symbolic A-Bitween.  The concept and the methodology are novel, and appear effective as argued in the paper.

The method's theoretical foundation is sound, with a clear connection to PAC learning. The proposed RSR-Bench could be a valueable contribution to the field.  Using 80 benchmark functions, the study validates performance against baselines PySR, GPLearn, and MILP.  Overall, the work reflects a substantial and well-motivated research effort.

**Weaknesses:**

The empirical comparison is not clearly presented.  It is written that "Vanilla Bitween surpasses traditional symbolic methods within the fixed query function paradigm, discovering 76 total verified RSRs compared to PySR’s 54, GP-Learn’s 47, and MILP’s 64".   The details should be provided.  The comparisons focus on PySR, GPLearn, and MILP, while other methods, e.g. AI Feynman and DSR are mentioned but not used.  Justification should be provided for that.  In addition, the detailed settings of other methods are not provided.  For example, GP is known to be a strong symbolic regressor, but its performance varies significantly with factors such as population size, number of generations, function set, and several other parameters. Without specifying these details, the comparision won't be convincing.

The proposed benchmark of 80 functions need to be better described and justifed.  For broader adoption, the coverage should be as comrehensive as possible, e.g. involving piecewise, discontinuous, or domain-restricted functions. Are these functions part of the consideration? Argubly, sigmoid functions are kind of toy problems.  Would it possible to add some end-to-end large functions/systems?

It’s unclear how LR's performance would scale with input dimension, e.g. number of queries. The scaling behavior/limits are not quantified. Also the study assumes uniformly samplable domains and relies on uniform draws. Would that limit its performance and robustness over more realistic and non-uniform distributions?

The introduction of the agentic component may dilute the focus. The paper could benefit from first establishing a more comprehensive empirical foundation, addressing the concerns mentioned above, e.g. scalability, sampling, and benchmark coverage.  Once a strong foundation is estalished, agentic-bitween can then be pursued as future work.

In Algorithm 1, maximum denominator constraint is not specified. What is the upper bound?

**Questions:**

Please see above.

---

> ### Author Response · Authors · 2025-11-18
>
> **Q1: Empirical Comparison Details**
>
> We added per-category analysis to Evaluation (after Contribution 1, blue) providing the requested details.
>
> The visual comparison in Figure 2 shows average verified RSRs by function category (left panel) and function RSR coverage (right panel). The underlying numbers are: V-Bitween-LR discovered 76 total RSRs with 52.5% function RSR coverage. Alternative backends: V-Bitween-MILP 64 RSRs (45% coverage), V-Bitween-PySR 54 RSRs (35% coverage), V-Bitween-GPLearn 47 RSRs (33.75% coverage).
>
> Complete per-benchmark statistics are in appendix tables.
>
> **Q2: Why PySR/GPLearn/MILP not AI Feynman/DSR**
>
> PySR supersedes AI Feynman/DSR (Cranmer 2023, arXiv:2305.01582 Tables 1&3). Authors note both "could not be configured despite significant effort". We experienced same.
>
> We clarified in our revised Related Work section as follows: we use symbolic regression methods as computational backends, not as competing approaches. Our task of discovering RSR properties differs fundamentally from symbolic regression-we seek randomized self-reduction properties for known mathematical functions, not approximate unknown functions from data.
>
> Main intuition: we require **exact RSR properties** for formal verification. SR methods generate numerical approximations (e.g., `f(x) ≈ 0.9987·f(x+r)·f(r) - 0.4912·f(x+r)`) unsuitable for symbolic verification. Our linear regression backend with rational approximation produces verifiable expressions like  (e.g., `f(x) = (1/2)·f(x+r) + (3/5)·f(r)`).
>
> Selected PySR/GPLearn/MILP as distinct paradigms (evolutionary/genetic/optimization), integrated. Recent methods (RAG-SR, ParFam, MetaSymNet-cited) could be future backends.
>
> **Q3: Hyperparameter settings for Symbolic Regression Backends**
>
> Appendix "Experimental Hyperparameters" has detailed tables now. Common: function set (add,sub,mul), degree 2-3, ε=0.001, n=30, domain [-10,10], max denominator=20, 80/20 split, ~12h budget. V-Bitween-PySR: 50 iter, ops=[+,×]. V-Bitween-GPLearn: pop=1000, gen=20. V-Bitween-MILP: Gurobi. V-Bitween-LR: grid search (Linear/Ridge/Lasso), 5-fold CV, R². Philosophy: equal budgets, no per-function tuning, established defaults.
>
> **Q4: Benchmark Coverage (piecewise, discontinous, domain-restricted functions)**
>
> We clarified RSR-Bench's coverage in Evaluation. The 80 functions span: piecewise (ReLU, Leaky ReLU), domain-restricted (log, sqrt, inverse-Bitween supports domain constraints per Definition 2), continuous smooth functions (basic arithmetic, trigonometric, hyperbolic, exponential, logarithmic, ML activation functions, loss functions), and special mathematical functions (gamma, error function, Gudermannian). Discontinuous functions are excluded because our framework (Definition 2) requires marginal uniformity in sampling, and extending to fully discontinuous domains would require new theoretical development.
>
> Sigmoid RSR `σ(x) = σ(x+r)(σ(r)-1)/(2σ(x+r)σ(r)-σ(x+r)-σ(r))` is **first reported property** in functional equations/RSR literature despite decades of ML use.
>
> Our 80 functions provide comprehensive coverage relevant to machine learning, scientific computing, and cryptography (foundational work by Blum, Rubinfeld, Goldwasser-Micali).
>
> **Q5: Scalability**
>
> Implementation shows polynomial time `O(t_terms × n_samples × n_features²)` for low-degree polynomials. Monomial growth is exponential with degree, necessitating careful degree (2-5) and query selection-why A-Bitween's intelligent selection is valuable. Performance decreases as queries increase (combinatorial explosion). Used n=30 samples (standard). Uniform sampling assumed for tractability per `Definitions 2&3`. Non-uniform would require new theory.
>
> **Q6: Agentic Component**
>
> The two variants tell a complete story. `Vanilla Bitween` proves approach works, outperforms backends, provides inference and verification engine. `Agentic Bitween` shows possibilities without query constraints, reveals query discovery as bottleneck, validates modular architecture using V-Bitween as tool.
>
> Agentic Bitween directly addresses the reviewer's scalability concerns by scaling RSR discovery to complex query functions involving derivatives, integrals, and domain-specific transformations. It discovers functions like `f(x/(1+x))`, `f(x+log(k))`, `atan(sinh(x))` that humans wouldn't manually try. This IS the scalability solution: letting LLMs intelligently explore the infinite query space.
>
> With only V-Bitween, readers would ask "Why only test fixed queries?".
>
>
> **Q7: Max Denominator Constraint**
>
> Algorithm 1 line 8: 20 (configurable). Converts floats to exact rationals balancing interpretability with precision. See appendix.

---

### Author Response · Authors · 2025-11-18

We are grateful to the reviewers for their thorough and constructive reviews. We have addressed all questions and concerns through extensive revisions (all changes highlighted in blue) throughout the paper, including clarifying terminology, expanding the Agentic Bitween architecture description, revising all evaluation figures with additional results and per-category analysis, adding a new hyperparameters appendix, and revising the Related Work section to better position our contributions.

---

### Note · Program_Chairs · 2026-01-17
**Submission Desk Rejected by Program Chairs**

The following references in this submission do not refer to real documents and/or have major errors in bibliographic information:

 Shlomo Hoory, Amir Feder, Aviya Tendler, Sofia Cohen, Sofia Erell, Itay Laish, Hootan Nakhost,
Uri Stemmer, Ayelet Benjamini, Avinatan Hassidim, et al. Learning to correct errors in large
language models. arXiv preprint arXiv:2402.05865, 2024.